# SSRGD: Simple Stochastic Recursive Gradient Descent for Escaping Saddle Points

**Zhize Li**
Tsinghua University, China and KAUST, Saudi Arabia
zhizeli.thu@gmail.com

## Abstract

We analyze stochastic gradient algorithms for optimizing nonconvex problems. In particular, our goal is to find local minima (second-order stationary points) instead of just finding first-order stationary points which may be some bad unstable saddle points. We show that a simple perturbed version of stochastic recursive gradient descent algorithm (called SSRGD) can find an $(\epsilon, \delta)$-second-order stationary point with $\widetilde{O}(\sqrt{n}/\epsilon^2 + \sqrt{n}/\delta^4 + n/\delta^3)$ stochastic gradient complexity for nonconvex finite-sum problems. As a by-product, SSRGD finds an $\epsilon$-first-order stationary point with $O(n + \sqrt{n}/\epsilon^2)$ stochastic gradients. These results are almost optimal since Fang et al. [11] provided a lower bound $\Omega(\sqrt{n}/\epsilon^2)$ for finding even just an $\epsilon$-first-order stationary point. We emphasize that SSRGD algorithm for finding second-order stationary points is as simple as for finding first-order stationary points just by adding a uniform perturbation sometimes, while all other algorithms for finding second-order stationary points with similar gradient complexity need to combine with a negative-curvature search subroutine (e.g., Neon2 [4]). Moreover, the simple SSRGD algorithm gets a simpler analysis. Besides, we also extend our results from nonconvex finite-sum problems to nonconvex online (expectation) problems, and prove the corresponding convergence results.

## 1 Introduction

Nonconvex optimization is ubiquitous in machine learning applications especially for deep neural networks. For convex optimization, every local minimum is a global minimum and it can be achieved by any first-order stationary point, i.e., $\nabla f(x) = 0$. However, for nonconvex problems, the point with zero gradient can be a local minimum, a local maximum or a saddle point. To avoid converging to bad saddle points (including local maxima), we want to find a second-order stationary point, i.e., $\nabla f(x) = 0$ and $\nabla^2 f(x) \succeq 0$ (this is a necessary condition for $x$ to be a local minimum). All second-order stationary points indeed are local minima if function $f$ satisfies strict saddle property [12]. Note that finding the global minimum in nonconvex problems is NP-hard in general. Also note that it was shown that all local minima are also global minima for some nonconvex problems, e.g., matrix sensing [5], matrix completion [13], and some neural networks [14]. Thus, our goal in this paper is to find an approximate second-order stationary point (local minimum) with proved convergence.

There has been extensive research for finding $\epsilon$-first-order stationary point (i.e., $\|\nabla f(x)\| \leq \epsilon$), e.g., GD, SGD and SVRG. See Table 1 for an overview. Although Xu et al. [33] and Allen-Zhu and Li [4] independently proposed reduction algorithms Neon/Neon2 that can be combined with previous $\epsilon$-first-order stationary points finding algorithms to find an $(\epsilon, \delta)$-second-order stationary point (i.e., $\|\nabla f(x)\| \leq \epsilon$ and $\lambda_{\min}(\nabla^2 f(x)) \geq -\delta$). However, algorithms obtained by this reduction are very complicated in practice, and they need to extract negative curvature directions from the Hessian to escape saddle points by using a negative curvature search subroutine: given a point $x$, find an

approximate smallest eigenvector of $\nabla^2 f(x)$. This also involves a more complicated analysis. Note that in practice, standard first-order stationary point finding algorithms can often work (escape bad saddle points) in nonconvex setting without a negative curvature search subroutine. The reason may be that the saddle points are usually not very stable. So there is a natural question:

"Is there any simple modification to allow first-order stationary point finding algorithms to get a theoretical second-order guarantee?".

Table 1: Gradient complexity of optimization algorithms for nonconvex finite-sum problem (1)

| Algorithm | Stochastic gradient complexity | Guarantee | Negative-curvature search subroutine |
|---|---|---|---|
| GD [24] | $O(\frac{n}{\epsilon^2})$ | 1st-order | No |
| SVRG [28, 3], SCSG [22], SVRG+ [23] | $O(n + \frac{n^{2/3}}{\epsilon^2})$ | 1st-order | No |
| SNVRG [35], SPIDER [11], SpiderBoost [32], SARAH [27] | $O(n + \frac{n^{1/2}}{\epsilon^2})$ | 1st-order | No |
| SSRGD (this paper) | $O(n + \frac{n^{1/2}}{\epsilon^2})$ | 1st-order | No |
| PGD [18] | $\widetilde{O}(\frac{n}{\epsilon^2} + \frac{n}{\delta^4})$ | 2nd-order | No |
| Neon2+FastCubic/CDHS [1, 6] | $\widetilde{O}(\frac{n}{\epsilon^{1.5}} + \frac{n}{\delta^3} + \frac{n^{3/4}}{\epsilon^{1.75}} + \frac{n^{3/4}}{\delta^{3.5}})$ | 2nd-order | Needed |
| Neon2+SVRG [4] | $\widetilde{O}(\frac{n^{2/3}}{\epsilon^2} + \frac{n}{\delta^3} + \frac{n^{3/4}}{\delta^{3.5}})$ | 2nd-order | Needed |
| Stabilized SVRG [15] | $\widetilde{O}(\frac{n^{2/3}}{\epsilon^2} + \frac{n}{\delta^3} + \frac{n^{2/3}}{\delta^4})$ | 2nd-order | No |
| SNVRG$^+$+Neon2 [34] | $\widetilde{O}(\frac{n^{1/2}}{\epsilon^2} + \frac{n}{\delta^3} + \frac{n^{3/4}}{\delta^{3.5}})$ | 2nd-order | Needed |
| SPIDER-SFO$^+$(+Neon2) [11] | $\widetilde{O}(\frac{n^{1/2}}{\epsilon^2} + \frac{n^{1/2}}{\epsilon\delta^2} + \frac{1}{\epsilon\delta^3} + \frac{1}{\delta^5})$ | 2nd-order | Needed |
| SSRGD (this paper) | $\widetilde{O}(\frac{n^{1/2}}{\epsilon^2} + \frac{n^{1/2}}{\delta^4} + \frac{n}{\delta^3})$ | 2nd-order | No |

Table 2: Gradient complexity of optimization algorithms for nonconvex online problem (2)

| Algorithm | Stochastic gradient complexity | Guarantee | Negative-curvature search subroutine |
|---|---|---|---|
| SGD [16] | $O(\frac{1}{\epsilon^4})$ | 1st-order | No |
| SCSG [22]; SVRG+ [23] | $O(\frac{1}{\epsilon^{3.5}})$ | 1st-order | No |
| SNVRG [35]; SPIDER [11]; SpiderBoost [32]; SARAH [27] | $O(\frac{1}{\epsilon^3})$ | 1st-order | No |
| SSRGD (this paper) | $O(\frac{1}{\epsilon^3})$ | 1st-order | No |
| Perturbed SGD [12] | $\mathrm{poly}(d, \frac{1}{\epsilon}, \frac{1}{\delta})$ | 2nd-order | No |
| CNC-SGD [8] | $\widetilde{O}(\frac{1}{\epsilon^4} + \frac{1}{\delta^{10}})$ | 2nd-order | No |
| Neon2+SCSG [4] | $\widetilde{O}(\frac{1}{\epsilon^{10/3}} + \frac{1}{\epsilon^2\delta^3} + \frac{1}{\delta^5})$ | 2nd-order | Needed |
| Neon2+Natasha2 [2] | $\widetilde{O}(\frac{1}{\epsilon^{3.25}} + \frac{1}{\epsilon^3\delta} + \frac{1}{\delta^5})$ | 2nd-order | Needed |
| SNVRG$^+$+Neon2 [34] | $\widetilde{O}(\frac{1}{\epsilon^3} + \frac{1}{\epsilon^2\delta^3} + \frac{1}{\delta^5})$ | 2nd-order | Needed |
| SPIDER-SFO$^+$(+Neon2) [11] | $\widetilde{O}(\frac{1}{\epsilon^3} + \frac{1}{\epsilon^2\delta^2} + \frac{1}{\delta^5})$ | 2nd-order | Needed |
| SSRGD (this paper) | $\widetilde{O}(\frac{1}{\epsilon^3} + \frac{1}{\epsilon^2\delta^3} + \frac{1}{\epsilon\delta^4})$ | 2nd-order | No |

**Note:** 1. Guarantee (see Definition 1): $\epsilon$-first-order stationary point $\|\nabla f(x)\| \leq \epsilon$; $(\epsilon, \delta)$-second-order stationary point $\|\nabla f(x)\| \leq \epsilon$ and $\lambda_{\min}(\nabla^2 f(x)) \geq -\delta$.

2. In the classical setting where $\delta = O(\sqrt{\epsilon})$ [25, 18], our simple SSRGD is always (no matter what $n$ and $\epsilon$ are) not worse than all other algorithms (in both Table 1 and 2) except FastCubic/CDHS (which need to compute Hessian-vector product) and SPIDER-SFO$^+$. Moreover, our simple SSRGD is not worse than FastCubic/CDHS if $n \geq 1/\epsilon$ and is better than SPIDER-SFO$^+$ if $\delta$ is very small (e.g., $\delta \leq 1/\sqrt{n}$) in Table 1.

**Algorithm 1** Simple Stochastic Recursive Gradient Descent (SSRGD)

---

**Input:** initial point $x_0$, epoch length $m$, minibatch size $b$, step size $\eta$, perturbation radius $r$, threshold gradient $g_{\text{thres}}$

1: **for** $s = 0, 1, 2, \ldots$ **do**
2:     **if** not currently in a super epoch and $\|\nabla f(x_{sm})\| \leq g_{\text{thres}}$ **then**
3:         $x_{sm} \leftarrow x_{sm} + \xi$, where $\xi$ uniformly $\sim \mathbb{B}_0(r)$, start a super epoch
        // we use super epoch since we do not want to add the perturbation too often near a saddle point
4:     **end if**
5:     $v_{sm} \leftarrow \nabla f(x_{sm})$
6:     **for** $k = 1, 2, \ldots, m$ **do**
7:         $t \leftarrow sm + k$
8:         $x_t \leftarrow x_{t-1} - \eta v_{t-1}$
9:         $v_t \leftarrow \frac{1}{b} \sum_{i \in I_b} \left( \nabla f_i(x_t) - \nabla f_i(x_{t-1}) \right) + v_{t-1}$
        // $I_b$ are i.i.d. uniform samples with $|I_b| = b$
10:         **if** meet stop condition **then** stop super epoch
11:     **end for**
12: **end for**

---

For gradient descent (GD), Jin et al. [18] showed that a simple perturbation step is enough to escape saddle points for finding a second-order stationary point, and this is necessary [10]. Very recently, Ge et al. [15] showed that a simple perturbation step is also enough to find a second-order stationary point for SVRG algorithm [23]. Moreover, Ge et al. [15] also developed a stabilized trick to further improve the dependency of Hessian Lipschitz parameter.

## 1.1 Our Contributions

In this paper, we propose a simple SSRGD algorithm (described in Algorithm 1) showed that a simple perturbation step is enough to find a second-order stationary point for stochastic recursive gradient descent algorithm. Our results and previous results are summarized in Table 1 and 2. We would like to highlight the following points:

- We improve the result in [15] to the almost optimal one (i.e., from $n^{2/3}/\epsilon^2$ to $n^{1/2}/\epsilon^2$) since Fang et al. [11] provided a lower bound $\Omega(\sqrt{n}/\epsilon^2)$ for finding even just an $\epsilon$-first-order stationary point. Note that for the other two $n^{1/2}$ algorithms (i.e., SNVRG$^+$ and SPIDER-SFO$^+$), they both need the negative curvature search subroutine (e.g. Neon2) thus are more complicated in practice and in analysis compared with their first-order guarantee algorithms (SNVRG and SPIDER), while our SSRGD is as simple as its first-order guarantee algorithm just by adding a uniform perturbation sometimes.

- For more general nonconvex online (expectation) problems (2), we obtain the first algorithm which is as simple as finding first-order stationary points for finding a second-order stationary point with similar state-of-the-art convergence result. See the last column of Table 2.

- Our simple SSRGD algorithm gets simpler analysis. Also, the result for finding a first-order stationary point is a by-product from our analysis. We also give a clear interpretation to show why our analysis for SSRGD algorithm can improve the original SVRG from $n^{2/3}$ to $n^{1/2}$ in Section 5.1. We believe it is very useful for better understanding these two algorithms.

## 2 Preliminaries

**Notation:** Let $[n]$ denote the set $\{1, 2, \cdots, n\}$ and $\|\cdot\|$ denote the Eculidean norm for a vector and the spectral norm for a matrix. Let $\langle u, v \rangle$ denote the inner product of two vectors $u$ and $v$. Let $\lambda_{\min}(A)$ denote the smallest eigenvalue of a symmetric matrix $A$. Let $\mathbb{B}_x(r)$ denote a Euclidean ball with center $x$ and radius $r$. We use $O(\cdot)$ to hide the constant and $\widetilde{O}(\cdot)$ to hide the polylogarithmic factor.

In this paper, we consider two types of nonconvex problems. The finite-sum problem has the form

$$\min_{x \in \mathbb{R}^d} f(x) := \frac{1}{n} \sum_{i=1}^n f_i(x), \tag{1}$$

where $f(x)$ and all individual $f_i(x)$ are possibly nonconvex. This form usually models the empirical risk minimization in machine learning problems.

The online (expectation) problem has the form

$$\min_{x \in \mathbb{R}^d} f(x) := \mathbb{E}_{\zeta \sim D}[F(x, \zeta)], \tag{2}$$

where $f(x)$ and $F(x, \zeta)$ are possibly nonconvex. This form usually models the population risk minimization in machine learning problems.

Now, we make standard smoothness assumptions for these two problems.

**Assumption 1 (Gradient Lipschitz)**      *1. For finite-sum problem* (1)*, each $f_i(x)$ is differentiable and has L-Lipschitz continuous gradient, i.e.,*

$$\|\nabla f_i(x_1) - \nabla f_i(x_2)\| \le L\|x_1 - x_2\|, \quad \forall x_1, x_2 \in \mathbb{R}^d. \tag{3}$$

   *2. For online problem* (2)*, $F(x, \zeta)$ is differentiable and has L-Lipschitz continuous gradient, i.e.,*

$$\|\nabla F(x_1, \zeta) - \nabla F(x_2, \zeta)\| \le L\|x_1 - x_2\|, \quad \forall x_1, x_2 \in \mathbb{R}^d. \tag{4}$$

**Assumption 2 (Hessian Lipschitz)**      *1. For finite-sum problem* (1)*, each $f_i(x)$ is twice-differentiable and has $\rho$-Lipschitz continuous Hessian, i.e.,*

$$\|\nabla^2 f_i(x_1) - \nabla^2 f_i(x_2)\| \le \rho\|x_1 - x_2\|, \quad \forall x_1, x_2 \in \mathbb{R}^d. \tag{5}$$

   *2. For online problem* (2)*, $F(x, \zeta)$ is twice-differentiable and has $\rho$-Lipschitz continuous Hessian, i.e.,*

$$\|\nabla^2 F(x_1, \zeta) - \nabla^2 F(x_2, \zeta)\| \le \rho\|x_1 - x_2\|, \quad \forall x_1, x_2 \in \mathbb{R}^d. \tag{6}$$

These two assumptions are standard for finding first-order stationary points (Assumption 1) and second-order stationary points (Assumption 1 and 2) for all algorithms in both Table 1 and 2.

Now we define the approximate first-order stationary points and approximate second-order stationary points.

**Definition 1** *$x$ is an $\epsilon$-first-order stationary point for a differentiable function $f$ if*

$$\|\nabla f(x)\| \le \epsilon. \tag{7}$$

*$x$ is an $(\epsilon, \delta)$-second-order stationary point for a twice-differentiable function $f$ if*

$$\|\nabla f(x)\| \le \epsilon \ \text{ and } \ \lambda_{\min}(\nabla^2 f(x)) \ge -\delta. \tag{8}$$

The definition of $(\epsilon, \delta)$-second-order stationary point is the same as [4, 8, 34, 11] and it generalizes the classical version where $\delta = \sqrt{\rho\epsilon}$ used in [25, 18, 15].

## 3   Simple Stochastic Recursive Gradient Descent

In this section, we propose the simple stochastic recursive gradient descent algorithm called SSRGD. The high-level description (which omits the stop condition details in Line 10) of this algorithm is in Algorithm 1 and the full algorithm (containing the stop condition) is described in Algorithm 2. Compared with the high-level Algorithm 1, the only difference is that Algorithm 2 contains the stop condition of super epoch (Line 13–14 of Algorithm 2) and the random stop of epoch (Line 15–16 of Algorithm 2). Note that we call each outer loop an ***epoch***, i.e., iterations $t$ from $sm$ to $(s+1)m$ for an epoch $s$. We call the iterations between the beginning of perturbation and end of perturbation a ***super epoch***.

**Algorithm 2** Simple Stochastic Recursive Gradient Descent (SSRGD)

---

**Input:** initial point $x_0$, epoch length $m$, minibatch size $b$, step size $\eta$, perturbation radius $r$, threshold gradient $g_{\text{thres}}$, threshold function value $f_{\text{thres}}$, super epoch length $t_{\text{thres}}$

1: $super\_epoch \leftarrow 0$
2: **for** $s = 0, 1, 2, \ldots$ **do**
3:  **if** $super\_epoch = 0$ and $\|\nabla f(x_{sm})\| \leq g_{\text{thres}}$ **then**
4:    $super\_epoch \leftarrow 1$   // start a super epoch near a saddle point
5:    $\widetilde{x} \leftarrow x_{sm}, t_{\text{init}} \leftarrow sm$
6:    $x_{sm} \leftarrow \widetilde{x} + \xi$, where $\xi$ uniformly $\sim \mathbb{B}_0(r)$
7:  **end if**
8:  $v_{sm} \leftarrow \nabla f(x_{sm})$
9:  **for** $k = 1, 2, \ldots, m$ **do**
10:    $t \leftarrow sm + k$
11:    $x_t \leftarrow x_{t-1} - \eta v_{t-1}$
12:    $v_t \leftarrow \frac{1}{b} \sum_{i \in I_b} \left( \nabla f_i(x_t) - \nabla f_i(x_{t-1}) \right) + v_{t-1}$
      // $I_b$ are i.i.d. uniform samples with $|I_b| = b$
13:    **if** $super\_epoch = 1$ and $(f(\widetilde{x}) - f(x_t) \geq f_{\text{thres}}$  or  $t - t_{\text{init}} \geq t_{\text{thres}})$ **then**
14:      $super\_epoch \leftarrow 0$; break
15:    **else if** $super\_epoch = 0$ **then**
16:      break with probability $\frac{1}{m-k+1}$
        // we use random stop since we want to randomly choose a point as the starting point of the next epoch
17:    **end if**
18:  **end for**
19:   $x_{(s+1)m} \leftarrow x_t$
20: **end for**

---

The SSRGD algorithm is based on the stochastic recursive gradient descent which is introduced in [26] for convex optimization. In particular, Nguyen et al. [26] want to save the storage of past gradients in SAGA [9] by using the recursive gradient. However, this stochastic recursive gradient descent is widely used in recent work for nonconvex optimization such as SPIDER [11], SpiderBoost [32] and some variants of SARAH (e.g., ProxSARAH [27]).

Recall that in the well-known SVRG algorithm, Johnson and Zhang [19] reused a fixed snapshot full gradient $\nabla f(\widetilde{x})$ (which is computed at the beginning of each epoch) in the gradient estimator:

$$v_t \leftarrow \frac{1}{b} \sum_{i \in I_b} \left( \nabla f_i(x_t) - \nabla f_i(\widetilde{x}) \right) + \nabla f(\widetilde{x}), \tag{9}$$

while the stochastic recursive gradient descent uses a recursive update form (more timely update):

$$v_t \leftarrow \frac{1}{b} \sum_{i \in I_b} \left( \nabla f_i(x_t) - \nabla f_i(x_{t-1}) \right) + v_{t-1}. \tag{10}$$

## 4 Convergence Results

Similar to the perturbed GD [18] and perturbed SVRG [15], we add simple perturbations to the stochastic recursive gradient descent algorithm to escape saddle points efficiently. Besides, we also consider the more general online case. In the following theorems, we provide the convergence results of SSRGD for finding an $\epsilon$-first-order stationary point and an $(\epsilon, \delta)$-second-order stationary point for both nonconvex finite-sum problem (1) and online problem (2). The detailed proofs are provided in Appendix C. We give an overview of the proofs in next Section 5.

### 4.1 Nonconvex Finite-sum Problem

**Theorem 1** *Under Assumption 1 (i.e. (3)), let $\Delta f := f(x_0) - f^*$, where $x_0$ is the initial point and $f^*$ is the optimal value of $f$. By letting step size $\eta \leq \frac{\sqrt{5}-1}{2L}$, epoch length $m = \sqrt{n}$ and minibatch*

*size $b = \sqrt{n}$, SSRGD will find an $\epsilon$-first-order stationary point in expectation using*

$$O\Big(n + \frac{L\Delta f\sqrt{n}}{\epsilon^2}\Big)$$

*stochastic gradients for nonconvex finite-sum problem* (1).

**Theorem 2** *Under Assumption 1 and 2 (i.e. (3) and (5)), let $\Delta f := f(x_0) - f^*$, where $x_0$ is the initial point and $f^*$ is the optimal value of $f$. By letting step size $\eta = \widetilde{O}(\frac{1}{L})$, epoch length $m = \sqrt{n}$, minibatch size $b = \sqrt{n}$, perturbation radius $r = \widetilde{O}\big(\min(\frac{\delta^3}{\rho^2\epsilon}, \frac{\delta^{3/2}}{\rho\sqrt{L}})\big)$, threshold gradient $g_{\mathrm{thres}} = \epsilon$, threshold function value $f_{\mathrm{thres}} = \widetilde{O}(\frac{\delta^3}{\rho^2})$ and super epoch length $t_{\mathrm{thres}} = \widetilde{O}(\frac{1}{\eta\delta})$, SSRGD will at least once get to an $(\epsilon, \delta)$-second-order stationary point with high probability using*

$$\widetilde{O}\Big(\frac{L\Delta f\sqrt{n}}{\epsilon^2} + \frac{L\rho^2\Delta f\sqrt{n}}{\delta^4} + \frac{\rho^2\Delta f n}{\delta^3}\Big)$$

*stochastic gradients for nonconvex finite-sum problem* (1).

### 4.2 Nonconvex Online (Expectation) Problem

For nonconvex online problem (2), one usually needs the following bounded variance assumption. For notational convenience, we also consider this online case as the finite-sum form by letting $\nabla f_i(x) := \nabla F(x, \zeta_i)$ and thinking of $n$ as infinity (infinite data samples). Although we try to write it as finite-sum form, the convergence analysis of optimization methods in this online case is a little different from the finite-sum case.

**Assumption 3 (Bounded Variance)** *For $\forall x \in \mathbb{R}^d$, $\mathbb{E}_i[\|\nabla f_i(x) - \nabla f(x)\|^2] := \mathbb{E}_{\zeta_i}[\|\nabla F(x, \zeta_i) - \nabla f(x)\|^2] \leq \sigma^2$, where $\sigma > 0$ is a constant.*

Note that this assumption is standard and necessary for this online case since the full gradients are not available (see e.g., [16, 22, 23, 21, 20, 35, 11, 32, 27]). Moreover, we need to modify the full gradient computation step at the beginning of each epoch to a large batch stochastic gradient computation step (similar to [22, 23]), i.e., change $v_{sm} \leftarrow \nabla f(x_{sm})$ (Line 8 of Algorithm 2) to

$$v_{sm} \leftarrow \frac{1}{B}\sum_{j \in I_B}\nabla f_j(x_{sm}), \tag{11}$$

where $I_B$ are i.i.d. samples with $|I_B| = B$. We call $B$ the batch size and $b$ the minibatch size. Also, we need to change $\|\nabla f(x_{sm})\| \leq g_{\mathrm{thres}}$ (Line 3 of Algorithm 2) to $\|v_{sm}\| \leq g_{\mathrm{thres}}$.

**Theorem 3** *Under Assumption 1 (i.e. (4)) and Assumption 3, let $\Delta f := f(x_0) - f^*$, where $x_0$ is the initial point and $f^*$ is the optimal value of $f$. By letting step size $\eta \leq \frac{\sqrt{5}-1}{2L}$, batch size $B = \frac{4\sigma^2}{\epsilon^2}$, minibatch size $b = \sqrt{B} = \frac{\sigma}{\epsilon}$ and epoch length $m = b$, SSRGD will find an $\epsilon$-first-order stationary point in expectation using*

$$O\Big(\frac{\sigma^2}{\epsilon^2} + \frac{L\Delta f\sigma}{\epsilon^3}\Big)$$

*stochastic gradients for nonconvex online problem* (2).

For achieving a high probability result of finding second-order stationary points in this online case (i.e., Theorem 4), we need a stronger version of Assumption 3 as in the following Assumption 4.

**Assumption 4 (Bounded Variance)** *For $\forall i, x$, $\|\nabla f_i(x) - \nabla f(x)\|^2 := \|\nabla F(x, \zeta_i) - \nabla f(x)\|^2 \leq \sigma^2$, where $\sigma > 0$ is a constant.*

We want to point out that Assumption 4 can be relaxed such that $\|\nabla f_i(x) - \nabla f(x)\|$ has sub-Gaussian tail, i.e., $\mathbb{E}[\exp(\lambda\|\nabla f_i(x) - \nabla f(x)\|)] \leq \exp(\lambda^2\sigma^2/2)$, for $\forall \lambda \in \mathbb{R}$. Then it is sufficient for us to get a high probability bound by using Hoeffding bound on these sub-Gaussian variables. Note that Assumption 4 (or the relaxed sub-Gaussian version) is also standard in online case for second-order stationary point finding algorithms (see e.g., [4, 34, 11]).

**Theorem 4** *Under Assumption 1, 2 (i.e. (4) and (6)) and Assumption 4, let $\Delta f := f(x_0) - f^*$, where $x_0$ is the initial point and $f^*$ is the optimal value of $f$. By letting step size $\eta = \widetilde{O}(\frac{1}{L})$, batch size $B = \widetilde{O}(\frac{\sigma^2}{g_{\text{thres}}^2}) = \widetilde{O}(\frac{\sigma^2}{\epsilon^2})$, minibatch size $b = \sqrt{B} = \widetilde{O}(\frac{\sigma}{\epsilon})$, epoch length $m = b$, perturbation radius $r = \widetilde{O}\big(\min(\frac{\delta^3}{\rho^2 \epsilon}, \frac{\delta^{3/2}}{\rho\sqrt{L}})\big)$, threshold gradient $g_{\text{thres}} = \epsilon \leq \delta^2/\rho$, threshold function value $f_{\text{thres}} = \widetilde{O}(\frac{\delta^3}{\rho^2})$ and super epoch length $t_{\text{thres}} = \widetilde{O}(\frac{1}{\eta\delta})$, SSRGD will at least once get to an $(\epsilon, \delta)$-second-order stationary point with high probability using*

$$\widetilde{O}\Big(\frac{L\Delta f\sigma}{\epsilon^3} + \frac{\rho^2\Delta f\sigma^2}{\epsilon^2\delta^3} + \frac{L\rho^2\Delta f\sigma}{\epsilon\delta^4}\Big)$$

*stochastic gradients for nonconvex online problem (2).*

## 5 Overview of the Proofs

### 5.1 Finding First-order Stationary Points

In this section, we first show that why SSRGD algorithm can improve previous SVRG type algorithm (see e.g., [23, 15]) from $n^{2/3}/\epsilon^2$ to $n^{1/2}/\epsilon^2$. Then we give a simple high-level proof for achieving the $n^{1/2}/\epsilon^2$ convergence result (i.e., Theorem 1).

**Why it can be improved from $n^{2/3}/\epsilon^2$ to $n^{1/2}/\epsilon^2$:** First, we need a key relation between $f(x_t)$ and $f(x_{t-1})$, where $x_t := x_{t-1} - \eta v_{t-1}$,

$$f(x_t) \leq f(x_{t-1}) - \frac{\eta}{2}\|\nabla f(x_{t-1})\|^2 - \big(\frac{1}{2\eta} - \frac{L}{2}\big)\|x_t - x_{t-1}\|^2 + \frac{\eta}{2}\|\nabla f(x_{t-1}) - v_{t-1}\|^2, \quad (12)$$

where (12) holds since $f$ has $L$-Lipschitz continuous gradient (Assumption 1). The details for obtaining (12) can be found in Appendix C.1 (see (27)).

Note that (12) is very meaningful and also very important for the proofs. The first term $-\frac{\eta}{2}\|\nabla f(x_{t-1})\|^2$ indicates that the function value will decrease a lot if the gradient $\nabla f(x_{t-1})$ is large. The second term $-\big(\frac{1}{2\eta} - \frac{L}{2}\big)\|x_t - x_{t-1}\|^2$ indicates that the function value will also decrease a lot if the moving distance $x_t - x_{t-1}$ is large (note that here we require the step size $\eta \leq \frac{1}{L}$). The additional third term $+\frac{\eta}{2}\|\nabla f(x_{t-1}) - v_{t-1}\|^2$ exists since we use $v_{t-1}$ as an estimator of the actual gradient $\nabla f(x_{t-1})$ (i.e., $x_t := x_{t-1} - \eta v_{t-1}$). So it may increase the function value if $v_{t-1}$ is a bad direction in this step.

To get an $\epsilon$-first-order stationary point, we want to cancel the last two terms in (12). Firstly, we want to bound the last variance term. Recall the variance bound (see Equation (29) in [23]) for SVRG algorithm, i.e., estimator (9):

$$\mathbb{E}\big[\|\nabla f(x_{t-1}) - v_{t-1}\|^2\big] \leq \frac{L^2}{b}\mathbb{E}[\|x_{t-1} - \widetilde{x}\|^2]. \quad (13)$$

In order to connect the last two terms in (12), we use Young's inequality for the second term $\|x_t - x_{t-1}\|^2$, i.e., $-\|x_t - x_{t-1}\|^2 \leq \frac{1}{\alpha}\|x_{t-1} - \widetilde{x}\|^2 - \frac{1}{1+\alpha}\|x_t - \widetilde{x}\|^2$ (for any $\alpha > 0$). By plugging this Young's inequality and (13) into (12), we can cancel the last two terms in (12) by summing up (12) for each epoch, i.e., for each epoch $s$ (i.e., iterations $sm + 1 \leq t \leq sm + m$), we have (see Equation (35) in [23])

$$\mathbb{E}[f(x_{(s+1)m})] \leq \mathbb{E}[f(x_{sm})] - \frac{\eta}{2}\sum_{j=sm+1}^{sm+m}\mathbb{E}[\|\nabla f(x_{j-1})\|^2]. \quad (14)$$

However, due to the Young's inequality, we need to let $b \geq m^2$ to cancel the last two terms in (12) for obtaining (14), where $b$ denotes minibatch size and $m$ denotes the epoch length. According to (14), it is not hard to see that $\hat{x}$ is an $\epsilon$-first-order stationary point in expectation (i.e., $\mathbb{E}[\|\nabla f(\hat{x})\|] \leq \epsilon$) if $\hat{x}$ is chosen uniformly randomly from $\{x_{t-1}\}_{t\in[T]}$ and the number of iterations $T = Sm = \frac{2(f(x_0)-f^*)}{\eta\epsilon^2}$. Note that for each iteration we need to compute $b + \frac{n}{m}$ stochastic gradients, where we amortize the full gradient computation of the beginning point of each epoch ($n$ stochastic gradients) into each iteration

in its epoch (i.e., $n/m$) for simple presentation. Thus, the convergence result is $T(b + \frac{n}{m}) \geq \frac{n^{2/3}}{\epsilon^2}$ since $b \geq m^2$, where equality holds if $b = m^2 = n^{2/3}$. Note that here we ignore the factors of $f(x_0) - f^*$ and $\eta = O(1/L)$.

However, for stochastic recursive gradient descent estimator (10), we can bound the last variance term in (12) as (see Equation (33) in Appendix C.1):

$$\mathbb{E}\big[\|\nabla f(x_{t-1}) - v_{t-1}\|^2\big] \leq \frac{L^2}{b} \sum_{j=sm+1}^{t-1} \mathbb{E}[\|x_j - x_{j-1}\|^2]. \qquad (15)$$

Now, the advantage of (15) compared with (13) is that it is already connected to the second term in (12), i.e., moving distances $\{\|x_t - x_{t-1}\|^2\}_t$. Thus we do not need an additional Young's inequality to transform the second term as before. This makes the function value decrease bound tighter. Similarly, we plug (15) into (12) and sum it up for each epoch to cancel the last two terms in (12), i.e., for each epoch $s$, we have (see Equation (35) in Appendix C.1)

$$\mathbb{E}[f(x_{(s+1)m})] \leq \mathbb{E}[f(x_{sm})] - \frac{\eta}{2} \sum_{j=sm+1}^{sm+m} \mathbb{E}[\|\nabla f(x_{j-1})\|^2]. \qquad (16)$$

Compared with (14) (which requires $b \geq m^2$), here (16) only requires $b \geq m$ due to the tighter function value decrease bound since it does not involve the additional Young's inequality.

**High-level proof for achieving $n^{1/2}/\epsilon^2$ result:** Now, according to (16), we can use the same above SVRG arguments to show the $n^{1/2}/\epsilon^2$ convergence result of SSRGD, i.e., $\hat{x}$ is an $\epsilon$-first-order stationary point in expectation (i.e., $\mathbb{E}[\|\nabla f(\hat{x})\|] \leq \epsilon$) if $\hat{x}$ is chosen uniformly randomly from $\{x_{t-1}\}_{t \in [T]}$ and the number of iterations $T = Sm = \frac{2(f(x_0) - f^*)}{\eta \epsilon^2}$. Also, for each iteration, we compute $b + \frac{n}{m}$ stochastic gradients. The only difference is that now the convergence result is $T(b + \frac{n}{m}) = O(\frac{L\Delta f \sqrt{n}}{\epsilon^2})$ since $b \geq m$ (rather than $b \geq m^2$), where we let $b = m = n^{1/2}$, $\eta = O(1/L)$ and $\Delta f := f(x_0) - f^*$. Moreover, it is optimal since it matches the lower bound $\Omega(\frac{L\Delta f \sqrt{n}}{\epsilon^2})$ provided by [11].

## 5.2 Finding Second-order Stationary Points

In this section, we only discuss some high-level proof ideas for finding a second-order stationary point with high probability due to the space limit. We provide a more detailed proof sketch in Appendix A. We have discussed the difference of the first-order guarantee analysis between estimator (9) and estimator (10) in previous Section 5.1. For the second-order analysis, since the estimator (10) in our SSRGD is more correlated than (9), thus we will use martingales to handle it. Besides, different estimators will incur more differences in the detailed proofs of second-order guarantee analysis than that of first-order guarantee analysis.

We divide the proof into two situations, i.e., *large gradients* and *around saddle points*. According to (16), a natural way to prove the convergence result is that the function value will decrease at a *desired rate* with high probability. Note that the total amount for function value decrease is at most $\Delta f := f(x_0) - f^*$.

**Large gradients:** $\|\nabla f(x)\| \geq g_{\text{thres}}$.
In this situation, due to the large gradients, it is sufficient to adjust the first-order analysis to show that the function value will decrease a lot in an epoch with high probability. Concretely, we want to show that the function value decrease bound (16) holds with high probability by using Azuma-Hoeffding inequality to bound the variance term (15) with high probability. Then, according to (16), it is not hard to see that the desired rate of function value decrease is $O(\eta g_{\text{thres}}^2) = \widetilde{O}(\frac{\epsilon^2}{L})$ per iteration in this situation (recall the parameters $g_{\text{thres}} = \epsilon$ and $\eta = \widetilde{O}(1/L)$ in our Theorem 2). Also note that we compute $b + \frac{n}{m} = 2\sqrt{n}$ stochastic gradients at each iteration (recall $m = b = \sqrt{n}$ in our Theorem 2). Here we amortize the full gradient computation of the beginning point of each epoch ($n$ stochastic gradients) into each iteration in its epoch (i.e., $n/m$) for simple presentation (we will analyze this more rigorously in the detailed proofs in appendices). Thus the number of stochastic gradient computation is at most $\widetilde{O}(\sqrt{n} \frac{\Delta f}{\epsilon^2/L}) = \widetilde{O}(\frac{L\Delta f \sqrt{n}}{\epsilon^2})$ for this large gradients situation.

Note that (16) only guarantees function value decrease when the summation of gradients in this epoch is large. However, in order to connect the guarantees between first situation (large gradients) and second situation (around saddle points), we need to show guarantees that are related to the *gradient of the starting point* of each epoch (see Line 3 of Algorithm 2). Similar to [15], we achieve this by stopping the epoch at a uniformly random point (see Line 16 of Algorithm 2). Then, we will know that either the function value already decreases a lot in this epoch $s$ or the starting point of the next epoch $x_{(s+1)m}$ is around a saddle point (or $x_{(s+1)m}$ is already a second-order stationary point).

**Around saddle points:** $\|\nabla f(\widetilde{x})\| \leq g_{\mathrm{thres}}$ and $\lambda_{\min}(\nabla^2 f(\widetilde{x})) \leq -\delta$ at the initial point $\widetilde{x}$ of a super epoch.

In this situation, we want to show that the function value will decrease a lot in a *super epoch* (instead of an epoch as in the first situation) with high probability by adding a random perturbation at the initial point $\widetilde{x}$. To simplify the presentation, we use $x_0 := \widetilde{x} + \xi$ to denote the starting point of the super epoch after the perturbation, where $\xi$ uniformly $\sim \mathbb{B}_0(r)$ (see Line 6 in Algorithm 2).

Firstly, we show that if function value does not decrease a lot, then all iteration points are not far from the starting point with high probability (*localization*). Concretely, we have

$$\forall t, \ \|x_t - x_0\| \leq \sqrt{\tfrac{4t(f(x_0)-f(x_t))}{C'L}}, \tag{17}$$

where $C' = \widetilde{O}(1)$. Then we show that the stuck region is relatively small in the random perturbation ball, i.e., $x_t$ will go far away from the *perturbed* starting point $x_0$ with high probability (*small stuck region*). Concretely, we have

$$\exists t \leq t_{\mathrm{thres}}, \ \|x_t - x_0\| \geq \tfrac{\delta}{C_1\rho}, \tag{18}$$

where $C_1 = \widetilde{O}(1)$. Based on (17) and (18), we can prove that

$$\exists t \leq t_{\mathrm{thres}}, \ f(\widetilde{x}) - f(x_t) \geq f_{\mathrm{thres}}$$

holds with high probability.

Now, we can obtain that the desired rate of function value decrease in this situation is $f_{\mathrm{thres}}/t_{\mathrm{thres}} = \widetilde{O}(\frac{\delta^3/\rho^2}{1/(\eta\delta)}) = \widetilde{O}(\frac{\delta^4}{L\rho^2})$ per iteration (recall the parameters $f_{\mathrm{thres}} = \widetilde{O}(\delta^3/\rho^2)$, $t_{\mathrm{thres}} = \widetilde{O}(1/(\eta\delta))$ and $\eta = \widetilde{O}(1/L)$ in our Theorem 2). Same as before, we compute $b + \frac{n}{m} = 2\sqrt{n}$ stochastic gradients at each iteration (recall $m = b = \sqrt{n}$ in our Theorem 2). Thus the number of stochastic gradient computation is at most $\widetilde{O}(\sqrt{n}\frac{\Delta f}{\delta^4/(L\rho^2)}) = \widetilde{O}(\frac{L\rho^2\Delta f\sqrt{n}}{\delta^4})$ for this around saddle points situation.

In sum, the number of stochastic gradient computation is at most $\widetilde{O}(\frac{L\Delta f\sqrt{n}}{\epsilon^2})$ for the large gradients situation and is at most $\widetilde{O}(\frac{L\rho^2\Delta f\sqrt{n}}{\delta^4})$ for the around saddle points situation. Moreover, for the classical version where $\delta = \sqrt{\rho\epsilon}$ [25, 18], then $\widetilde{O}(\frac{L\rho^2\Delta f\sqrt{n}}{\delta^4}) = \widetilde{O}(\frac{L\Delta f\sqrt{n}}{\epsilon^2})$, i.e., both situations get the same stochastic gradient complexity. This also matches the convergence result for finding first-order stationary points (see our Theorem 1) if we ignore the logarithmic factor. More importantly, it also almost matches the lower bound $\Omega(\frac{L\Delta f\sqrt{n}}{\epsilon^2})$ provided by [11] for finding even just an $\epsilon$-first-order stationary point.

Finally, we point out that there is an extra term $\frac{\rho^2\Delta fn}{\delta^3}$ in Theorem 2 beyond these two terms obtained from the above two situations. The reason is that we amortize the full gradient computation of the beginning point of each epoch ($n$ stochastic gradients) into each iteration in its epoch (i.e., $n/m$) for simple presentation. We will analyze this more rigorously in the appendices, which incurs the term $\frac{\rho^2\Delta fn}{\delta^3}$. For the more general online problem (2), the high-level proofs are almost the same as the finite-sum problem (1). The difference is that we need to use more concentration bounds in the detailed proofs since the full gradients are not available in online case.

**Acknowledgments**

This work was supported by Office of Sponsored Research of KAUST, through the Baseline Research Fund of Prof. Peter Richtárik. The author would like to thank Rong Ge (Duke), Jian Li (Tsinghua) and the anonymous reviewers for their useful discussions/suggestions.

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
