[Supplementary Material]

# A   Proof Sketch for Finding Second-order Stationary Points

In previous Section 5.2, we only discuss some high-level proof ideas for finding a second-order stationary point with high probability due to the space limit. In this appendix, we give a more detailed proof sketch for finding a second-order stationary point with high probability (Theorem 2). The complete proof of Theorem 2 is deferred to Appendix C.1.1.

We divide the proof into two situations, i.e., large gradients and around saddle points. According to (16), a natural way to prove the convergence result is that the function value will decrease at a *desired rate* with high probability. Note that the amount for function value decrease is at most $\Delta f := f(x_0) - f^*$.

**Large gradients:** $\|\nabla f(x)\| \geq g_{\text{thres}}$
In this situation, due to the large gradients, it is sufficient to adjust the first-order analysis to show that the function value will decrease a lot in an epoch. Concretely, we want to show the function value decrease bound (16) holds with high probability. It is not hard to see that the desired rate of function value decrease is $O(\eta g_{\text{thres}}^2) = \widetilde{O}(\frac{\epsilon^2}{L})$ per iteration (recall the parameters $g_{\text{thres}} = \epsilon$ and $\eta = \widetilde{O}(1/L)$ in our Theorem 2). Also note that we compute $b + \frac{n}{m} = 2\sqrt{n}$ stochastic gradients at each iteration (recall $m = b = \sqrt{n}$ in our Theorem 2). Here we amortize the full gradient computation of the beginning point of each epoch ($n$ stochastic gradients) into each iteration in its epoch (i.e., $n/m$) for simple presentation (we will analyze this more rigorous in the complete proof in Appendix C.1.1). Thus the number of stochastic gradient computation is at most $\widetilde{O}(\sqrt{n}\frac{\Delta f}{\epsilon^2/L}) = \widetilde{O}(\frac{L\Delta f\sqrt{n}}{\epsilon^2})$ for this large gradients situation.

For the proof, to show the function value decrease bound (16) holds with high probability, we need to show that the bound for variance term $(\|v_k - \nabla f(x_k)\|^2)$ holds with high probability. Note that the estimator $v_k$ defined in (10) is correlated with previous $v_{k-1}$. Fortunately, let $y_k := v_k - \nabla f(x_k)$, then it is not hard to see that $\{y_k\}$ is a martingale vector sequence with respect to a filtration $\{\mathscr{F}_k\}$ such that $\mathbb{E}[y_k|\mathscr{F}_{k-1}] = y_{k-1}$. Moreover, let $\{z_k\}$ denote the associated martingale difference sequence with respect to the filtration $\{\mathscr{F}_k\}$, i.e., $z_k := y_k - \mathbb{E}[y_k|\mathscr{F}_{k-1}] = y_k - y_{k-1}$ and $\mathbb{E}[z_k|\mathscr{F}_{k-1}] = 0$. Thus to bound the variance term $\|v_k - \nabla f(x_k)\|^2$ with high probability, it is sufficient to bound the martingale sequence $\{y_k\}$. This can be bounded with high probability by using the martingale Azuma-Hoeffding inequality. Note that in order to apply Azuma-Hoeffding inequality, we first need to use the Bernstein inequality to bound the associated difference sequence $\{z_k\}$. In sum, we will get the high probability function value decrease bound by applying these two inequalities (see (44) in Appendix C.1).

Note that (44) only guarantees function value decrease when the summation of gradients in this epoch is large. However, in order to connect the guarantees between first situation (large gradients) and second situation (around saddle points), we need to show guarantees that are related to the *gradient of the starting point* of each epoch (see Line 3 of Algorithm 2). Similar to [15], we achieve this by stopping the epoch at a uniformly random point (see Line 16 of Algorithm 2). We use the following lemma to connect these two situations (large gradients and around saddle points):

**Lemma 1 (Connection of Two Situations)** *For any epoch $s$, let $x_t$ be a point uniformly sampled from this epoch $\{x_j\}_{j=sm}^{(s+1)m}$. Moreover, let the step size $\eta \leq \frac{\sqrt{4C'^2+1}-1}{2C'^2 L}$ (where $C' = O(\log\frac{dn}{\zeta}) = \widetilde{O}(1)$) and the minibatch size $b \geq m$, there are two cases:*

1. *If at least half of points in this epoch have gradient norm no larger than $g_{\text{thres}}$, then $\|\nabla f(x_t)\| \leq g_{\text{thres}}$ holds with probability at least $1/2$;*

2. *Otherwise, we know $f(x_{sm}) - f(x_t) \geq \frac{\eta m g_{\text{thres}}^2}{8}$ holds with probability at least $1/5$.*

*Moreover, $f(x_t) \leq f(x_{sm})$ holds with high probability no matter which case happens.*

Note that if Case 2 happens, the function value already decreases a lot in this epoch $s$ (as we already discussed at the beginning of this situation). Otherwise, Case 1 happens, we know the starting point of the next epoch $x_{(s+1)m} = x_t$ (i.e., Line 19 of Algorithm 2), then we know $\|\nabla f(x_{(s+1)m})\| = \|\nabla f(x_t)\| \leq g_{\text{thres}}$. Then we will start a super epoch (see Line 3 of Algorithm 2). This corresponds

to the following second situation around saddle points. Note that if $\lambda_{\min}(\nabla^2 f(x_{(s+1)m})) > -\delta$, this point $x_{(s+1)m}$ is already an $(\epsilon, \delta)$-second-order stationary point (recall $g_{\mathrm{thres}} = \epsilon$ in our Theorem 2).

**Around saddle points:** $\|\nabla f(\widetilde{x})\| \leq g_{\mathrm{thres}}$ and $\lambda_{\min}(\nabla^2 f(\widetilde{x})) \leq -\delta$ at the initial point $\widetilde{x}$ of a super epoch

In this situation, we want to show that the function value will decrease a lot in a *super epoch* (instead of an epoch as in the first situation) with high probability by adding a random perturbation at the initial point $\widetilde{x}$. To simplify the presentation, we use $x_0 := \widetilde{x} + \xi$ to denote the starting point of the super epoch after the perturbation, where $\xi$ uniformly $\sim \mathbb{B}_0(r)$ and the perturbation radius is $r$ (see Line 6 in Algorithm 2). Following the classical widely used *two-point analysis* developed in [18], we consider two coupled points $x_0$ and $x_0'$ with $w_0 := x_0 - x_0' = r_0 e_1$, where $r_0$ is a scalar and $e_1$ denotes the smallest eigenvector direction of Hessian $\nabla^2 f(\widetilde{x})$. Then we get two coupled sequences $\{x_t\}$ and $\{x_t'\}$ by running SSRGD update steps (Line 8–12 of Algorithm 2) with the same choice of minibatches (i.e., $I_b$'s in Line 12 of Algorithm 2) for a super epoch. We will show that at least one of these two coupled sequences will decrease the function value a lot (escape the saddle point) with high probability, i.e.,

$$\exists t \leq t_{\mathrm{thres}}, \text{ such that } \max\{f(x_0) - f(x_t), f(x_0') - f(x_t')\} \geq 2f_{\mathrm{thres}}. \tag{19}$$

Similar to the classical argument in [18], according to (19), we know that in the random perturbation ball, the stuck points can only be a short interval in the $e_1$ direction, i.e., at least one of two points in the $e_1$ direction will escape the saddle point if their distance is larger than $r_0 = \frac{\zeta' r}{\sqrt{d}}$. Thus, we know that the probability of the starting point $x_0 = \widetilde{x} + \xi$ (where $\xi$ uniformly $\sim \mathbb{B}_0(r)$) located in the stuck region is less than $\zeta'$ (see (50) in Appendix C.1). By a union bound ($x_0$ is not in a stuck region and (19) holds), with high probability, we have

$$\exists t \leq t_{\mathrm{thres}}, f(x_0) - f(x_t) \geq 2f_{\mathrm{thres}}. \tag{20}$$

Note that the initial point of this super epoch is $\widetilde{x}$ before the perturbation (see Line 6 of Algorithm 2), thus we also need to show that the perturbation step $x_0 = \widetilde{x} + \xi$ (where $\xi$ uniformly $\sim \mathbb{B}_0(r)$) does not increase the function value a lot, i.e.,

$$
\begin{aligned}
f(x_0) &\leq f(\widetilde{x}) + \langle \nabla f(\widetilde{x}), x_0 - \widetilde{x} \rangle + \frac{L}{2} \|x_0 - \widetilde{x}\|^2 \\
&\leq f(\widetilde{x}) + g_{\mathrm{thres}} \cdot r + \frac{L}{2} r^2 \\
&= f(\widetilde{x}) + f_{\mathrm{thres}},
\end{aligned}
\tag{21}
$$

where the last inequality holds since the initial point $\widetilde{x}$ satisfying $\|\nabla f(\widetilde{x})\| \leq g_{\mathrm{thres}}$ and the perturbation radius is $r$, and the last equality holds by letting the perturbation radius $r$ small enough. By combining (20) and (21), we obtain with high probability

$$f(\widetilde{x}) - f(x_t) = f(\widetilde{x}) - f(x_0) + f(x_0) - f(x_t) \geq -f_{\mathrm{thres}} + 2f_{\mathrm{thres}} = f_{\mathrm{thres}}. \tag{22}$$

Now, we can obtain the desired rate of function value decrease in this situation is $\frac{f_{\mathrm{thres}}}{t_{\mathrm{thres}}} = \widetilde{O}(\frac{\delta^3/\rho^2}{1/(\eta\delta)}) = \widetilde{O}(\frac{\delta^4}{L\rho^2})$ per iteration (recall the parameters $f_{\mathrm{thres}} = \widetilde{O}(\delta^3/\rho^2)$, $t_{\mathrm{thres}} = \widetilde{O}(1/(\eta\delta))$ and $\eta = \widetilde{O}(1/L)$ in our Theorem 2). Same as before, we compute $b + \frac{n}{m} = 2\sqrt{n}$ stochastic gradients at each iteration (recall $m = b = \sqrt{n}$ in our Theorem 2). Thus the number of stochastic gradient computation is at most $\widetilde{O}(\sqrt{n} \frac{\Delta f}{\delta^4/(L\rho^2)}) = \widetilde{O}(\frac{L\rho^2 \Delta f \sqrt{n}}{\delta^4})$ for this around saddle points situation.

Now, the remaining thing is to prove (19). It can be proved by contradiction. Assume the contrary, $f(x_0) - f(x_t) < 2f_{\mathrm{thres}}$ and $f(x_0') - f(x_t') < 2f_{\mathrm{thres}}$. First, we show that if function value does not decrease a lot, then all iteration points are not far from the starting point with high probability.

**Lemma 2 (Localization)** *Let $\{x_t\}$ denote the sequence by running SSRGD update steps (Line 8–12 of Algorithm 2) from $x_0$. Moreover, let the step size $\eta \leq \frac{1}{2C'L}$ and minibatch size $b \geq m$, with probability $1 - \zeta$, we have*

$$\forall t, \ \|x_t - x_0\| \leq \sqrt{\frac{4t(f(x_0) - f(x_t))}{C'L}}, \tag{23}$$

*where $C' = O(\log \frac{dt}{\zeta}) = \widetilde{O}(1)$.*

Then we will show that the stuck region is relatively small in the random perturbation ball, i.e., at least one of $x_t$ and $x_t'$ will go far away from their starting point $x_0$ and $x_0'$ with high probability.

**Lemma 3 (Small Stuck Region)** *If the initial point $\widetilde{x}$ satisfies $-\gamma := \lambda_{\min}(\nabla^2 f(\widetilde{x})) \leq -\delta$, then let $\{x_t\}$ and $\{x_t'\}$ be two coupled sequences by running SSRGD update steps (Line 8–12 of Algorithm 2) with the same choice of minibatches (i.e., $I_b$'s in Line 12) from $x_0$ and $x_0'$ with $w_0 := x_0 - x_0' = r_0 e_1$, where $x_0 \in \mathbb{B}_{\widetilde{x}}(r)$, $x_0' \in \mathbb{B}_{\widetilde{x}}(r)$ , $r_0 = \frac{\zeta' r}{\sqrt{d}}$ and $e_1$ denotes the smallest eigenvector direction of Hessian $\nabla^2 f(\widetilde{x})$. Moreover, let the super epoch length $t_{\mathrm{thres}} = \frac{2\log(\frac{8\delta\sqrt{d}}{C_1 \rho \zeta' r})}{\eta\delta} = \widetilde{O}(\frac{1}{\eta\delta})$, the step size $\eta \leq \min\left(\frac{1}{8\log(\frac{8\delta\sqrt{d}}{C_1 \rho \zeta' r})L}, \frac{1}{4C_2 L \log t_{\mathrm{thres}}}\right) = \widetilde{O}(\frac{1}{L})$, minibatch size $b \geq m$ and the perturbation radius $r \leq \frac{\delta}{C_1 \rho}$, then with probability $1 - \zeta$, we have*

$$\exists T \leq t_{\mathrm{thres}}, \quad \max\{\|x_T - x_0\|, \|x_T' - x_0'\|\} \geq \frac{\delta}{C_1 \rho}, \tag{24}$$

*where $C_1 \geq \frac{20 C_2}{\eta L}$ and $C_2 = O(\log \frac{d t_{\mathrm{thres}}}{\zeta}) = \widetilde{O}(1)$.*

Based on these two lemmas, we are ready to show that (19) holds with high probability. Without loss of generality, we assume $\|x_T - x_0\| \geq \frac{\delta}{C_1 \rho}$ in (24) (note that (23) holds for both $\{x_t\}$ and $\{x_t'\}$), then by plugging it into (23) to obtain

$$\sqrt{\frac{4T(f(x_0) - f(x_T))}{C' L}} \geq \frac{\delta}{C_1 \rho}$$

$$f(x_0) - f(x_T) \geq \frac{C' L \delta^2}{4 C_1^2 \rho^2 T}$$

$$\geq \frac{\eta C' L \delta^3}{8 C_1^2 \rho^2 \log(\frac{8\delta\sqrt{d}}{C_1 \rho \zeta' r})}$$

$$= \frac{\delta^3}{C_1' \rho^2}$$

$$= 2 f_{\mathrm{thres}},$$

where the last inequality is due to $T \leq t_{\mathrm{thres}}$ and the first equality holds by letting $C_1' = \frac{8 C_1^2 \log(\frac{8\delta\sqrt{d}}{C_1 \rho \zeta' r})}{\eta C' L} = \widetilde{O}(1)$ (recall the parameters $f_{\mathrm{thres}} = \widetilde{O}(\delta^3/\rho^2)$ and $\eta = \widetilde{O}(1/L)$ in our Theorem 2). Now, the high-level proof for this situation is finished.

In sum, the number of stochastic gradient computation is at most $\widetilde{O}(\frac{L\Delta f \sqrt{n}}{\epsilon^2})$ for the large gradients situation and is at most $\widetilde{O}(\frac{L\rho^2 \Delta f \sqrt{n}}{\delta^4})$ for the around saddle points situation. Moreover, for the classical version where $\delta = \sqrt{\rho\epsilon}$ [25, 18], then $\widetilde{O}(\frac{L\rho^2 \Delta f \sqrt{n}}{\delta^4}) = \widetilde{O}(\frac{L\Delta f \sqrt{n}}{\epsilon^2})$, i.e., both situations get the same stochastic gradient complexity. It also matches the convergence result for finding first-order stationary points (see our Theorem 1) if we ignore the logarithmic factor.

Finally, we point out that there is an extra term $\frac{\rho^2 \Delta f n}{\delta^3}$ in Theorem 2 beyond these two terms obtained from the above two situations. The reason is that we amortize the full gradient computation of the beginning point of each epoch ($n$ stochastic gradients) into each iteration in its epoch (i.e., $n/m$) for simple presentation. We will analyze this more rigorous in Appendix C.1.1, which incurs the term $\frac{\rho^2 \Delta f n}{\delta^3}$. For the more general online problem (2), the high-level proofs are almost the same as the finite-sum problem (1). The difference is that we need to use more concentration bounds in the detailed proofs since the full gradients are not available in online case.

# B  Tools

In this appendix, we recall some classical concentration bounds for matrices and vectors.

**Proposition 1 (Bernstein Inequality [31])** *Consider a finite sequence $\{Z_k\}$ of independent, random matrices with dimension $d_1 \times d_2$. Assume that each random matrix satisfies*

$$\mathbb{E}[Z_k] = 0 \ \ and \ \ \|Z_k\| \leq R \ \ almost \ surely.$$

*Define*

$$\sigma^2 := \max \Big\{ \big\| \sum_k \mathbb{E}[Z_k Z_k^*] \big\|, \big\| \sum_k \mathbb{E}[Z_k^* Z_k] \big\| \Big\}.$$

*Then, for all $t \geq 0$,*

$$\mathbb{P}\Big\{ \big\| \sum_k Z_k \big\| \geq t \Big\} \leq (d_1 + d_2) \exp \Big( \frac{-t^2/2}{\sigma^2 + Rt/3} \Big).$$

In our proof, we only need its special case vector version as follows, where $z_k = v_k - \mathbb{E}[v_k]$.

**Proposition 2 (Bernstein Inequality [31])** *Consider a finite sequence $\{v_k\}$ of independent, random vectors with dimension $d$. Assume that each random matrix satisfies*

$$\|v_k - \mathbb{E}[v_k]\| \leq R \ \ almost \ surely.$$

*Define*

$$\sigma^2 := \sum_k \mathbb{E}\|v_k - \mathbb{E}[v_k]\|^2.$$

*Then, for all $t \geq 0$,*

$$\mathbb{P}\Big\{ \big\| \sum_k (v_k - \mathbb{E}[v_k]) \big\| \geq t \Big\} \leq (d+1) \exp \Big( \frac{-t^2/2}{\sigma^2 + Rt/3} \Big).$$

Moreover, we also need the martingale concentration bounds, i.e., Azuma-Hoffding inequality. Now, we will only write the vector version not repeat the more general matrix version.

**Proposition 3 (Azuma-Hoeffding Inequality [17, 30])** *Consider a martingale vector sequence $\{y_k\}$ with dimension $d$, and let $\{z_k\}$ denote the associated martingale difference sequence with respect to a filtration $\{\mathscr{F}_k\}$, i.e., $z_k := y_k - \mathbb{E}[y_k|\mathscr{F}_{k-1}] = y_k - y_{k-1}$ and $\mathbb{E}[z_k|\mathscr{F}_{k-1}] = 0$. Suppose that $\{z_k\}$ satisfies*

$$\|z_k\| = \|y_k - y_{k-1}\| \leq c_k \ \ almost \ surely. \tag{25}$$

*Then, for all $t \geq 0$,*

$$\mathbb{P}\Big\{ \|y_k - y_0\| \geq t \Big\} \leq (d+1) \exp \Big( \frac{-t^2}{8 \sum_{i=1}^k c_i^2} \Big).$$

However, the assumption that $\|z_k\| \leq c_k$ in (25) with probability one sometime fails. Fortunately, the Azuma-Hoffding inequality also holds with a slackness if $\|z_k\| \leq c_k$ with high probability.

**Proposition 4 (Azuma-Hoeffding Inequality with High Probability [7, 29])** *Consider a martingale vector sequence $\{y_k\}$ with dimension $d$, and let $\{z_k\}$ denote the associated martingale difference sequence with respect to a filtration $\{\mathscr{F}_k\}$, i.e., $z_k := y_k - \mathbb{E}[y_k|\mathscr{F}_{k-1}] = y_k - y_{k-1}$ and $\mathbb{E}[z_k|\mathscr{F}_{k-1}] = 0$. Suppose that $\{z_k\}$ satisfies*

$$\|z_k\| = \|y_k - y_{k-1}\| \leq c_k \ \ with \ high \ probability \ 1 - \zeta_k.$$

*Then, for all $t \geq 0$,*

$$\mathbb{P}\Big\{ \|y_k - y_0\| \geq t \Big\} \leq (d+1) \exp \Big( \frac{-t^2}{8 \sum_{i=1}^k c_i^2} \Big) + \sum_{i=1}^k \zeta_k.$$

## C Missing Proofs

In this appendix, we provide the detailed proofs for Theorem 1–4.

### C.1 Proofs for Finite-sum Problem

In this section, we provide the detailed proofs for finite-sum problem (1) (i.e., Theorem 1–2).

First, we obtain the relation between $f(x_t)$ and $f(x_{t-1})$ as follows similar to [23, 15], where we let $x_t := x_{t-1} - \eta v_{t-1}$ and $\bar{x}_t := x_{t-1} - \eta \nabla f(x_{t-1})$,

$$
\begin{aligned}
f(x_t) \leq & f(x_{t-1}) + \langle \nabla f(x_{t-1}), x_t - x_{t-1} \rangle + \frac{L}{2} \|x_t - x_{t-1}\|^2 \qquad (26) \\
= & f(x_{t-1}) + \langle \nabla f(x_{t-1}) - v_{t-1}, x_t - x_{t-1} \rangle + \langle v_{t-1}, x_t - x_{t-1} \rangle + \frac{L}{2} \|x_t - x_{t-1}\|^2 \\
= & f(x_{t-1}) + \langle \nabla f(x_{t-1}) - v_{t-1}, -\eta v_{t-1} \rangle - \left(\frac{1}{\eta} - \frac{L}{2}\right) \|x_t - x_{t-1}\|^2 \\
= & f(x_{t-1}) + \eta \|\nabla f(x_{t-1}) - v_{t-1}\|^2 - \eta \langle \nabla f(x_{t-1}) - v_{t-1}, \nabla f(x_{t-1}) \rangle - \left(\frac{1}{\eta} - \frac{L}{2}\right) \|x_t - x_{t-1}\|^2 \\
= & f(x_{t-1}) + \eta \|\nabla f(x_{t-1}) - v_{t-1}\|^2 - \frac{1}{\eta} \langle x_t - \bar{x}_t, x_{t-1} - \bar{x}_t \rangle - \left(\frac{1}{\eta} - \frac{L}{2}\right) \|x_t - x_{t-1}\|^2 \\
= & f(x_{t-1}) + \eta \|\nabla f(x_{t-1}) - v_{t-1}\|^2 - \left(\frac{1}{\eta} - \frac{L}{2}\right) \|x_t - x_{t-1}\|^2 \\
& \quad - \frac{1}{2\eta} \left( \|x_t - \bar{x}_t\|^2 + \|x_{t-1} - \bar{x}_t\|^2 - \|x_t - x_{t-1}\|^2 \right) \\
= & f(x_{t-1}) + \frac{\eta}{2} \|\nabla f(x_{t-1}) - v_{t-1}\|^2 - \frac{\eta}{2} \|\nabla f(x_{t-1})\|^2 - \left(\frac{1}{2\eta} - \frac{L}{2}\right) \|x_t - x_{t-1}\|^2, \qquad (27)
\end{aligned}
$$

where (26) holds since $f$ has $L$-Lipschitz continuous gradient (Assumption 1). Now, we bound the variance term as follows, where we take expectations with the history:

$$
\begin{aligned}
& \mathbb{E}[\|v_{t-1} - \nabla f(x_{t-1})\|^2] \\
= & \mathbb{E}\left[\left\| \frac{1}{b} \sum_{i \in I_b} \left( \nabla f_i(x_{t-1}) - \nabla f_i(x_{t-2}) \right) + v_{t-2} - \nabla f(x_{t-1}) \right\|^2 \right] \\
= & \mathbb{E}\left[\left\| \frac{1}{b} \sum_{i \in I_b} \left( \left(\nabla f_i(x_{t-1}) - \nabla f_i(x_{t-2})\right) - \left(\nabla f(x_{t-1}) - \nabla f(x_{t-2})\right) \right) + v_{t-2} - \nabla f(x_{t-2}) \right\|^2 \right] \\
= & \mathbb{E}\left[\left\| \frac{1}{b} \sum_{i \in I_b} \left( \left(\nabla f_i(x_{t-1}) - \nabla f_i(x_{t-2})\right) - \left(\nabla f(x_{t-1}) - \nabla f(x_{t-2})\right) \right) \right\|^2 \right] + \mathbb{E}[\|v_{t-2} - \nabla f(x_{t-2})\|^2] \\
& \qquad (28) \\
= & \frac{1}{b^2} \mathbb{E}\left[ \sum_{i \in I_b} \left\| \left(\nabla f_i(x_{t-1}) - \nabla f_i(x_{t-2})\right) - \left(\nabla f(x_{t-1}) - \nabla f(x_{t-2})\right) \right\|^2 \right] + \mathbb{E}[\|v_{t-2} - \nabla f(x_{t-2})\|^2] \\
& \qquad (29) \\
\leq & \frac{1}{b^2} \mathbb{E}\left[ \sum_{i \in I_b} \left\| \nabla f_i(x_{t-1}) - \nabla f_i(x_{t-2}) \right\|^2 \right] + \mathbb{E}[\|v_{t-2} - \nabla f(x_{t-2})\|^2] \qquad (30) \\
\leq & \frac{L^2}{b} \mathbb{E}[\|x_{t-1} - x_{t-2}\|^2] + \mathbb{E}[\|v_{t-2} - \nabla f(x_{t-2})\|^2], \qquad (31)
\end{aligned}
$$

where (28) and (29) use the law of total expectation and $\mathbb{E}[\|x_1 + x_2 + \cdots + x_k\|^2] = \sum_{i=1}^{k} \mathbb{E}[\|x_i\|^2]$ if $x_1, x_2, \ldots, x_k$ are independent and of mean zero, (30) uses the fact $\mathbb{E}[\|x - \mathbb{E}x\|^2] \leq \mathbb{E}[\|x\|^2]$, and (31) holds due to the gradient Lipschitz Assumption 1.

Note that for $\mathbb{E}[\|v_{t-2} - \nabla f(x_{t-2})\|^2]$ in (31), we can reuse the same computation above. Thus we can sum up (31) from the beginning of this epoch $sm$ to the point $t-1$,

$$\mathbb{E}[\|v_{t-1} - \nabla f(x_{t-1})\|^2] \leq \frac{L^2}{b} \sum_{j=sm+1}^{t-1} \mathbb{E}[\|x_j - x_{j-1}\|^2] + \mathbb{E}[\|v_{sm} - \nabla f(x_{sm})\|^2] \quad (32)$$

$$\leq \frac{L^2}{b} \sum_{j=sm+1}^{t-1} \mathbb{E}[\|x_j - x_{j-1}\|^2], \quad (33)$$

where (33) holds since we compute the full gradient at the beginning point of this epoch, i.e., $v_{sm} = \nabla f(x_{sm})$ (see Line 5 of Algorithm 1). Now, we take expectations for (27) and then sum it up from the beginning of this epoch $s$, i.e., iterations from $sm$ to $t$, by plugging the variance (33) into them to get:

$$\mathbb{E}[f(x_t)] \leq \mathbb{E}[f(x_{sm})] - \frac{\eta}{2} \sum_{j=sm+1}^{t} \mathbb{E}[\|\nabla f(x_{j-1})\|^2] - \left(\frac{1}{2\eta} - \frac{L}{2}\right) \sum_{j=sm+1}^{t} \mathbb{E}[\|x_j - x_{j-1}\|^2]$$

$$+ \frac{\eta L^2}{2b} \sum_{k=sm+1}^{t-1} \sum_{j=sm+1}^{k} \mathbb{E}[\|x_j - x_{j-1}\|^2]$$

$$\leq \mathbb{E}[f(x_{sm})] - \frac{\eta}{2} \sum_{j=sm+1}^{t} \mathbb{E}[\|\nabla f(x_{j-1})\|^2] - \left(\frac{1}{2\eta} - \frac{L}{2}\right) \sum_{j=sm+1}^{t} \mathbb{E}[\|x_j - x_{j-1}\|^2]$$

$$+ \frac{\eta L^2 (t-1-sm)}{2b} \sum_{j=sm+1}^{t} \mathbb{E}[\|x_j - x_{j-1}\|^2]$$

$$\leq \mathbb{E}[f(x_{sm})] - \frac{\eta}{2} \sum_{j=sm+1}^{t} \mathbb{E}[\|\nabla f(x_{j-1})\|^2] - \left(\frac{1}{2\eta} - \frac{L}{2}\right) \sum_{j=sm+1}^{t} \mathbb{E}[\|x_j - x_{j-1}\|^2]$$

$$+ \frac{\eta L^2}{2} \sum_{j=sm+1}^{t} \mathbb{E}[\|x_j - x_{j-1}\|^2] \quad (34)$$

$$\leq \mathbb{E}[f(x_{sm})] - \frac{\eta}{2} \sum_{j=sm+1}^{t} \mathbb{E}[\|\nabla f(x_{j-1})\|^2], \quad (35)$$

where (34) holds if the minibatch size $b \geq m$ (note that here $t \leq (s+1)m$), and (35) holds if the step size $\eta \leq \frac{\sqrt{5}-1}{2L}$.

**Proof of Theorem 1.** Let $b = m = \sqrt{n}$ and step size $\eta \leq \frac{\sqrt{5}-1}{2L}$, then (35) holds. Now, the proof is directly obtained by summing up (35) for all epochs $0 \leq s \leq S$ as follows:

$$\mathbb{E}[f(x_T)] \leq \mathbb{E}[f(x_0)] - \frac{\eta}{2} \sum_{j=1}^{T} \mathbb{E}[\|\nabla f(x_{j-1})\|^2]$$

$$\mathbb{E}[\|\nabla f(\hat{x})\|] \leq \sqrt{\mathbb{E}[\|\nabla f(\hat{x})\|^2]} \leq \sqrt{\frac{2(f(x_0) - f^*)}{\eta T}} = \epsilon, \quad (36)$$

where (36) holds by choosing $\hat{x}$ uniformly from $\{x_{t-1}\}_{t \in [T]}$ and letting $Sm \leq T = \frac{2(f(x_0)-f^*)}{\eta \epsilon^2} = O(\frac{L(f(x_0)-f^*)}{\epsilon^2})$. Note that the total number of computation of stochastic gradients equals to

$$Sn + Smb \leq \left\lceil \frac{T}{m} \right\rceil n + Tb \leq \left(\frac{T}{\sqrt{n}} + 1\right)n + T\sqrt{n} = n + 2T\sqrt{n} = O\left(n + \frac{L(f(x_0) - f^*)\sqrt{n}}{\epsilon^2}\right).$$

$\square$

### C.1.1 Proof of Theorem 2

For proving the second-order guarantee, we divide the proof into two situations. The first situation (**large gradients**) is almost the same as the above arguments for first-order guarantee, where the function value will decrease a lot since the gradients are large (see (35)). For the second situation (**around saddle points**), we will show that the function value can also decrease a lot by adding a random perturbation. The reason is that saddle points are usually unstable and the stuck region is relatively small in a random perturbation ball.

**Large Gradients**: First, we need a high probability bound for the variance term instead of the expectation one (33). Then we use it to get a high probability bound of (35) for function value decrease. Recall that $v_k = \frac{1}{b} \sum_{i \in I_b} \left( \nabla f_i(x_k) - \nabla f_i(x_{k-1}) \right) + v_{k-1}$ (see Line 9 of Algorithm 1), we let $y_k := v_k - \nabla f(x_k)$ and $z_k := y_k - y_{k-1}$. It is not hard to verify that $\{y_k\}$ is a martingale sequence and $\{z_k\}$ is the associated martingale difference sequence. In order to apply the Azuma-Hoeffding inequalities to get a high probability bound, we first need to bound the difference sequence $\{z_k\}$. We use the Bernstein inequality to bound the differences as follows.

$$
\begin{aligned}
z_k = y_k - y_{k-1} &= v_k - \nabla f(x_k) - (v_{k-1} - \nabla f(x_{k-1})) \\
&= \frac{1}{b} \sum_{i \in I_b} \left( \nabla f_i(x_k) - \nabla f_i(x_{k-1}) \right) + v_{k-1} - \nabla f(x_k) - (v_{k-1} - \nabla f(x_{k-1})) \\
&= \frac{1}{b} \sum_{i \in I_b} \left( \nabla f_i(x_k) - \nabla f_i(x_{k-1}) - (\nabla f(x_k) - \nabla f(x_{k-1})) \right).
\end{aligned}
\tag{37}
$$

We define $u_i := \nabla f_i(x_k) - \nabla f_i(x_{k-1}) - (\nabla f(x_k) - \nabla f(x_{k-1}))$, and then we have

$$
\|u_i\| = \|\nabla f_i(x_k) - \nabla f_i(x_{k-1}) - (\nabla f(x_k) - \nabla f(x_{k-1}))\| \le 2\|x_k - x_{k-1}\|,
\tag{38}
$$

where the last inequality holds due to the gradient Lipschitz Assumption 1. Then, consider the variance term $\sigma^2$

$$
\begin{aligned}
\sigma^2 &= \sum_{i \in I_b} \mathbb{E}[\|u_i\|^2] \\
&= \sum_{i \in I_b} \mathbb{E}[\|\nabla f_i(x_k) - \nabla f_i(x_{k-1}) - (\nabla f(x_k) - \nabla f(x_{k-1}))\|^2] \\
&\le \sum_{i \in I_b} \mathbb{E}[\|\nabla f_i(x_k) - \nabla f_i(x_{k-1})\|^2] \\
&\le bL^2 \|x_k - x_{k-1}\|^2,
\end{aligned}
\tag{39}
$$

where the first inequality uses the fact $\mathbb{E}[\|x - \mathbb{E}x\|^2] \le \mathbb{E}[\|x\|^2]$, and the last inequality uses the gradient Lipschitz Assumption 1. According to (38) and (39), we can bound the difference $z_k$ by Bernstein inequality (Proposition 2) as

$$
\begin{aligned}
\mathbb{P}\left\{ \|z_k\| \ge \frac{t}{b} \right\} &\le (d+1) \exp\left( \frac{-t^2/2}{\sigma^2 + Rt/3} \right) \\
&= (d+1) \exp\left( \frac{-t^2/2}{bL^2\|x_k - x_{k-1}\|^2 + 2\|x_k - x_{k-1}\|t/3} \right) \\
&= \zeta_k,
\end{aligned}
$$

where the last equality holds by letting $t = CL\sqrt{b}\|x_k - x_{k-1}\|$, where $C = O(\log \frac{d}{\zeta_k}) = \widetilde{O}(1)$. Now, we have a high probability bound for the difference sequence $\{z_k\}$, i.e.,

$$
\|z_k\| \le \frac{CL\|x_k - x_{k-1}\|}{\sqrt{b}} \quad \text{with probability } 1 - \zeta_k.
\tag{40}
$$

Now, we are ready to get a high probability bound for our original variance term (33) by using the martingale Azuma-Hoeffding inequality. Consider in a specific epoch $s$, i.e, iterations $t$ from $sm+1$ to current $sm + k$, where $k$ is less than $m$ (note that we only need to consider the current epoch since

each epoch we start with $y = 0$), we use a union bound for the difference sequence $\{z_t\}$ by letting $\zeta_k = \zeta/m$ such that

$$\|z_t\| \leq c_t = \frac{CL\|x_t - x_{t-1}\|}{\sqrt{b}} \quad \text{for all } sm+1 \leq t \leq sm+k \text{ with probability } 1 - \zeta. \quad (41)$$

Then according to Azuma-Hoeffding inequality (Proposition 4) and noting that $\zeta_k = \zeta/m$, we have

$$\mathbb{P}\Big\{\|y_{sm+k} - y_{sm}\| \geq \beta\Big\} \leq (d+1)\exp\Big(\frac{-\beta^2}{8\sum_{t=sm+1}^{sm+k} c_t^2}\Big) + \zeta$$

$$= 2\zeta,$$

where the last equality holds by letting $\beta = \sqrt{8\sum_{t=sm+1}^{sm+k} c_t^2 \log \frac{d}{\zeta}} = \frac{C'L\sqrt{\sum_{t=sm+1}^{sm+k}\|x_t - x_{t-1}\|^2}}{\sqrt{b}}$, where $C' = O(C\sqrt{\log \frac{d}{\zeta}}) = \widetilde{O}(1)$. Recall that $y_k := v_k - \nabla f(x_k)$ and at the beginning point of this epoch $y_{sm} = 0$ due to $v_{sm} = \nabla f(x_{sm})$ (see Line 5 of Algorithm 1), thus we have

$$\|v_{t-1} - \nabla f(x_{t-1})\| = \|y_{t-1}\| \leq \frac{C'L\sqrt{\sum_{j=sm+1}^{t-1}\|x_j - x_{j-1}\|^2}}{\sqrt{b}} \quad (42)$$

with probability $1 - 2\zeta$, where $t$ belongs to $[sm+1, (s+1)m]$.

Now, we use this high probability version (42) instead of the expectation one (33) to obtain the high probability bound for function value decrease (see (35)). We sum up (27) from the beginning of this epoch $s$, i.e., iterations from $sm$ to $t$, by plugging (42) into them to get:

$$f(x_t) \leq f(x_{sm}) - \frac{\eta}{2}\sum_{j=sm+1}^{t}\|\nabla f(x_{j-1})\|^2 - \Big(\frac{1}{2\eta} - \frac{L}{2}\Big)\sum_{j=sm+1}^{t}\|x_j - x_{j-1}\|^2$$

$$+ \frac{\eta}{2}\sum_{k=sm+1}^{t-1}\frac{C'^2L^2\sum_{j=sm+1}^{k}\|x_j - x_{j-1}\|^2}{b}$$

$$\leq f(x_{sm}) - \frac{\eta}{2}\sum_{j=sm+1}^{t}\|\nabla f(x_{j-1})\|^2 - \Big(\frac{1}{2\eta} - \frac{L}{2}\Big)\sum_{j=sm+1}^{t}\|x_j - x_{j-1}\|^2$$

$$+ \frac{\eta C'^2L^2}{2b}\sum_{k=sm+1}^{t-1}\sum_{j=sm+1}^{k}\|x_j - x_{j-1}\|^2$$

$$\leq f(x_{sm}) - \frac{\eta}{2}\sum_{j=sm+1}^{t}\|\nabla f(x_{j-1})\|^2 - \Big(\frac{1}{2\eta} - \frac{L}{2}\Big)\sum_{j=sm+1}^{t}\|x_j - x_{j-1}\|^2$$

$$+ \frac{\eta C'^2L^2(t-1-sm)}{2b}\sum_{j=sm+1}^{t}\|x_j - x_{j-1}\|^2$$

$$\leq f(x_{sm}) - \frac{\eta}{2}\sum_{j=sm+1}^{t}\|\nabla f(x_{j-1})\|^2 - \Big(\frac{1}{2\eta} - \frac{L}{2} - \frac{\eta C'^2L^2}{2}\Big)\sum_{j=sm+1}^{t}\|x_j - x_{j-1}\|^2 \quad (43)$$

$$\leq f(x_{sm}) - \frac{\eta}{2}\sum_{j=sm+1}^{t}\|\nabla f(x_{j-1})\|^2, \quad (44)$$

where (43) holds if the minibatch size $b \geq m$ (note that here $t \leq (s+1)m$), and (44) holds if the step size $\eta \leq \frac{\sqrt{4C'^2+1}-1}{2C'^2L}$.

Note that (44) only guarantees function value decrease when the summation of gradients in this epoch is large. However, in order to connect the guarantees between first situation (large gradients) and second situation (around saddle points), we need to show guarantees that are related to the *gradient of the starting point* of each epoch (see Line 3 of Algorithm 2). Similar to [15], we achieve this by stopping the epoch at a uniformly random point (see Line 16 of Algorithm 2).

Now we recall Lemma 1 to connect these two situations (large gradients and around saddle points):

**Lemma 1 (Connection of Two Situations)** *For any epoch $s$, let $x_t$ be a point uniformly sampled from this epoch $\{x_j\}_{j=sm}^{(s+1)m}$. Moreover, let the step size $\eta \leq \frac{\sqrt{4C'^2+1}-1}{2C'^2L}$ (where $C' = O(\log\frac{dn}{\zeta}) = \widetilde{O}(1)$) and the minibatch size $b \geq m$, there are two cases:*

1. *If at least half of points in this epoch have gradient norm no larger than $g_{\text{thres}}$, then $\|\nabla f(x_t)\| \leq g_{\text{thres}}$ holds with probability at least $1/2$;*

2. *Otherwise, we know $f(x_{sm}) - f(x_t) \geq \frac{\eta m g_{\text{thres}}^2}{8}$ holds with probability at least $1/5$.*

*Moreover, $f(x_t) \leq f(x_{sm})$ holds with high probability no matter which case happens.*

**Proof of Lemma 1.** There are two cases in this epoch:

1. If at least half of points of in this epoch $\{x_j\}_{j=sm}^{(s+1)m}$ have gradient norm no larger than $g_{\text{thres}}$, then it is easy to see that a uniformly sampled point $x_t$ has gradient norm $\|\nabla f(x_t)\| \leq g_{\text{thres}}$ with probability at least $1/2$.

2. Otherwise, at least half of points have gradient norm larger than $g_{\text{thres}}$. Then, as long as the sampled point $x_t$ falls into the last quarter of $\{x_j\}_{j=sm}^{(s+1)m}$, we know $\sum_{j=sm+1}^{t} \|\nabla f(x_{j-1})\|^2 \geq \frac{m g_{\text{thres}}^2}{4}$. This holds with probability at least $1/4$ since $x_t$ is uniformly sampled. Then combining with (44), i.e., $f(x_{sm}) - f(x_t) \geq \frac{\eta}{2}\sum_{j=sm+1}^{t}\|\nabla f(x_{j-1})\|^2$, we obtain the function value decrease $f(x_{sm}) - f(x_t) \geq \frac{\eta m g_{\text{thres}}^2}{8}$. Note that (44) holds with high probability if we choose the minibatch size $b \geq m$ and the step size $\eta \leq \frac{\sqrt{4C'^2+1}-1}{2C'^2L}$. By a union bound, the function value decrease $f(x_{sm}) - f(x_t) \geq \frac{\eta m g_{\text{thres}}^2}{8}$ with probability at least $1/5$.

Again according to (44), $f(x_t) \leq f(x_{sm})$ always holds with high probability. $\qquad\square$

Note that if Case 2 happens, the function value already decreases a lot in this epoch $s$ (corresponding to the first situation large gradients). Otherwise, Case 1 happens, we know the starting point of the next epoch $x_{(s+1)m} = x_t$ (i.e., Line 19 of Algorithm 2), then we know $\|\nabla f(x_{(s+1)m})\| = \|\nabla f(x_t)\| \leq g_{\text{thres}}$. Then we will start a super epoch (corresponding to the second situation around saddle points). Note that if $\lambda_{\min}(\nabla^2 f(x_{(s+1)m})) > -\delta$, this point $x_{(s+1)m}$ is already an $(\epsilon, \delta)$-second-order stationary point (recall that $g_{\text{thres}} = \epsilon$ in our Theorem 2).

**Around Saddle Points** $\|\nabla f(\widetilde{x})\| \leq g_{\text{thres}}$ and $\lambda_{\min}(\nabla^2 f(\widetilde{x})) \leq -\delta$: In this situation, we will show that the function value decreases a lot in a *super epoch* (instead of an epoch as in the first situation) with high probability by adding a random perturbation at the initial point $\widetilde{x}$. To simplify the presentation, we use $x_0 := \widetilde{x} + \xi$ to denote the starting point of the super epoch after the perturbation, where $\xi$ uniformly $\sim \mathbb{B}_0(r)$ and the perturbation radius is $r$ (see Line 6 in Algorithm 2). Following the classical widely used *two-point analysis* developed in [18], we consider two coupled points $x_0$ and $x_0'$ with $w_0 := x_0 - x_0' = r_0 e_1$, where $r_0$ is a scalar and $e_1$ denotes the smallest eigenvector direction of Hessian $\mathcal{H} := \nabla^2 f(\widetilde{x})$. Then we get two coupled sequences $\{x_t\}$ and $\{x_t'\}$ by running SSRGD update steps (Line 8–12 of Algorithm 2) with the same choice of minibatches (i.e., $I_b$'s in Line 12 of Algorithm 2) for a super epoch. We will show that at least one of these two coupled sequences will decrease the function value a lot (escape the saddle point), i.e.,

$$\exists t \leq t_{\text{thres}}, \text{ such that } \max\{f(x_0) - f(x_t), f(x_0') - f(x_t')\} \geq 2f_{\text{thres}}. \qquad (45)$$

We will prove (45) by contradiction. Assume the contrary, $f(x_0) - f(x_t) < 2f_{\text{thres}}$ and $f(x_0') - f(x_t') < 2f_{\text{thres}}$. First, we show that if function value does not decrease a lot, then all iteration points are not far from the starting point with high probability. Then we will show that the stuck region is relatively small in the random perturbation ball, i.e., at least one of $x_t$ and $x_t'$ will go far away from their starting point $x_0$ and $x_0'$ with high probability. Thus there is a contradiction. We recall these two lemmas here and their proofs are deferred to the end of this section.

**Lemma 2 (Localization)** *Let $\{x_t\}$ denote the sequence by running SSRGD update steps (Line 8–12 of Algorithm 2) from $x_0$. Moreover, let the step size $\eta \leq \frac{1}{2C'L}$ and minibatch size $b \geq m$, with*

*probability* $1 - \zeta$, *we have*

$$\forall t, \quad \|x_t - x_0\| \leq \sqrt{\frac{4t(f(x_0) - f(x_t))}{C'L}}, \tag{46}$$

*where* $C' = O(\log \frac{dt}{\zeta}) = \widetilde{O}(1)$.

**Lemma 3 (Small Stuck Region)** *If the initial point* $\widetilde{x}$ *satisfies* $-\gamma := \lambda_{\min}(\nabla^2 f(\widetilde{x})) \leq -\delta$, *then let* $\{x_t\}$ *and* $\{x_t'\}$ *be two coupled sequences by running SSRGD update steps (Line 8–12 of Algorithm 2) with the same choice of minibatches (i.e., $I_b$'s in Line 12) from $x_0$ and $x_0'$ with $w_0 := x_0 - x_0' = r_0 e_1$, where $x_0 \in \mathbb{B}_{\widetilde{x}}(r)$, $x_0' \in \mathbb{B}_{\widetilde{x}}(r)$, $r_0 = \frac{\zeta' r}{\sqrt{d}}$ and $e_1$ denotes the smallest eigenvector direction of Hessian $\nabla^2 f(\widetilde{x})$. Moreover, let the super epoch length $t_{\mathrm{thres}} = \frac{2 \log(\frac{8\delta\sqrt{d}}{C_1 \rho \zeta' r})}{\eta\delta} = \widetilde{O}(\frac{1}{\eta\delta})$, the step size $\eta \leq \min\left(\frac{1}{8\log(\frac{8\delta\sqrt{d}}{C_1\rho\zeta'r})L}, \frac{1}{4C_2 L \log t_{\mathrm{thres}}}\right) = \widetilde{O}(\frac{1}{L})$, minibatch size $b \geq m$ and the perturbation radius $r \leq \frac{\delta}{C_1\rho}$, then with probability $1 - \zeta$, we have*

$$\exists T \leq t_{\mathrm{thres}}, \quad \max\{\|x_T - x_0\|, \|x_T' - x_0'\|\} \geq \frac{\delta}{C_1\rho}, \tag{47}$$

*where* $C_1 \geq \frac{20 C_2}{\eta L}$ *and* $C_2 = O(\log \frac{dt_{\mathrm{thres}}}{\zeta}) = \widetilde{O}(1)$.

Based on these two lemmas, we are ready to show that (45) holds with high probability. Without loss of generality, we assume $\|x_T - x_0\| \geq \frac{\delta}{C_1\rho}$ in (47) (note that (46) holds for both $\{x_t\}$ and $\{x_t'\}$), then plugging it into (46) to obtain

$$\sqrt{\frac{4T(f(x_0) - f(x_T))}{C'L}} \geq \frac{\delta}{C_1\rho}$$

$$f(x_0) - f(x_T) \geq \frac{C'L\delta^2}{4C_1^2\rho^2 T}$$

$$\geq \frac{\eta C'L\delta^3}{8C_1^2\rho^2 \log(\frac{8\delta\sqrt{d}}{C_1\rho\zeta'r})}$$

$$= \frac{\delta^3}{C_1'\rho^2} \tag{48}$$

$$= 2f_{\mathrm{thres}},$$

where the last inequality is due to $T \leq t_{\mathrm{thres}}$ and (48) holds by letting $C_1' = \frac{8C_1^2 \log(\frac{8\delta\sqrt{d}}{C_1\rho\zeta'r})}{\eta C'L}$. Thus, we already prove that at least one of sequences $\{x_t\}$ and $\{x_t'\}$ escapes the saddle point with high probability, i.e.,

$$\exists T \leq t_{\mathrm{thres}}, \max\{f(x_0) - f(x_T), f(x_0') - f(x_T')\} \geq 2f_{\mathrm{thres}}, \tag{49}$$

if their starting points $x_0$ and $x_0'$ satisfying $w_0 := x_0 - x_0' = r_0 e_1$, where $r_0 = \frac{\zeta' r}{\sqrt{d}}$ and $e_1$ denotes the smallest eigenvector direction of Hessian $\mathcal{H} := \nabla^2 f(\widetilde{x})$. Similar to the classical argument in [18], we know that in the random perturbation ball, the stuck points can only be a short interval in the $e_1$ direction, i.e., at least one of two points in the $e_1$ direction will escape the saddle point if their distance is larger than $r_0 = \frac{\zeta' r}{\sqrt{d}}$. Thus, we know that the probability of the starting point $x_0 = \widetilde{x} + \xi$ (where $\xi$ uniformly $\sim \mathbb{B}_0(r)$) located in the stuck region is less than

$$\frac{r_0 V_{d-1}(r)}{V_d(r)} = \frac{r_0 \Gamma(\frac{d}{2} + 1)}{\sqrt{\pi} r \Gamma(\frac{d}{2} + \frac{1}{2})} \leq \frac{r_0}{\sqrt{\pi} r} \left(\frac{d}{2} + 1\right)^{1/2} \leq \frac{r_0 \sqrt{d}}{r} = \zeta', \tag{50}$$

where $V_d(r)$ denotes the volume of a Euclidean ball with radius $r$ in $d$ dimension, and the first inequality holds due to Gautschi's inequality. By a union bound for (50) and (48) (holds with high probability if $x_0$ is not in a stuck region), we know

$$f(x_0) - f(x_T) \geq 2f_{\mathrm{thres}} = \frac{\delta^3}{C_1'\rho^2} \tag{51}$$

with high probability. Note that the initial point of this super epoch is $\widetilde{x}$ before the perturbation (see Line 6 of Algorithm 2), thus we need to show that the perturbation step $x_0 = \widetilde{x} + \xi$ (where $\xi$ uniformly $\sim \mathbb{B}_0(r)$) does not increase the function value a lot, i.e.,

$$
\begin{aligned}
f(x_0) &\leq f(\widetilde{x}) + \langle \nabla f(\widetilde{x}), x_0 - \widetilde{x} \rangle + \frac{L}{2}\|x_0 - \widetilde{x}\|^2 \\
&\leq f(\widetilde{x}) + \|\nabla f(\widetilde{x})\|\|x_0 - \widetilde{x}\| + \frac{L}{2}\|x_0 - \widetilde{x}\|^2 \\
&\leq f(\widetilde{x}) + g_{\text{thres}} \cdot r + \frac{L}{2}r^2 \\
&\leq f(\widetilde{x}) + \frac{\delta^3}{2C_1'\rho^2} \\
&= f(\widetilde{x}) + f_{\text{thres}},
\end{aligned}
\tag{52}
$$

where the last inequality holds by letting the perturbation radius $r \leq \min\{\frac{\delta^3}{4C_1'\rho^2 g_{\text{thres}}}, \sqrt{\frac{\delta^3}{2C_1'\rho^2 L}}\}$.

Now we combine with (51) and (52) to obtain with high probability

$$
f(\widetilde{x}) - f(x_T) = f(\widetilde{x}) - f(x_0) + f(x_0) - f(x_T) \geq -f_{\text{thres}} + 2f_{\text{thres}} = \frac{\delta^3}{2C_1'\rho^2}.
\tag{53}
$$

Thus we have finished the proof for the second situation (around saddle points), i.e., we show that the function value decrease a lot ($f_{\text{thres}} = \frac{\delta^3}{2C_1'\rho^2}$) in a *super epoch* (recall that $T \leq t_{\text{thres}} = \frac{2\log(\frac{8\delta\sqrt{d}}{C_1\rho\zeta' r})}{\eta\delta}$) by adding a random perturbation $\xi \sim \mathbb{B}_0(r)$ at the initial point $\widetilde{x}$.

**Combing these two situations (large gradients and around saddle points) to prove Theorem 2:** First, we recall Theorem 2 here since we want to recall the parameter setting.

**Theorem 2** *Under Assumption 1 and 2 (i.e. (3) and (5)), let $\Delta f := f(x_0) - f^*$, where $x_0$ is the initial point and $f^*$ is the optimal value of $f$. By letting step size $\eta = \widetilde{O}(\frac{1}{L})$, epoch length $m = \sqrt{n}$, minibatch size $b = \sqrt{n}$, perturbation radius $r = \widetilde{O}\big(\min(\frac{\delta^3}{\rho^2\epsilon}, \frac{\delta^{3/2}}{\rho\sqrt{L}})\big)$, threshold gradient $g_{\text{thres}} = \epsilon$, threshold function value $f_{\text{thres}} = \widetilde{O}(\frac{\delta^3}{\rho^2})$ and super epoch length $t_{\text{thres}} = \widetilde{O}(\frac{1}{\eta\delta})$, SSRGD will at least once get to an $(\epsilon, \delta)$-second-order stationary point with high probability using*

$$
\widetilde{O}\Big(\frac{L\Delta f\sqrt{n}}{\epsilon^2} + \frac{L\rho^2\Delta f\sqrt{n}}{\delta^4} + \frac{\rho^2\Delta f n}{\delta^3}\Big)
$$

*stochastic gradients for nonconvex finite-sum problem* (1).

**Proof of Theorem 2.** Now, we prove this theorem by distinguishing the epochs into three types as follows:

1. *Type-1 useful epoch*: If at least half of points in this epoch have gradient norm larger than $g_{\text{thres}}$ (Case 2 of Lemma 1);

2. *Wasted epoch*: If at least half of points in this epoch have gradient norm no larger than $g_{\text{thres}}$ and the starting point of the next epoch has gradient norm larger than $g_{\text{thres}}$ (it means that this epoch does not guarantee decreasing the function value a lot as the large gradients situation, also it cannot connect to the second super epoch situation since the starting point of the next epoch has gradient norm larger than $g_{\text{thres}}$);

3. *Type-2 useful super epoch*: If at least half of points in this epoch have gradient norm no larger than $g_{\text{thres}}$ and the starting point of the next epoch (here we denote this point as $\widetilde{x}$) has gradient norm no larger than $g_{\text{thres}}$ (i.e., $\|\nabla f(\widetilde{x})\| \leq g_{\text{thres}}$) (Case 1 of Lemma 1), according to Line 3 of Algorithm 2, we will start a super epoch. So here we denote this epoch along with its following super epoch as a type-2 useful super epoch.

First, it is easy to see that the probability of a wasted epoch happened is less than $1/2$ due to the random stop (see Case 1 of Lemma 1 and Line 16 of Algorithm 2) and different wasted epoch are

independent. Thus, with high probability, there are at most $\widetilde{O}(1)$ wasted epochs happened before a type-1 useful epoch or type-2 useful super epoch. Now, we use $N_1$ and $N_2$ to denote the number of type-1 useful epochs and type-2 useful super epochs that the algorithm is needed. Recall that $\Delta f := f(x_0) - f^*$, where $x_0$ is the initial point and $f^*$ is the optimal value of $f$. Also recall that the function value always does not increase with high probability (see Lemma 1).

For type-1 useful epoch, according to Case 2 of Lemma 1, we know that the function value decreases at least $\frac{\eta m g_{\text{thres}}^2}{8}$ with probability at least $1/5$. Using a standard concentration, we know that with high probability $N_1$ type-1 useful epochs will decrease the function value at least $\frac{\eta m g_{\text{thres}}^2 N_1}{80}$, note that the function value can decrease at most $\Delta f$. So $\frac{\eta m g_{\text{thres}}^2 N_1}{80} \leq \Delta f$, we get $N_1 \leq \frac{80\Delta f}{\eta m g_{\text{thres}}^2}$.

For type-2 useful super epoch, first we know that the starting point of the super epoch $\widetilde{x}$ has gradient norm $\|\nabla f(\widetilde{x})\| \leq g_{\text{thres}}$. Now if $\lambda_{\min}(\nabla^2 f(\widetilde{x})) \geq -\delta$, then $\widetilde{x}$ is already a $(\epsilon, \delta)$-second-order stationary point. Otherwise, $\|\nabla f(\widetilde{x})\| \leq g_{\text{thres}}$ and $\lambda_{\min}(\nabla^2 f(\widetilde{x})) \leq -\delta$, this is exactly our second situation (around saddle points). According to (53), we know that the the function value decrease $(f(\widetilde{x}) - f(x_T))$ is at least $f_{\text{thres}} = \frac{\delta^3}{2C_1'\rho^2}$ with high probability. Similar to type-1 useful epoch, we know $N_2 \leq \frac{C_1''\rho^2 \Delta f}{\delta^3}$ by a union bound (so we change $C_1'$ to $C_1''$, anyway we also have $C_1'' = \widetilde{O}(1)$).

Now, we are ready to compute the convergence results to finish the proof for Theorem 2.

$$N_1(\widetilde{O}(1)n + n + mb) + N_2(\widetilde{O}(1)n + \lceil \frac{t_{\text{thres}}}{m} \rceil n + t_{\text{thres}}b)$$

$$\leq \widetilde{O}\Big( \frac{\Delta f n}{\eta m g_{\text{thres}}^2} + \frac{\rho^2 \Delta f}{\delta^3}(n + \frac{\sqrt{n}}{\eta\delta}) \Big)$$

$$\leq \widetilde{O}\Big( \frac{L\Delta f \sqrt{n}}{\epsilon^2} + \frac{L\rho^2 \Delta f \sqrt{n}}{\delta^4} + \frac{\rho^2 \Delta f n}{\delta^3} \Big) \tag{54}$$

$\square$

Now, the only remaining thing is to prove Lemma 2 and 3. We provide these two proofs as follows.

**Lemma 2 (Localization)** *Let $\{x_t\}$ denote the sequence by running SSRGD update steps (Line 8–12 of Algorithm 2) from $x_0$. Moreover, let the step size $\eta \leq \frac{1}{2C'L}$ and minibatch size $b \geq m$, with probability $1 - \zeta$, we have*

$$\forall t, \ \|x_t - x_0\| \leq \sqrt{\frac{4t(f(x_0) - f(x_t))}{C'L}},$$

*where $C' = O(\log \frac{dt}{\zeta}) = \widetilde{O}(1)$.*

**Proof of Lemma 2.** First, we assume the variance bound (42) holds for all $0 \leq j \leq t - 1$ (this is true with high probability using a union bound by letting $C' = O(\log \frac{dt}{\zeta})$). Then, according to (43), we know for any $\tau \leq t$ in some epoch $s$

$$f(x_\tau) \leq f(x_{sm}) - \frac{\eta}{2} \sum_{j=sm+1}^{\tau} \|\nabla f(x_{j-1})\|^2 - \Big( \frac{1}{2\eta} - \frac{L}{2} - \frac{\eta C'^2 L^2}{2} \Big) \sum_{j=sm+1}^{\tau} \|x_j - x_{j-1}\|^2$$

$$\leq f(x_{sm}) - \Big( \frac{1}{2\eta} - \frac{L}{2} - \frac{\eta C'^2 L^2}{2} \Big) \sum_{j=sm+1}^{\tau} \|x_j - x_{j-1}\|^2$$

$$\leq f(x_{sm}) - \frac{C'L}{4} \sum_{j=sm+1}^{\tau} \|x_j - x_{j-1}\|^2, \tag{55}$$

where the last inequality holds since the step size $\eta \leq \frac{1}{2C'L}$ and assuming $C' \geq 1$. Now, we sum up (55) for all epochs before iteration $t$,

$$f(x_t) \leq f(x_0) - \frac{C'L}{4} \sum_{j=1}^{t} \|x_j - x_{j-1}\|^2.$$

Then, the proof is finished as

$$\|x_t - x_0\| \le \sum_{j=1}^{t} \|x_j - x_{j-1}\| \le \sqrt{t \sum_{j=1}^{t} \|x_j - x_{j-1}\|^2} \le \sqrt{\frac{4t(f(x_0) - f(x_t))}{C'L}}.$$

$\square$

**Lemma 3 (Small Stuck Region)** *If the initial point $\widetilde{x}$ satisfies $-\gamma := \lambda_{\min}(\nabla^2 f(\widetilde{x})) \le -\delta$, then let $\{x_t\}$ and $\{x_t'\}$ be two coupled sequences by running SSRGD update steps (Line 8–12 of Algorithm 2) with the same choice of minibatches (i.e., $I_b$'s in Line 12) from $x_0$ and $x_0'$ with $w_0 := x_0 - x_0' = r_0 e_1$, where $x_0 \in \mathbb{B}_{\widetilde{x}}(r)$, $x_0' \in \mathbb{B}_{\widetilde{x}}(r)$, $r_0 = \frac{\zeta'r}{\sqrt{d}}$ and $e_1$ denotes the smallest eigenvector direction of Hessian $\nabla^2 f(\widetilde{x})$. Moreover, let the super epoch length $t_{\mathrm{thres}} = \frac{2\log(\frac{8\delta\sqrt{d}}{C_1\rho\zeta'r})}{\eta\delta} = \widetilde{O}(\frac{1}{\eta\delta})$, the step size $\eta \le \min\big(\frac{1}{8\log(\frac{8\delta\sqrt{d}}{C_1\rho\zeta'r})L}, \frac{1}{4C_2L\log t_{\mathrm{thres}}}\big) = \widetilde{O}(\frac{1}{L})$, minibatch size $b \ge m$ and the perturbation radius $r \le \frac{\delta}{C_1\rho}$, then with probability $1 - \zeta$, we have*

$$\exists T \le t_{\mathrm{thres}}, \quad \max\{\|x_T - x_0\|, \|x_T' - x_0'\|\} \ge \frac{\delta}{C_1\rho},$$

*where $C_1 \ge \frac{20C_2}{\eta L}$ and $C_2 = O(\log \frac{dt_{\mathrm{thres}}}{\zeta}) = \widetilde{O}(1)$.*

**Proof of Lemma 3.** We prove this lemma by contradiction. Assume the contrary,

$$\forall t \le t_{\mathrm{thres}}, \|x_t - x_0\| \le \frac{\delta}{C_1\rho} \text{ and } \|x_t' - x_0'\| \le \frac{\delta}{C_1\rho} \tag{56}$$

We will show that the distance between these two coupled sequences $w_t := x_t - x_t'$ will grow exponentially since they have a gap in the $e_1$ direction at the beginning, i.e., $w_0 := x_0 - x_0' = r_0 e_1$, where $r_0 = \frac{\zeta'r}{\sqrt{d}}$ and $e_1$ denotes the smallest eigenvector direction of Hessian $\mathcal{H} := \nabla^2 f(\widetilde{x})$. However, $\|w_t\| = \|x_t - x_t'\| \le \|x_t - x_0\| + \|x_0 - \widetilde{x}\| + \|x_t' - x_0'\| + \|x_0' - \widetilde{x}\| \le 2r + 2\frac{\delta}{C_1\rho}$ according to (56) and the perturbation radius $r$. It is not hard to see that the exponential increase will break this upper bound, thus we get a contradiction.

In the following, we prove the exponential increase of $w_t$ by induction. First, we need the expression of $w_t$ (recall that $x_t = x_{t-1} - \eta v_{t-1}$ (see Line 11 of Algorithm 2)):

$$\begin{aligned}
w_t &= w_{t-1} - \eta(v_{t-1} - v_{t-1}') \\
&= w_{t-1} - \eta\big(\nabla f(x_{t-1}) - \nabla f(x_{t-1}') + v_{t-1} - \nabla f(x_{t-1}) - v_{t-1}' + \nabla f(x_{t-1}')\big) \\
&= w_{t-1} - \eta\Big(\Big(\int_0^1 \nabla^2 f(x_{t-1}' + \theta(x_{t-1} - x_{t-1}'))d\theta(x_{t-1} - x_{t-1}') \\
&\qquad\qquad + v_{t-1} - \nabla f(x_{t-1}) - v_{t-1}' + \nabla f(x_{t-1}')\Big) \\
&= (I - \eta\mathcal{H})w_{t-1} - \eta(\Delta_{t-1}w_{t-1} + y_{t-1}) \\
&= (I - \eta\mathcal{H})^t w_0 - \eta\sum_{\tau=0}^{t-1}(I - \eta\mathcal{H})^{t-1-\tau}(\Delta_\tau w_\tau + y_\tau) \tag{57}
\end{aligned}$$

where $\Delta_\tau := \int_0^1 (\nabla^2 f(x_\tau' + \theta(x_\tau - x_\tau')) - \mathcal{H})d\theta$ and $y_\tau := v_\tau - \nabla f(x_\tau) - v_\tau' + \nabla f(x_\tau')$. Note that the first term of (57) is in the $e_1$ direction and is exponential with respect to $t$, i.e., $(1 + \eta\gamma)^t r_0 e_1$, where $-\gamma := \lambda_{\min}(\mathcal{H}) = \lambda_{\min}(\nabla^2 f(\widetilde{x})) \le -\delta$. To prove the exponential increase of $w_t$, it is sufficient to show that the first term of (57) will dominate the second term. We inductively prove the following two bounds

1. $\frac{1}{2}(1 + \eta\gamma)^t r_0 \le \|w_t\| \le \frac{3}{2}(1 + \eta\gamma)^t r_0$
2. $\|y_t\| \le \eta\gamma L(1 + \eta\gamma)^t r_0$

First, check the base case $t = 0$, $\|w_0\| = \|r_0 e_1\| = r_0$ and $\|y_0\| = \|v_0 - \nabla f(x_0) - v_0' + \nabla f(x_0')\| = \|\nabla f(x_0) - \nabla f(x_0) - \nabla f(x_0') + \nabla f(x_0')\| = 0$. Assume they hold for all $\tau \leq t - 1$, we now prove they hold for $t$ one by one. For Bound 1, it is enough to show the second term of (57) is dominated by half of the first term.

$$\left\| \eta \sum_{\tau=0}^{t-1} (I - \eta\mathcal{H})^{t-1-\tau}(\Delta_\tau w_\tau) \right\| \leq \eta \sum_{\tau=0}^{t-1} (1 + \eta\gamma)^{t-1-\tau} \|\Delta_\tau\| \|w_\tau\|$$

$$\leq \frac{3}{2}\eta(1 + \eta\gamma)^{t-1} r_0 \sum_{\tau=0}^{t-1} \|\Delta_\tau\| \tag{58}$$

$$\leq \frac{3}{2}\eta(1 + \eta\gamma)^{t-1} r_0 \sum_{\tau=0}^{t-1} \rho D_\tau^x \tag{59}$$

$$\leq \frac{3}{2}\eta(1 + \eta\gamma)^{t-1} r_0 t\rho \left(\frac{\delta}{C_1\rho} + r\right) \tag{60}$$

$$\leq \frac{3}{C_1}\eta\delta t(1 + \eta\gamma)^{t-1} r_0 \tag{61}$$

$$\leq \frac{6\log(\frac{8\delta\sqrt{d}}{C_1\rho\zeta'r})}{C_1}(1 + \eta\gamma)^{t-1} r_0 \tag{62}$$

$$\leq \frac{1}{4}(1 + \eta\gamma)^t r_0, \tag{63}$$

where (58) uses the induction for $w_\tau$ with $\tau \leq t - 1$, (59) uses the definition $D_\tau^x := \max\{\|x_\tau - \widetilde{x}\|, \|x_\tau' - \widetilde{x}\|\}$, (60) follows from $\|x_t - \widetilde{x}\| \leq \|x_t - x_0\| + \|x_0 - \widetilde{x}\| = \frac{\delta}{C_1\rho} + r$ due to (56) and the perturbation radius $r$, (61) holds by letting the perturbation radius $r \leq \frac{\delta}{C_1\rho}$, (62) holds since $t \leq t_{\text{thres}} = \frac{2\log(\frac{8\delta\sqrt{d}}{C_1\rho\zeta'r})}{\eta\delta}$, and (63) holds by letting $C_1 \geq 24\log(\frac{8\delta\sqrt{d}}{\rho\zeta'r})$.

$$\left\| \eta \sum_{\tau=0}^{t-1} (I - \eta\mathcal{H})^{t-1-\tau} y_\tau \right\| \leq \eta \sum_{\tau=0}^{t-1} (1 + \eta\gamma)^{t-1-\tau} \|y_\tau\|$$

$$\leq \eta \sum_{\tau=0}^{t-1} (1 + \eta\gamma)^{t-1-\tau} \eta\gamma L(1 + \eta\gamma)^\tau r_0 \tag{64}$$

$$= \eta\eta\gamma Lt(1 + \eta\gamma)^{t-1} r_0$$

$$\leq \eta\eta\gamma L\frac{2\log(\frac{8\delta\sqrt{d}}{C_1\rho\zeta'r})}{\eta\delta}(1 + \eta\gamma)^{t-1} r_0 \tag{65}$$

$$\leq 2\eta\log(\frac{8\delta\sqrt{d}}{C_1\rho\zeta'r})L(1 + \eta\gamma)^{t-1} r_0 \tag{66}$$

$$\leq \frac{1}{4}(1 + \eta\gamma)^t r_0, \tag{67}$$

where (64) uses the induction for $y_\tau$ with $\tau \leq t - 1$, (65) holds since $t \leq t_{\text{thres}} = \frac{2\log(\frac{8\delta\sqrt{d}}{C_1\rho\zeta'r})}{\eta\delta}$, (66) holds $\gamma \geq \delta$ (recall $-\gamma := \lambda_{\min}(\mathcal{H}) = \lambda_{\min}(\nabla^2 f(\widetilde{x})) \leq -\delta$), and (67) holds by letting $\eta \leq \frac{1}{8\log(\frac{8\delta\sqrt{d}}{C_1\rho\zeta'r})L}$.

Combining (63) and (67), we proved the second term of (57) is dominated by half of the first term. Note that the first term of (57) is $\|(I - \eta\mathcal{H})^t w_0\| = (1 + \eta\gamma)^t r_0$. Thus, we have

$$\frac{1}{2}(1 + \eta\gamma)^t r_0 \leq \|w_t\| \leq \frac{3}{2}(1 + \eta\gamma)^t r_0 \tag{68}$$

Now, the remaining thing is to prove the second bound $\|y_t\| \leq \eta\gamma L(1 + \eta\gamma)^t r_0$. First, we write the concrete expression of $y_t$:

$$
\begin{aligned}
y_t &= v_t - \nabla f(x_t) - v'_t + \nabla f(x'_t) \\
&= \frac{1}{b}\sum_{i\in I_b}\big(\nabla f_i(x_t) - \nabla f_i(x_{t-1})\big) + v_{t-1} - \nabla f(x_t) \\
&\quad - \frac{1}{b}\sum_{i\in I_b}\big(\nabla f_i(x'_t) - \nabla f_i(x'_{t-1})\big) - v'_{t-1} + \nabla f(x'_t) \\
&= \frac{1}{b}\sum_{i\in I_b}\big(\nabla f_i(x_t) - \nabla f_i(x_{t-1})\big) + \nabla f(x_{t-1}) - \nabla f(x_t) \\
&\quad - \frac{1}{b}\sum_{i\in I_b}\big(\nabla f_i(x'_t) - \nabla f_i(x'_{t-1})\big) - \nabla f(x'_{t-1}) + \nabla f(x'_t) \\
&\quad + v_{t-1} - \nabla f(x_{t-1}) - v'_{t-1} + \nabla f(x'_{t-1}) \\
&= \frac{1}{b}\sum_{i\in I_b}\big(\nabla f_i(x_t) - \nabla f_i(x'_t) - \nabla f_i(x_{t-1}) + \nabla f_i(x'_{t-1})\big) \\
&\quad - \big(\nabla f(x_t) - \nabla f(x'_t) - \nabla f(x_{t-1}) + \nabla f(x'_{t-1})\big) + y_{t-1},
\end{aligned}
\tag{69}
$$

where (69) due to the definition of the estimator $v_t$ (see Line 12 of Algorithm 2). We further define the difference $z_t := y_t - y_{t-1}$. It is not hard to verify that $\{y_t\}$ is a martingale sequence and $\{z_t\}$ is the associated martingale difference sequence. We will apply the Azuma-Hoeffding inequalities to get an upper bound for $\|y_t\|$ and then we prove $\|y_t\| \leq \eta\gamma L(1 + \eta\gamma)^t r_0$ based on that upper bound. In order to apply the Azuma-Hoeffding inequalities for martingale sequence $\|y_t\|$, we first need to bound the difference sequence $\{z_t\}$. We use the Bernstein inequality to bound the differences as follows.

$$
\begin{aligned}
z_t = y_t - y_{t-1} &= \frac{1}{b}\sum_{i\in I_b}\big(\nabla f_i(x_t) - \nabla f_i(x'_t) - \nabla f_i(x_{t-1}) + \nabla f_i(x'_{t-1})\big) \\
&\quad - \big(\nabla f(x_t) - \nabla f(x'_t) - \nabla f(x_{t-1}) + \nabla f(x'_{t-1})\big) \\
&= \frac{1}{b}\sum_{i\in I_b}\Big(\big(\nabla f_i(x_t) - \nabla f_i(x'_t)\big) - \big(\nabla f_i(x_{t-1}) - \nabla f_i(x'_{t-1})\big) \\
&\quad - \big(\nabla f(x_t) - \nabla f(x'_t)\big) + \big(\nabla f(x_{t-1}) - \nabla f(x'_{t-1})\big)\Big).
\end{aligned}
\tag{70}
$$

We define $u_i := \big(\nabla f_i(x_t) - \nabla f_i(x'_t)\big) - \big(\nabla f_i(x_{t-1}) - \nabla f_i(x'_{t-1})\big) - \big(\nabla f(x_t) - \nabla f(x'_t)\big) + \big(\nabla f(x_{t-1}) - \nabla f(x'_{t-1})\big)$, and then we have

$$
\begin{aligned}
\|u_i\| &= \|\big(\nabla f_i(x_t) - \nabla f_i(x'_t)\big) - \big(\nabla f_i(x_{t-1}) - \nabla f_i(x'_{t-1})\big) \\
&\quad - \big(\nabla f(x_t) - \nabla f(x'_t)\big) + \big(\nabla f(x_{t-1}) - \nabla f(x'_{t-1})\big)\| \\
&\leq \Big\|\int_0^1 \nabla^2 f_i(x'_t + \theta(x_t - x'_t))d\theta(x_t - x'_t) - \int_0^1 \nabla^2 f_i(x'_{t-1} + \theta(x_{t-1} - x'_{t-1}))d\theta(x_{t-1} - x'_{t-1}) \\
&\quad - \int_0^1 \nabla^2 f(x'_t + \theta(x_t - x'_t))d\theta(x_t - x'_t) + \int_0^1 \nabla^2 f(x'_{t-1} + \theta(x_{t-1} - x'_{t-1}))d\theta(x_{t-1} - x'_{t-1})\Big\| \\
&= \|\mathcal{H}_i w_t + \Delta_t^i w_t - (\mathcal{H}_i w_{t-1} + \Delta_{t-1}^i w_{t-1}) - (\mathcal{H} w_t + \Delta_t w_t) + (\mathcal{H} w_{t-1} + \Delta_{t-1} w_{t-1})\|
\end{aligned}
\tag{71}
$$

$$
\begin{aligned}
&\leq \|(\mathcal{H}_i - \mathcal{H})(w_t - w_{t-1})\| + \|(\Delta_t^i - \Delta_t)w_t - (\Delta_{t-1}^i - \Delta_{t-1})w_{t-1}\| \\
&\leq 2L\|w_t - w_{t-1}\| + 2\rho D_t^x\|w_t\| + 2\rho D_{t-1}^x\|w_{t-1}\|,
\end{aligned}
\tag{72}
$$

where (71) holds since we define $\Delta_t := \int_0^1(\nabla^2 f(x'_t + \theta(x_t - x'_t)) - \mathcal{H})d\theta$ and $\Delta_t^i := \int_0^1(\nabla^2 f_i(x'_t + \theta(x_t - x'_t)) - \mathcal{H}_i)d\theta$, and the last inequality holds due to the gradient Lipschitz Assumption 1 and Hessian Lipschitz Assumption 2 (recall $D_t^x := \max\{\|x_t - \widetilde{x}\|, \|x'_t - \widetilde{x}\|\}$). Then, consider the

variance term $\sigma^2$

$$\sigma^2 = \sum_{i \in I_b} \mathbb{E}[\|u_i\|^2]$$

$$\leq \sum_{i \in I_b} \mathbb{E}[\|\left(\nabla f_i(x_t) - \nabla f_i(x'_t)\right) - \left(\nabla f_i(x_{t-1}) - \nabla f_i(x'_{t-1})\right)\|^2]$$

$$= \sum_{i \in I_b} \mathbb{E}[\|\mathcal{H}_i w_t + \Delta_t^i w_t - (\mathcal{H}_i w_{t-1} + \Delta_{t-1}^i w_{t-1})\|^2]$$

$$\leq b(L\|w_t - w_{t-1}\| + \rho D_t^x \|w_t\| + \rho D_{t-1}^x \|w_{t-1}\|)^2, \tag{73}$$

where the first inequality uses the fact $\mathbb{E}[\|x - \mathbb{E}x\|^2] \leq \mathbb{E}[\|x\|^2]$, and the last inequality uses the gradient Lipschitz Assumption 1 and Hessian Lipschitz Assumption 2. According to (72) and (73), we can bound the difference $z_k$ by Bernstein inequality (Proposition 2) as (where $R = 2L\|w_t - w_{t-1}\| + 2\rho D_t^x\|w_t\| + 2\rho D_{t-1}^x\|w_{t-1}\|$ and $\sigma^2 = b(L\|w_t - w_{t-1}\| + \rho D_t^x\|w_t\| + \rho D_{t-1}^x\|w_{t-1}\|)^2$)

$$\mathbb{P}\left\{\|z_t\| \geq \frac{\alpha}{b}\right\} \leq (d+1)\exp\left(\frac{-\alpha^2/2}{\sigma^2 + R\alpha/3}\right) = \zeta_k,$$

where the last equality holds by letting $\alpha = C_4\sqrt{b}(L\|w_t - w_{t-1}\| + \rho D_t^x\|w_t\| + \rho D_{t-1}^x\|w_{t-1}\|)$, where $C_4 = O(\log \frac{d}{\zeta_k}) = \widetilde{O}(1)$.

Now, we have a high probability bound for the difference sequence $\{z_k\}$, i.e.,

$$\|z_k\| \leq c_k = \frac{C_4(L\|w_t - w_{t-1}\| + \rho D_t^x\|w_t\| + \rho D_{t-1}^x\|w_{t-1}\|)}{\sqrt{b}} \quad \text{with probability } 1 - \zeta_k. \tag{74}$$

Now, we are ready to get an upper bound for $y_t$ by using the martingale Azuma-Hoeffding inequality. Note that we only need to consider the current epoch that contains the iteration $t$ since each epoch we start with $y = 0$. Let $s$ denote the current epoch, i.e, iterations from $sm + 1$ to current $t$, where $t$ is no larger than $(s+1)m$. According to Azuma-Hoeffding inequality (Proposition 4) and letting $\zeta_k = \zeta/m$, we have

$$\mathbb{P}\left\{\|y_t - y_{sm}\| \geq \beta\right\} \leq (d+1)\exp\left(\frac{-\beta^2}{8\sum_{k=sm+1}^{t} c_k^2}\right) + \zeta$$

$$= 2\zeta,$$

where the last equality holds by letting $\beta = \sqrt{8\sum_{k=sm+1}^{t} c_k^2 \log \frac{d}{\zeta}} = \frac{C_3\sqrt{\sum_{k=sm+1}^{t}(L\|w_t - w_{t-1}\| + \rho D_t^x\|w_t\| + \rho D_{t-1}^x\|w_{t-1}\|)^2}}{\sqrt{b}}$, where $C_3 = O(C_4\sqrt{\log \frac{d}{\zeta}}) = \widetilde{O}(1)$. Recall that $y_k := v_k - \nabla f(x_k) - v'_k + \nabla f(x'_k)$ and at the beginning point of this epoch $y_{sm} = 0$ due to $v_{sm} = \nabla f(x_{sm})$ and $v'_{sm} = \nabla f(x'_{sm})$ (see Line 5 of Algorithm 1), thus we have

$$\|y_t\| = \|y_t - y_{sm}\| \leq \frac{C_3\sqrt{\sum_{k=sm+1}^{t}(L\|w_t - w_{t-1}\| + \rho D_t^x\|w_t\| + \rho D_{t-1}^x\|w_{t-1}\|)^2}}{\sqrt{b}} \tag{75}$$

with probability $1 - 2\zeta$, where $t$ belongs to $[sm + 1, (s+1)m]$. Note that we can further relax the parameter $C_3$ in (75) to $C_2 = O(\log \frac{dt_{\text{thres}}}{\zeta})$ (see (76)) for making sure the above arguments hold with probability $1 - \zeta$ for all $t \leq t_{\text{thres}}$ by using a union bound for $\zeta_t$'s:

$$\|y_t\| = \|y_t - y_{sm}\| \leq \frac{C_2\sqrt{\sum_{k=sm+1}^{t}(L\|w_t - w_{t-1}\| + \rho D_t^x\|w_t\| + \rho D_{t-1}^x\|w_{t-1}\|)^2}}{\sqrt{b}}. \tag{76}$$

Now, we will show how to bound the right-hand-side of (76) to finish the proof, i.e., prove the remaining second bound $\|y_t\| \leq \eta\gamma L(1 + \eta\gamma)^t r_0$.

First, we show that the last two terms in the right-hand-side of (76) can be bounded as

$$\rho D_t^x \|w_t\| + \rho D_{t-1}^x \|w_{t-1}\| \leq \rho \left(\frac{\delta}{C_1 \rho} + r\right)\frac{3}{2}(1+\eta\gamma)^t r_0 + \rho\left(\frac{\delta}{C_1\rho} + r\right)\frac{3}{2}(1+\eta\gamma)^{t-1} r_0$$

$$\leq 3\rho\left(\frac{\delta}{C_1\rho} + r\right)(1+\eta\gamma)^t r_0$$

$$\leq \frac{6\delta}{C_1}(1+\eta\gamma)^t r_0, \tag{77}$$

where the first inequality follows from the induction of $\|w_{t-1}\| \leq \frac{3}{2}(1+\eta\gamma)^{t-1} r_0$ and the already proved $\|w_t\| \leq \frac{3}{2}(1+\eta\gamma)^t r_0$ in (68), and the last inequality holds by letting the perturbation radius $r \leq \frac{\delta}{C_1\rho}$.

Now, we show that the first term of right-hand-side of (76) can be bounded as

$$L\|w_t - w_{t-1}\|$$

$$= L\Big\| - \eta\mathcal{H}(I - \eta\mathcal{H})^{t-1} w_0 - \eta \sum_{\tau=0}^{t-2} \eta\mathcal{H}(I - \eta\mathcal{H})^{t-2-\tau}(\Delta_\tau w_\tau + y_\tau) + \eta(\Delta_{t-1} w_{t-1} + y_{t-1})\Big\|$$

$$\leq L\eta\gamma(1+\eta\gamma)^{t-1} r_0 + L\Big\|\eta\sum_{\tau=0}^{t-2} \eta\mathcal{H}(I - \eta\mathcal{H})^{t-2-\tau}(\Delta_\tau w_\tau + y_\tau)\Big\| + L\|\eta(\Delta_{t-1} w_{t-1} + y_{t-1})\|$$

$$\leq L\eta\gamma(1+\eta\gamma)^{t-1} r_0 + L\eta\Big\|\sum_{\tau=0}^{t-2} \eta\mathcal{H}(I - \eta\mathcal{H})^{t-2-\tau}\Big\| \max_{0\leq k\leq t-2} \|\Delta_k w_k + y_k\|$$

$$+ L\eta\rho\left(\frac{\delta}{C_1\rho} + r\right)\|w_{t-1}\| + L\eta\|y_{t-1}\| \tag{78}$$

$$\leq L\eta\gamma(1+\eta\gamma)^{t-1} r_0 + L\eta \sum_{\tau=0}^{t-2} \frac{1}{t-1-\tau} \max_{0\leq k\leq t-2} \|\Delta_k w_k + y_k\|$$

$$+ L\eta\rho\left(\frac{\delta}{C_1\rho} + r\right)\|w_{t-1}\| + L\eta\|y_{t-1}\| \tag{79}$$

$$\leq L\eta\gamma(1+\eta\gamma)^{t-1} r_0 + L\eta \log t \max_{0\leq k\leq t-2} \|\Delta_k w_k + y_k\|$$

$$+ L\eta\rho\left(\frac{\delta}{C_1\rho} + r\right)\|w_{t-1}\| + L\eta\|y_{t-1}\|$$

$$\leq L\eta\gamma(1+\eta\gamma)^{t-1} r_0 + L\eta \log t \max_{0\leq k\leq t-2} \|\Delta_k w_k + y_k\|$$

$$+ L\eta\rho\left(\frac{\delta}{C_1\rho} + r\right)\frac{3}{2}(1+\eta\gamma)^{t-1} r_0 + L\eta\eta\gamma L(1+\eta\gamma)^{t-1} r_0 \tag{80}$$

$$\leq L\eta\gamma(1+\eta\gamma)^{t-1} r_0 + L\eta \log t \left(\rho\left(\frac{\delta}{C_1\rho} + r\right)\frac{3}{2}(1+\eta\gamma)^{t-2} r_0 + \eta\gamma L(1+\eta\gamma)^{t-2} r_0\right)$$

$$+ L\eta\rho\left(\frac{\delta}{C_1\rho} + r\right)\frac{3}{2}(1+\eta\gamma)^{t-1} r_0 + L\eta\eta\gamma L(1+\eta\gamma)^{t-1} r_0 \tag{81}$$

$$\leq L\eta\gamma(1+\eta\gamma)^{t-1} r_0 + L\eta \log t \left(\frac{3\delta}{C_1}(1+\eta\gamma)^{t-2} r_0 + \eta\gamma L(1+\eta\gamma)^{t-2} r_0\right)$$

$$+ \frac{3L\eta\delta}{C_1}(1+\eta\gamma)^{t-1} r_0 + L\eta\eta\gamma L(1+\eta\gamma)^{t-1} r_0 \tag{82}$$

$$\leq \left(\frac{4}{C_1}\log t + 2L\eta \log t\right)\eta\gamma L(1+\eta\gamma)^t r_0, \tag{83}$$

where the first equality follows from (57), (78) holds from the following (84),

$$\|\Delta_t\| \leq \rho D_t^x \leq \rho\left(\frac{\delta}{C_1\rho} + r\right), \tag{84}$$

where (84) holds due to Hessian Lipschitz Assumption 2, (56) and the perturbation radius $r$ (recall that $\Delta_t := \int_0^1 (\nabla^2 f(x_t' + \theta(x_t - x_t')) - \mathcal{H})d\theta$, $\mathcal{H} := \nabla^2 f(\widetilde{x})$ and $D_t^x := \max\{\|x_t - \widetilde{x}\|, \|x_t' - \widetilde{x}\|\}$), (79)

holds due to $\|\eta\mathcal{H}(I-\eta\mathcal{H})^t\| \leq \frac{1}{t+1}$, (80) holds by plugging the induction $\|w_{t-1}\| \leq \frac{3}{2}(1+\eta\gamma)^{t-1}r_0$ and $\|y_{t-1}\| \leq \eta\gamma L(1+\eta\gamma)^{t-1}r_0$, (81) follows from (84), the induction $\|w_k\| \leq \frac{3}{2}(1+\eta\gamma)^k r_0$ and $\|y_k\| \leq \eta\gamma L(1+\eta\gamma)^k r_0$ (hold for all $k \leq t-1$), (82) holds by letting the perturbation radius $r \leq \frac{\delta}{C_1\rho}$, and the last inequality holds due to $\gamma \geq \delta$ (recall $-\gamma := \lambda_{\min}(\mathcal{H}) = \lambda_{\min}(\nabla^2 f(\widetilde{x})) \leq -\delta$).

By plugging (77) and (83) into (76), we have

$$
\begin{aligned}
\|y_t\| &\leq C_2\left(\frac{6\delta}{C_1}(1+\eta\gamma)^t r_0 + \left(\frac{4}{C_1}\log t + 2L\eta\log t\right)\eta\gamma L(1+\eta\gamma)^t r_0\right) \\
&\leq C_2\left(\frac{6}{C_1\eta L} + \frac{4}{C_1}\log t + 2L\eta\log t\right)\eta\gamma L(1+\eta\gamma)^t r_0 \\
&\leq \eta\gamma L(1+\eta\gamma)^t r_0,
\end{aligned}
\tag{85}
$$

where the second inequality holds due to $\gamma \geq \delta$, and the last inequality holds by letting $C_1 \geq \frac{20C_2}{\eta L}$ and $\eta \leq \frac{1}{4C_2 L \log t}$. Recall that $C_2 = O(\log\frac{dt_{\text{thres}}}{\zeta})$ is enough to let the arguments in this proof hold with probability $1-\zeta$ for all $t \leq t_{\text{thres}}$.

From (68) and (85), we know that the two induction bounds hold for $t$. We recall the first induction bound here:

1. $\frac{1}{2}(1+\eta\gamma)^t r_0 \leq \|w_t\| \leq \frac{3}{2}(1+\eta\gamma)^t r_0$

Thus, we know that $\|w_t\| \geq \frac{1}{2}(1+\eta\gamma)^t r_0 = \frac{1}{2}(1+\eta\gamma)^t\frac{\zeta' r}{\sqrt{d}}$. However, $\|w_t\| := \|x_t - x_t'\| \leq \|x_t - x_0\| + \|x_0 - \widetilde{x}\| + \|x_t' - x_0'\| + \|x_0' - \widetilde{x}\| \leq 2r + 2\frac{\delta}{C_1\rho} \leq \frac{4\delta}{C_1\rho}$ according to (56) and the perturbation radius $r$. The last inequality is due to the perturbation radius $r \leq \frac{\delta}{C_1\rho}$ (we already used this condition in the previous arguments). This will give a contradiction for (56) if $\frac{1}{2}(1+\eta\gamma)^t\frac{\zeta' r}{\sqrt{d}} \geq \frac{4\delta}{C_1\rho}$ and it will happen if $t \geq \frac{2\log(\frac{8\delta\sqrt{d}}{C_1\rho\zeta' r})}{\eta\delta}$.

So the proof of this lemma is finished by contradiction if we let $t_{\text{thres}} := \frac{2\log(\frac{8\delta\sqrt{d}}{C_1\rho\zeta' r})}{\eta\delta}$, i.e., we have

$$
\exists T \leq t_{\text{thres}}, \quad \max\{\|x_T - x_0\|, \|x_T' - x_0'\|\} \geq \frac{\delta}{C_1\rho}.
$$

$\square$

## C.2 Proofs for Online Problem

In this section, we provide the detailed proofs for online problem (2) (i.e., Theorem 3–4). We will reuse some parts of our previous proofs for finite-sum problem (1) in previous Section C.1.

First, we recall the previous key relation (27) between $f(x_t)$ and $f(x_{t-1})$ as follows (recall $x_t := x_{t-1} - \eta v_{t-1}$):

$$f(x_t) \leq f(x_{t-1}) + \frac{\eta}{2}\|\nabla f(x_{t-1}) - v_{t-1}\|^2 - \frac{\eta}{2}\|\nabla f(x_{t-1})\|^2 - \left(\frac{1}{2\eta} - \frac{L}{2}\right)\|x_t - x_{t-1}\|^2. \quad (86)$$

Next, we recall the previous bound (31) for the variance term:

$$\mathbb{E}[\|v_{t-1} - \nabla f(x_{t-1})\|^2] \leq \frac{L^2}{b}\mathbb{E}[\|x_{t-1} - x_{t-2}\|^2] + \mathbb{E}[\|v_{t-2} - \nabla f(x_{t-2})\|^2]. \quad (87)$$

Now, the following bound for the variance term will be different from the previous finite-sum case. Similar to (32), we sum up (87) from the beginning of this epoch $sm$ to the point $t-1$,

$$\mathbb{E}[\|v_{t-1} - \nabla f(x_{t-1})\|^2] \leq \frac{L^2}{b}\sum_{j=sm+1}^{t-1}\mathbb{E}[\|x_j - x_{j-1}\|^2] + \mathbb{E}[\|v_{sm} - \nabla f(x_{sm})\|^2] \quad (88)$$

$$= \frac{L^2}{b}\sum_{j=sm+1}^{t-1}\mathbb{E}[\|x_j - x_{j-1}\|^2] + \mathbb{E}\left[\left\|\frac{1}{B}\sum_{j\in I_B}\nabla f_j(x_{sm}) - \nabla f(x_{sm})\right\|^2\right] \quad (89)$$

$$\leq \frac{L^2}{b}\sum_{j=sm+1}^{t-1}\mathbb{E}[\|x_j - x_{j-1}\|^2] + \frac{\sigma^2}{B}, \quad (90)$$

where (88) is the same as (32), (89) uses the modification (11) (i.e., $v_{sm} = \frac{1}{B}\sum_{j\in I_B}\nabla f_j(x_{sm})$ instead of the full gradient computation $v_{sm} = \nabla f(x_{sm})$ in the finite-sum case), and the last inequality (90) follows from the bounded variance Assumption 3.

Now, we take expectations for (86) and then sum it up from the beginning of this epoch $s$, i.e., iterations from $sm$ to $t$, by plugging the variance (90) into them to get:

$$\mathbb{E}[f(x_t)] \leq \mathbb{E}[f(x_{sm})] - \frac{\eta}{2}\sum_{j=sm+1}^{t}\mathbb{E}[\|\nabla f(x_{j-1})\|^2] - \left(\frac{1}{2\eta} - \frac{L}{2}\right)\sum_{j=sm+1}^{t}\mathbb{E}[\|x_j - x_{j-1}\|^2]$$

$$+ \frac{\eta L^2}{2b}\sum_{k=sm+1}^{t-1}\sum_{j=sm+1}^{k}\mathbb{E}[\|x_j - x_{j-1}\|^2] + \frac{\eta}{2}\sum_{j=sm+1}^{t}\frac{\sigma^2}{B}$$

$$\leq \mathbb{E}[f(x_{sm})] - \frac{\eta}{2}\sum_{j=sm+1}^{t}\mathbb{E}[\|\nabla f(x_{j-1})\|^2] - \left(\frac{1}{2\eta} - \frac{L}{2}\right)\sum_{j=sm+1}^{t}\mathbb{E}[\|x_j - x_{j-1}\|^2]$$

$$+ \frac{\eta L^2(t-1-sm)}{2b}\sum_{j=sm+1}^{t}\mathbb{E}[\|x_j - x_{j-1}\|^2] + \frac{(t-sm)\eta\sigma^2}{2B}$$

$$\leq \mathbb{E}[f(x_{sm})] - \frac{\eta}{2}\sum_{j=sm+1}^{t}\mathbb{E}[\|\nabla f(x_{j-1})\|^2] - \left(\frac{1}{2\eta} - \frac{L}{2}\right)\sum_{j=sm+1}^{t}\mathbb{E}[\|x_j - x_{j-1}\|^2]$$

$$+ \frac{\eta L^2}{2}\sum_{j=sm+1}^{t}\mathbb{E}[\|x_j - x_{j-1}\|^2] + \frac{(t-sm)\eta\sigma^2}{2B} \quad (91)$$

$$\leq \mathbb{E}[f(x_{sm})] - \frac{\eta}{2}\sum_{j=sm+1}^{t}\mathbb{E}[\|\nabla f(x_{j-1})\|^2] + \frac{(t-sm)\eta\sigma^2}{2B}, \quad (92)$$

where (91) holds if the minibatch size $b \geq m$ (note that here $t \leq (s+1)m$), (92) holds if the step size $\eta \leq \frac{\sqrt{5}-1}{2L}$.

**Proof of Theorem 3.** Let $b = m = \frac{2\sigma}{\epsilon}$ and step size $\eta \leq \frac{\sqrt{5}-1}{2L}$, then (92) holds. Now, the proof is directly obtained by summing up (92) for all epochs $0 \leq s \leq S$ as follows:

$$\mathbb{E}[f(x_T)] \leq \mathbb{E}[f(x_0)] - \frac{\eta}{2} \sum_{j=1}^{T} \mathbb{E}[\|\nabla f(x_{j-1})\|^2] + \frac{T\eta\sigma^2}{2B}$$

$$\mathbb{E}[\|\nabla f(\hat{x})\|] \leq \sqrt{\mathbb{E}[\|\nabla f(\hat{x})\|^2]} \leq \sqrt{\frac{2(f(x_0) - f^*)}{\eta T} + \frac{\sigma^2}{B}} = \frac{\epsilon}{2} + \frac{\epsilon}{2} = \epsilon, \tag{93}$$

where (93) holds by choosing $\hat{x}$ uniformly from $\{x_{t-1}\}_{t \in [T]}$ and letting $Sm \leq T = \frac{8(f(x_0) - f^*)}{\eta\epsilon^2} = O(\frac{L(f(x_0) - f^*)}{\epsilon^2})$ and $B = \frac{4\sigma^2}{\epsilon^2}$. Note that the total number of computation of stochastic gradients equals to

$$SB + Smb \leq \left\lceil \frac{T}{m} \right\rceil B + Tb \leq \left( \frac{T}{2\sigma/\epsilon} + 1 \right) \frac{4\sigma^2}{\epsilon^2} + T \frac{2\sigma}{\epsilon} = O\left( \frac{\sigma^2}{\epsilon^2} + \frac{L(f(x_0) - f^*)\sigma}{\epsilon^3} \right).$$

$\square$

### C.2.1 Proof of Theorem 4

Similar to the proof of Theorem 2, for proving the second-order guarantee, we will divide the proof into two situations. The first situation (**large gradients**) is also almost the same as the above arguments for first-order guarantee, where the function value will decrease a lot since the gradients are large (see (92)). For the second situation (**around saddle points**), we will show that the function value can also decrease a lot by adding a random perturbation. The reason is that saddle points are usually unstable and the stuck region is relatively small in a random perturbation ball.

**Large Gradients**: First, we need a high probability bound for the variance term instead of the expectation one (90). Then we use it to get a high probability bound of (92) for function value decrease. Note that in this online case, $v_{sm} = \frac{1}{B} \sum_{j \in I_B} \nabla f_j(x_{sm})$ at the beginning of each epoch (see (11)) instead of $v_{sm} = \nabla f(x_{sm})$ in the previous finite-sum case. Thus we first need a high probability bound for $\|v_{sm} - \nabla f(x_{sm})\|$. According to Assumption 4, we have

$$\|\nabla f_j(x) - \nabla f(x)\| \leq \sigma,$$

$$\sum_{j \in I_B} \|\nabla f_j(x) - \nabla f(x)\|^2 \leq B\sigma^2.$$

By applying Bernstein inequality (Proposition 2), we get the high probability bound for $\|v_{sm} - \nabla f(x_{sm})\|$ as follows:

$$\mathbb{P}\left\{ \|v_{sm} - \nabla f(x_{sm})\| \geq \frac{t}{B} \right\} \leq (d+1) \exp\left( \frac{-t^2/2}{B\sigma^2 + \sigma t/3} \right) = \zeta,$$

where the last equality holds by letting $t = C\sqrt{B}\sigma$, where $C = O(\log \frac{d}{\zeta}) = \widetilde{O}(1)$. Now, we have a high probability bound for $\|v_{sm} - \nabla f(x_{sm})\|$, i.e.,

$$\|v_{sm} - \nabla f(x_{sm})\| \leq \frac{C\sigma}{\sqrt{B}} \quad \text{with probability } 1 - \zeta. \tag{94}$$

Now we will try to obtain a high probability bound for the variance term of other points beyond the starting points. Recall that $v_k = \frac{1}{b} \sum_{i \in I_b} (\nabla f_i(x_k) - \nabla f_i(x_{k-1})) + v_{k-1}$ (see Line 9 of Algorithm 1), we let $y_k := v_k - \nabla f(x_k)$ and $z_k := y_k - y_{k-1}$. It is not hard to verify that $\{y_k\}$ is a martingale sequence and $\{z_k\}$ is the associated martingale difference sequence. In order to apply the Azuma-Hoeffding inequalities to get a high probability bound, we first need to bound the difference sequence $\{z_k\}$. We use the Bernstein inequality to bound the differences as follows.

$$z_k = y_k - y_{k-1} = v_k - \nabla f(x_k) - (v_{k-1} - \nabla f(x_{k-1}))$$

$$= \frac{1}{b} \sum_{i \in I_b} (\nabla f_i(x_k) - \nabla f_i(x_{k-1})) + v_{k-1} - \nabla f(x_k) - (v_{k-1} - \nabla f(x_{k-1}))$$

$$= \frac{1}{b} \sum_{i \in I_b} \left( \nabla f_i(x_k) - \nabla f_i(x_{k-1}) - (\nabla f(x_k) - \nabla f(x_{k-1})) \right). \tag{95}$$

We define $u_i := \nabla f_i(x_k) - \nabla f_i(x_{k-1}) - (\nabla f(x_k) - \nabla f(x_{k-1}))$, and then we have

$$\|u_i\| = \|\nabla f_i(x_k) - \nabla f_i(x_{k-1}) - (\nabla f(x_k) - \nabla f(x_{k-1}))\| \le 2\|x_k - x_{k-1}\|, \qquad (96)$$

where the last inequality holds due to the gradient Lipschitz Assumption 1. Then, consider the variance term

$$\begin{aligned}
&\sum_{i \in I_b} \mathbb{E}[\|u_i\|^2] \\
&= \sum_{i \in I_b} \mathbb{E}[\|\nabla f_i(x_k) - \nabla f_i(x_{k-1}) - (\nabla f(x_k) - \nabla f(x_{k-1}))\|^2] \\
&\le \sum_{i \in I_b} \mathbb{E}[\|\nabla f_i(x_k) - \nabla f_i(x_{k-1})\|^2] \\
&\le bL^2 \|x_k - x_{k-1}\|^2, \qquad\qquad (97)
\end{aligned}$$

where the first inequality uses the fact $\mathbb{E}[\|x - \mathbb{E}x\|^2] \le \mathbb{E}[\|x\|^2]$, and the last inequality uses the gradient Lipschitz Assumption 1. According to (96) and (97), we can bound the difference $z_k$ by Bernstein inequality (Proposition 2) as

$$\begin{aligned}
\mathbb{P}\left\{\|z_k\| \ge \frac{t}{b}\right\} &\le (d+1)\exp\left(\frac{-t^2/2}{\sigma^2 + Rt/3}\right) \\
&= (d+1)\exp\left(\frac{-t^2/2}{bL^2\|x_k - x_{k-1}\|^2 + 2\|x_k - x_{k-1}\|t/3}\right) \\
&= \zeta_k,
\end{aligned}$$

where the last equality holds by letting $t = CL\sqrt{b}\|x_k - x_{k-1}\|$, where $C = O(\log\frac{d}{\zeta_k}) = \widetilde{O}(1)$. Now, we have a high probability bound for the difference sequence $\{z_k\}$, i.e.,

$$\|z_k\| \le c_k = \frac{CL\|x_k - x_{k-1}\|}{\sqrt{b}} \quad \text{with probability } 1 - \zeta_k. \qquad (98)$$

Now, we are ready to get a high probability bound for our original variance term (90) by using the martingale Azuma-Hoeffding inequality. Consider in a specific epoch $s$, i.e, iterations $t$ from $sm+1$ to current $sm+k$, where $k$ is less than $m$. According to Azuma-Hoeffding inequality (Proposition 4) and letting $\zeta_k = \zeta/m$, we have

$$\begin{aligned}
\mathbb{P}\left\{\|y_{sm+k} - y_{sm}\| \ge \beta\right\} &\le (d+1)\exp\left(\frac{-\beta^2}{8\sum_{t=sm+1}^{sm+k} c_t^2}\right) + \zeta \\
&= 2\zeta,
\end{aligned}$$

where the last equality holds by letting $\beta = \sqrt{8\sum_{t=sm+1}^{sm+k} c_t^2 \log\frac{d}{\zeta}} = \frac{C'L\sqrt{\sum_{t=sm+1}^{sm+k}\|x_t - x_{t-1}\|^2}}{\sqrt{b}}$, where $C' = O(C\sqrt{\log\frac{d}{\zeta}}) = \widetilde{O}(1)$. Recall that $y_k := v_k - \nabla f(x_k)$ and at the beginning point of this epoch $\|y_{sm}\| = \|v_{sm} - \nabla f(x_{sm})\| \le C\sigma/\sqrt{B}$ with probability $1-\zeta$, where $C = O(\log\frac{d}{\zeta}) = \widetilde{O}(1)$ (see (94)). Combining with (94) and using a union bound, we have

$$\|v_{t-1} - \nabla f(x_{t-1})\| = \|y_{t-1}\| \le \beta + \|y_{sm}\| \le \frac{C'L\sqrt{\sum_{j=sm+1}^{t-1}\|x_j - x_{j-1}\|^2}}{\sqrt{b}} + \frac{C\sigma}{\sqrt{B}} \qquad (99)$$

with probability $1 - 3\zeta$, where $t$ belongs to $[sm + 1, (s+1)m]$.

Now, we use this high probability version (99) instead of the expectation one (90) to obtain the high probability bound for function value decrease (see (92)). We sum up (86) from the beginning of this

epoch $s$, i.e., iterations from $sm$ to $t$, by plugging (99) into them to get:

$$f(x_t) \leq f(x_{sm}) - \frac{\eta}{2} \sum_{j=sm+1}^{t} \|\nabla f(x_{j-1})\|^2 - (\frac{1}{2\eta} - \frac{L}{2}) \sum_{j=sm+1}^{t} \|x_j - x_{j-1}\|^2$$

$$+ \frac{\eta}{2} \sum_{k=sm+1}^{t-1} \frac{2C'^2 L^2 \sum_{j=sm+1}^{k} \|x_j - x_{j-1}\|^2}{b} + \frac{\eta}{2} \sum_{j=sm+1}^{t} \frac{2C^2 \sigma^2}{B}$$

$$\leq f(x_{sm}) - \frac{\eta}{2} \sum_{j=sm+1}^{t} \|\nabla f(x_{j-1})\|^2 - (\frac{1}{2\eta} - \frac{L}{2}) \sum_{j=sm+1}^{t} \|x_j - x_{j-1}\|^2$$

$$+ \frac{\eta C'^2 L^2}{b} \sum_{k=sm+1}^{t-1} \sum_{j=sm+1}^{k} \|x_j - x_{j-1}\|^2 + \frac{(t-sm)\eta C^2 \sigma^2}{B}$$

$$\leq f(x_{sm}) - \frac{\eta}{2} \sum_{j=sm+1}^{t} \|\nabla f(x_{j-1})\|^2 - (\frac{1}{2\eta} - \frac{L}{2}) \sum_{j=sm+1}^{t} \|x_j - x_{j-1}\|^2$$

$$+ \frac{\eta C'^2 L^2 (t-1-sm)}{b} \sum_{j=sm+1}^{t} \|x_j - x_{j-1}\|^2 + \frac{(t-sm)\eta C^2 \sigma^2}{B}$$

$$\leq f(x_{sm}) - \frac{\eta}{2} \sum_{j=sm+1}^{t} \|\nabla f(x_{j-1})\|^2 - (\frac{1}{2\eta} - \frac{L}{2} - \eta C'^2 L^2) \sum_{j=sm+1}^{t} \|x_j - x_{j-1}\|^2$$

$$+ \frac{(t-sm)\eta C^2 \sigma^2}{B} \tag{100}$$

$$\leq f(x_{sm}) - \frac{\eta}{2} \sum_{j=sm+1}^{t} \|\nabla f(x_{j-1})\|^2 + \frac{(t-sm)\eta C^2 \sigma^2}{B}, \tag{101}$$

where (100) holds if the minibatch size $b \geq m$ (note that here $t \leq (s+1)m$), and (101) holds if the step size $\eta \leq \frac{\sqrt{8C'^2+1}-1}{4C'^2 L}$.

Similar to the previous finite-sum case, (101) only guarantees function value decrease when the summation of gradients in this epoch is large. However, in order to connect the guarantees between first situation (large gradients) and second situation (around saddle points), we need to show guarantees that are related to the *gradient of the starting point* of each epoch (see Line 3 of Algorithm 2). As we discussed in previous Section C.1.1, we achieve this by stopping the epoch at a uniformly random point (see Line 16 of Algorithm 2).

We want to point out that the second situation will have a bit difference due to (11), i.e., the full gradient of the starting point is not available (see Line 3 of Algorithm 2). Thus some modifications are needed for previous Lemma 1, we use the following lemma to connect these two situations (large gradients and around saddle points):

**Lemma 4 (Connection of Two Situations)** *For any epoch $s$, let $x_t$ be a point uniformly sampled from this epoch $\{x_j\}_{j=sm}^{(s+1)m}$. Moreover, let the step size $\eta \leq \frac{\sqrt{8C'^2+1}-1}{4C'^2 L}$ (where $C' = O(\log \frac{dm}{\zeta}) = \widetilde{O}(1)$), the minibatch size $b \geq m$ and batch size $B \geq \frac{256C^2\sigma^2}{g_{\text{thres}}^2}$ (where $C = O(\log \frac{d}{\zeta}) = \widetilde{O}(1)$), there are two cases:*

1. *If at least half of points in this epoch have gradient norm no larger than $\frac{g_{\text{thres}}}{2}$, then $\|\nabla f(x_{(s+1)m})\| \leq \frac{g_{\text{thres}}}{2}$ and $\|v_{(s+1)m}\| \leq g_{\text{thres}}$ hold with probability at least $1/3$;*

2. *Otherwise, we know $f(x_{sm}) - f(x_t) \geq \frac{7\eta m g_{\text{thres}}^2}{256}$ holds with probability at least $1/5$.*

*Moreover, $f(x_t) \leq f(x_{sm}) + \frac{(t-sm)\eta C^2 \sigma^2}{B}$ holds with high probability no matter which case happens.*

**Proof of Lemma 4.** There are two cases in this epoch:

1. If at least half of points of in this epoch $\{x_j\}_{j=sm}^{(s+1)m}$ have gradient norm no larger than $\frac{g_{\text{thres}}}{2}$, then it is easy to see that a uniformly sampled point $x_t$ has gradient norm $\|\nabla f(x_t)\| \leq \frac{g_{\text{thres}}}{2}$ with probability at least $1/2$. Moreover, note that the starting point of the next epoch $x_{(s+1)m} = x_t$ (i.e., Line 19 of Algorithm 2), thus we have $\|\nabla f(x_{(s+1)m})\| \leq \frac{g_{\text{thres}}}{2}$ with probability $1/2$. According to (94), we have $\|v_{(s+1)m} - \nabla f(x_{(s+1)m})\| \leq \frac{C\sigma}{\sqrt{B}}$ with probability $1 - \zeta$, where $C = O(\log \frac{d}{\zeta}) = \widetilde{O}(1)$. By a union bound, with probability at least $1/3$, we have

$$\|v_{(s+1)m}\| \leq \frac{C\sigma}{\sqrt{B}} + \frac{g_{\text{thres}}}{2} \leq \frac{g_{\text{thres}}}{16} + \frac{g_{\text{thres}}}{2} \leq g_{\text{thres}}.$$

2. Otherwise, at least half of points have gradient norm larger than $\frac{g_{\text{thres}}}{2}$. Then, as long as the sampled point $x_t$ falls into the last quarter of $\{x_j\}_{j=sm}^{(s+1)m}$, we know $\sum_{j=sm+1}^{t} \|\nabla f(x_{j-1})\|^2 \geq \frac{mg_{\text{thres}}^2}{16}$. This holds with probability at least $1/4$ since $x_t$ is uniformly sampled. Then by combining with (101), we obtain the function value decrease

$$f(x_{sm}) - f(x_t) \geq \frac{\eta}{2} \sum_{j=sm+1}^{t} \|\nabla f(x_{j-1})\|^2 - \frac{(t-sm)\eta C^2 \sigma^2}{B}$$

$$\geq \frac{\eta m g_{\text{thres}}^2}{32} - \frac{\eta m g_{\text{thres}}^2}{256} = \frac{7\eta m g_{\text{thres}}^2}{256},$$

where the last inequality is due to $B \geq \frac{256 C^2 \sigma^2}{g_{\text{thres}}^2}$. Note that (101) holds with high probability if we choose the minibatch size $b \geq m$ and the step size $\eta \leq \frac{\sqrt{8C'^2+1}-1}{4C'^2 L}$. By a union bound, the function value decrease $f(x_{sm}) - f(x_t) \geq \frac{7\eta m g_{\text{thres}}^2}{256}$ with probability at least $1/5$.

Again according to (101), $f(x_t) \leq f(x_{sm}) + \frac{(t-sm)\eta C^2 \sigma^2}{B}$ always holds with high probability. $\qquad\square$

Note that if Case 2 happens, the function value already decreases a lot in this epoch $s$ (corresponding to the first situation large gradients). Otherwise, Case 1 happens, we know the starting point of the next epoch $x_{(s+1)m} = x_t$ (i.e., Line 19 of Algorithm 2), then we know $\|\nabla f(x_{(s+1)m})\| \leq \frac{g_{\text{thres}}}{2}$ and $\|v_{(s+1)m}\| \leq g_{\text{thres}}$. Then we will start a super epoch (corresponding to the second situation around saddle points). Note that if $\lambda_{\min}(\nabla^2 f(x_{(s+1)m})) > -\delta$, this point $x_{(s+1)m}$ is already an $(\epsilon, \delta)$-second-order stationary point (recall that $g_{\text{thres}} \leq \epsilon$ in our Theorem 4).

**Around Saddle Points** $\|v_{(s+1)m}\| \leq g_{\text{thres}}$ and $\lambda_{\min}(\nabla^2 f(x_{(s+1)m})) \leq -\delta$: In this situation, we will show that the function value decreases a lot in a *super epoch* (instead of an epoch as in the first situation) with high probability by adding a random perturbation at the initial point $\widetilde{x} = x_{(s+1)m}$. To simplify the presentation, we use $x_0 := \widetilde{x} + \xi$ to denote the starting point of the super epoch after the perturbation, where $\xi$ uniformly $\sim \mathbb{B}_0(r)$ and the perturbation radius is $r$ (see Line 6 in Algorithm 2). Following the classical widely used *two-point analysis* developed in [18], we consider two coupled points $x_0$ and $x_0'$ with $w_0 := x_0 - x_0' = r_0 e_1$, where $r_0$ is a scalar and $e_1$ denotes the smallest eigenvector direction of Hessian $\mathcal{H} := \nabla^2 f(\widetilde{x})$. Then we get two coupled sequences $\{x_t\}$ and $\{x_t'\}$ by running SSRGD update steps (Line 8–12 of Algorithm 2) with the same choice of batches and minibatches (i.e., $I_B$'s (see (11) and Line 8) and $I_b$'s (see Line 12))for a super epoch. We will show that at least one of these two coupled sequences will decrease the function value a lot (escape the saddle point), i.e.,

$$\exists t \leq t_{\text{thres}}, \text{ such that } \max\{f(x_0) - f(x_t), f(x_0') - f(x_t')\} \geq 2f_{\text{thres}}. \tag{102}$$

We will prove (102) by contradiction. Assume the contrary, $f(x_0) - f(x_t) < 2f_{\text{thres}}$ and $f(x_0') - f(x_t') < 2f_{\text{thres}}$. First, we show that if function value does not decrease a lot, then all iteration points are not far from the starting point with high probability. Then we will show that the stuck region is relatively small in the random perturbation ball, i.e., at least one of $x_t$ and $x_t'$ will go far away from their starting point $x_0$ and $x_0'$ with high probability. Thus there is a contradiction. Similar to Lemma 2 and Lemma 3, we need the following two lemmas. Their proofs are deferred to the end of this section.

**Lemma 5 (Localization)** *Let $\{x_t\}$ denote the sequence by running SSRGD update steps (Line 8–12 of Algorithm 2) from $x_0$. Moreover, let the step size $\eta \leq \frac{1}{4C'L}$ and minibatch size $b \geq m$, with probability $1 - \zeta$, we have*

$$\forall t, \quad \|x_t - x_0\| \leq \sqrt{\frac{4t(f(x_0) - f(x_t))}{5C'L} + \frac{4t^2\eta C^2\sigma^2}{5C'LB}}, \tag{103}$$

*where $C' = O(\log \frac{dt}{\zeta}) = \widetilde{O}(1)$ and $C = O(\log \frac{dt}{\zeta m}) = \widetilde{O}(1)$.*

**Lemma 6 (Small Stuck Region)** *If the initial point $\widetilde{x}$ satisfies $-\gamma := \lambda_{\min}(\nabla^2 f(\widetilde{x})) \leq -\delta$, then let $\{x_t\}$ and $\{x'_t\}$ be two coupled sequences by running SSRGD update steps (Line 8–12 of Algorithm 2) with the same choice of batches and minibatches (i.e., $I_B$'s (see (11) and Line 8) and $I_b$'s (see Line 12)) from $x_0$ and $x'_0$ with $w_0 := x_0 - x'_0 = r_0 e_1$, where $x_0 \in \mathbb{B}_{\widetilde{x}}(r)$, $x'_0 \in \mathbb{B}_{\widetilde{x}}(r)$, $r_0 = \frac{\zeta' r}{\sqrt{d}}$ and $e_1$ denotes the smallest eigenvector direction of Hessian $\nabla^2 f(\widetilde{x})$. Moreover, let the super epoch length $t_{\text{thres}} = \frac{2\log(\frac{8\delta\sqrt{d}}{C_1\rho\zeta'r})}{\eta\delta} = \widetilde{O}(\frac{1}{\eta\delta})$, the step size $\eta \leq \min\left(\frac{1}{16\log(\frac{8\delta\sqrt{d}}{C_1\rho\zeta'r})L}, \frac{1}{8C_2L\log t_{\text{thres}}}\right) = \widetilde{O}(\frac{1}{L})$, minibatch size $b \geq m$, batch size $B = \widetilde{O}(\frac{\sigma^2}{g_{\text{thres}}^2})$ and the perturbation radius $r \leq \frac{\delta}{C_1\rho}$, then with probability $1 - \zeta$, we have*

$$\exists T \leq t_{\text{thres}}, \quad \max\{\|x_T - x_0\|, \|x'_T - x'_0\|\} \geq \frac{\delta}{C_1\rho}, \tag{104}$$

*where $C_1 \geq \frac{20C_2}{\eta L}$, $C_2 = O(\log \frac{dt_{\text{thres}}}{\zeta}) = \widetilde{O}(1)$ and $C'_2 = O(\log \frac{dt_{\text{thres}}}{\zeta m}) = \widetilde{O}(1)$.*

Based on these two lemmas, we are ready to show that (102) holds with high probability. Without loss of generality, we assume $\|x_T - x_0\| \geq \frac{\delta}{C_1\rho}$ in (104) (note that (103) holds for both $\{x_t\}$ and $\{x'_t\}$), then plugging it into (103) to obtain

$$\sqrt{\frac{4T(f(x_0) - f(x_T))}{5C'L} + \frac{4T^2\eta C^2\sigma^2}{5C'LB}} \geq \frac{\delta}{C_1\rho}$$

$$f(x_0) - f(x_T) \geq \frac{5C'L\delta^2}{4C_1^2\rho^2 T} - \frac{T\eta C^2\sigma^2}{B}$$

$$\geq \frac{5\eta C'L\delta^3}{8C_1^2\rho^2 \log(\frac{8\delta\sqrt{d}}{C_1\rho\zeta'r})} - \frac{2C^2\sigma^2\log(\frac{8\delta\sqrt{d}}{C_1\rho\zeta'r})}{B\delta} \tag{105}$$

$$\geq \frac{\delta^3}{C'_1\rho^2} \tag{106}$$

$$= 2f_{\text{thres}},$$

where (105) is due to $T \leq t_{\text{thres}}$ and (106) holds by letting $C'_1 = \frac{8C_1^2\log(\frac{8\delta\sqrt{d}}{C_1\rho\zeta'r})}{4\eta C'L}$. Recall that $B = \widetilde{O}(\frac{\sigma^2}{g_{\text{thres}}^2})$ and $g_{\text{thres}} \leq \delta^2/\rho$. Thus, we already prove that at least one of sequences $\{x_t\}$ and $\{x'_t\}$ escapes the saddle point with high probability, i.e.,

$$\exists T \leq t_{\text{thres}} \ , \max\{f(x_0) - f(x_T), f(x'_0) - f(x'_T)\} \geq 2f_{\text{thres}}, \tag{107}$$

if their starting points $x_0$ and $x'_0$ satisfying $w_0 := x_0 - x'_0 = r_0 e_1$, where $r_0 = \frac{\zeta' r}{\sqrt{d}}$ and $e_1$ denotes the smallest eigenvector direction of Hessian $\mathcal{H} := \nabla^2 f(\widetilde{x})$. Similar to the classical argument in [18], we know that in the random perturbation ball, the stuck points can only be a short interval in the $e_1$ direction, i.e., at least one of two points in the $e_1$ direction will escape the saddle point if their distance is larger than $r_0 = \frac{\zeta' r}{\sqrt{d}}$. Thus, we know that the probability of the starting point $x_0 = \widetilde{x} + \xi$ (where $\xi$ uniformly $\sim \mathbb{B}_0(r)$) located in the stuck region is less than

$$\frac{r_0 V_{d-1}(r)}{V_d(r)} = \frac{r_0 \Gamma(\frac{d}{2} + 1)}{\sqrt{\pi} r \Gamma(\frac{d}{2} + \frac{1}{2})} \leq \frac{r_0}{\sqrt{\pi} r}\left(\frac{d}{2} + 1\right)^{1/2} \leq \frac{r_0\sqrt{d}}{r} = \zeta', \tag{108}$$

where $V_d(r)$ denotes the volume of a Euclidean ball with radius $r$ in $d$ dimension, and the first inequality holds due to Gautschi's inequality. By a union bound for (108) and (106) (holds with high probability if $x_0$ is not in a stuck region), we know

$$f(x_0) - f(x_T) \geq 2f_{\text{thres}} = \frac{\delta^3}{C_1'\rho^2} \tag{109}$$

with high probability. Note that the initial point of this super epoch is $\widetilde{x}$ before the perturbation (see Line 6 of Algorithm 2), thus we need to show that the perturbation step $x_0 = \widetilde{x} + \xi$ (where $\xi$ uniformly $\sim \mathbb{B}_0(r)$) does not increase the function value a lot, i.e.,

$$\begin{aligned}
f(x_0) &\leq f(\widetilde{x}) + \langle \nabla f(\widetilde{x}), x_0 - \widetilde{x} \rangle + \frac{L}{2}\|x_0 - \widetilde{x}\|^2 \\
&\leq f(\widetilde{x}) + \|\nabla f(\widetilde{x})\|\|x_0 - \widetilde{x}\| + \frac{L}{2}\|x_0 - \widetilde{x}\|^2 \\
&\leq f(\widetilde{x}) + g_{\text{thres}} \cdot r + \frac{L}{2}r^2 \\
&\leq f(\widetilde{x}) + \frac{\delta^3}{2C_1'\rho^2} \\
&= f(\widetilde{x}) + f_{\text{thres}},
\end{aligned} \tag{110}$$

where the last inequality holds by letting the perturbation radius $r \leq \min\{\frac{\delta^3}{4C_1'\rho^2 g_{\text{thres}}}, \sqrt{\frac{\delta^3}{2C_1'\rho^2 L}}\}$.

Now we combine with (109) and (110) to obtain with high probability

$$f(\widetilde{x}) - f(x_T) = f(\widetilde{x}) - f(x_0) + f(x_0) - f(x_T) \geq -f_{\text{thres}} + 2f_{\text{thres}} = \frac{\delta^3}{2C_1'\rho^2}. \tag{111}$$

Thus we have finished the proof for the second situation (around saddle points), i.e., we show that the function value decrease a lot ($f_{\text{thres}} = \frac{\delta^3}{2C_1'\rho^2}$) in a *super epoch* (recall that $T \leq t_{\text{thres}} = \frac{2\log(\frac{8\delta\sqrt{d}}{C_1\rho\zeta'r})}{\eta\delta}$) by adding a random perturbation $\xi \sim \mathbb{B}_0(r)$ at the initial point $\widetilde{x}$.

**Combining these two situations (large gradients and around saddle points) to prove Theorem 4:** First, we recall Theorem 4 here since we want to recall the parameter setting.

**Theorem 4** *Under Assumption 1, 2 (i.e. (4) and (6)) and Assumption 4, let $\Delta f := f(x_0) - f^*$, where $x_0$ is the initial point and $f^*$ is the optimal value of $f$. By letting step size $\eta = \widetilde{O}(\frac{1}{L})$, batch size $B = \widetilde{O}(\frac{\sigma^2}{g_{\text{thres}}^2}) = \widetilde{O}(\frac{\sigma^2}{\epsilon^2})$, minibatch size $b = \sqrt{B} = \widetilde{O}(\frac{\sigma}{\epsilon})$, epoch length $m = b$, perturbation radius $r = \widetilde{O}\big(\min(\frac{\delta^3}{\rho^2\epsilon}, \frac{\delta^{3/2}}{\rho\sqrt{L}})\big)$, threshold gradient $g_{\text{thres}} = \epsilon \leq \delta^2/\rho$, threshold function value $f_{\text{thres}} = \widetilde{O}(\frac{\delta^3}{\rho^2})$ and super epoch length $t_{\text{thres}} = \widetilde{O}(\frac{1}{\eta\delta})$, SSRGD will at least once get to an $(\epsilon, \delta)$-second-order stationary point with high probability using*

$$\widetilde{O}\Big(\frac{L\Delta f\sigma}{\epsilon^3} + \frac{\rho^2\Delta f\sigma^2}{\epsilon^2\delta^3} + \frac{L\rho^2\Delta f\sigma}{\epsilon\delta^4}\Big)$$

*stochastic gradients for nonconvex online problem (2).*

**Proof of Theorem 4.** Now, we prove this theorem by distinguishing the epochs into three types as follows:

1. *Type-1 useful epoch*: If at least half of points in this epoch have gradient norm larger than $g_{\text{thres}}$ (Case 2 of Lemma 4);

2. *Wasted epoch*: If at least half of points in this epoch have gradient norm no larger than $g_{\text{thres}}$ and the starting point of the next epoch has estimated gradient norm larger than $g_{\text{thres}}$ (it means that this epoch does not guarantee decreasing the function value a lot as the large gradients situation, also it cannot connect to the second super epoch situation since the starting point of the next epoch has estimated gradient norm larger than $g_{\text{thres}}$);

3. *Type-2 useful super epoch*: If at least half of points in this epoch have gradient norm no larger than $g_{\mathrm{thres}}$ and the starting point of the next epoch (here we denote this point as $x_{(s+1)m}$) has estimated gradient norm no larger than $g_{\mathrm{thres}}$ (i.e., $\|v_{(s+1)m}\| \leq g_{\mathrm{thres}}$) (Case 1 of Lemma 4), according to Line 3 of Algorithm 2, we will start a super epoch. So here we denote this epoch along with its following super epoch as a type-2 useful super epoch.

First, it is easy to see that the probability of a wasted epoch happened is less than $2/3$ due to the random stop (see Case 1 of Lemma 4 and Line 16 of Algorithm 2) and different wasted epoch are independent. Thus, with high probability, there are at most $\widetilde{O}(1)$ wasted epochs happened before a type-1 useful epoch or type-2 useful super epoch. Now, we use $N_1$ and $N_2$ to denote the number of type-1 useful epochs and type-2 useful super epochs that the algorithm is needed. Recall that $\Delta f := f(x_0) - f^*$, where $x_0$ is the initial point and $f^*$ is the optimal value of $f$.

For type-1 useful epoch, according to Case 2 of Lemma 4, we know that the function value decreases at least $\frac{7\eta m g_{\mathrm{thres}}^2}{256}$ with probability at least $1/5$. Using a standard concentration, we know that with high probability $N_1$ type-1 useful epochs will decrease the function value at least $\frac{7\eta m g_{\mathrm{thres}}^2 N_1}{1536}$, note that the function value can decrease at most $\Delta f$. So $\frac{7\eta m g_{\mathrm{thres}}^2 N_1}{1536} \leq \Delta f$, we get $N_1 \leq \frac{1536\Delta f}{7\eta m g_{\mathrm{thres}}^2}$.

For type-2 useful super epoch, first we know that the starting point of the super epoch $\widetilde{x} := x_{(s+1)m}$ has gradient norm $\|\nabla f(\widetilde{x})\| \leq g_{\mathrm{thres}}/2$ and estimated gradient norm $\|v_{(s+1)m}\| \leq g_{\mathrm{thres}}$. Now if $\lambda_{\min}(\nabla^2 f(\widetilde{x})) \geq -\delta$, then $\widetilde{x}$ is already a $(\epsilon, \delta)$-second-order stationary point. Otherwise, $\|v_{(s+1)m}\| \leq g_{\mathrm{thres}}$ and $\lambda_{\min}(\nabla^2 f(\widetilde{x})) \leq -\delta$, this is exactly our second situation (around saddle points). According to (111), we know that the the function value decrease $(f(\widetilde{x}) - f(x_T))$ is at least $f_{\mathrm{thres}} = \frac{\delta^3}{2C_1'\rho^2}$ with high probability. Similar to type-1 useful epoch, we know $N_2 \leq \frac{C_1''\rho^2\Delta f}{\delta^3}$ by a union bound (so we change $C_1'$ to $C_1''$, anyway we also have $C_1'' = \widetilde{O}(1)$).

Now, we are ready to compute the convergence results to finish the proof for Theorem 4.

$$N_1(\widetilde{O}(1)B + B + mb) + N_2(\widetilde{O}(1)B + \lceil\frac{t_{\mathrm{thres}}}{m}\rceil B + t_{\mathrm{thres}}b) \tag{112}$$

$$\leq \widetilde{O}\Big(\frac{\Delta f\sigma}{\eta g_{\mathrm{thres}}^2\epsilon} + \frac{\rho^2\Delta f}{\delta^3}(\frac{\sigma^2}{\epsilon^2} + \frac{\sigma}{\eta\delta\epsilon})\Big)$$

$$\leq \widetilde{O}\Big(\frac{L\Delta f\sigma}{\epsilon^3} + \frac{\rho^2\Delta f\sigma^2}{\epsilon^2\delta^3} + \frac{L\rho^2\Delta f\sigma}{\epsilon\delta^4}\Big) \tag{113}$$

□

Now, the only remaining thing is to prove Lemma 5 and 6. We provide these two proofs as follows.

**Lemma 5 (Localization)** *Let $\{x_t\}$ denote the sequence by running SSRGD update steps (Line 8–12 of Algorithm 2) from $x_0$. Moreover, let the step size $\eta \leq \frac{1}{4C'L}$ and minibatch size $b \geq m$, with probability $1 - \zeta$, we have*

$$\forall t, \ \|x_t - x_0\| \leq \sqrt{\frac{4t(f(x_0) - f(x_t))}{5C'L} + \frac{4t^2\eta C^2\sigma^2}{5C'LB}},$$

*where $C' = O(\log\frac{dt}{\zeta}) = \widetilde{O}(1)$ and $C = O(\log\frac{dt}{\zeta m}) = \widetilde{O}(1)$.*

**Proof of Lemma 5.** First, we assume the variance bound (99) holds for all $0 \leq j \leq t-1$ (this is true with high probability using a union bound by letting $C' = O(\log\frac{dt}{\zeta})$ and $C = O(\log\frac{dt}{\zeta m})$). Then,

according to (100), we know for any $\tau \le t$ in some epoch $s$

$$f(x_\tau) \le f(x_{sm}) - \frac{\eta}{2} \sum_{j=sm+1}^{\tau} \|\nabla f(x_{j-1})\|^2 - \left(\frac{1}{2\eta} - \frac{L}{2} - \eta C'^2 L^2\right) \sum_{j=sm+1}^{\tau} \|x_j - x_{j-1}\|^2$$
$$+ \frac{(\tau - sm)\eta C^2 \sigma^2}{B}$$
$$\le f(x_{sm}) - \left(\frac{1}{2\eta} - \frac{L}{2} - \eta C'^2 L^2\right) \sum_{j=sm+1}^{\tau} \|x_j - x_{j-1}\|^2 + \frac{(\tau - sm)\eta C^2 \sigma^2}{B}$$
$$\le f(x_{sm}) - \frac{5C'L}{4} \sum_{j=sm+1}^{\tau} \|x_j - x_{j-1}\|^2 + \frac{(\tau - sm)\eta C^2 \sigma^2}{B}, \tag{114}$$

where the last inequality holds since the step size $\eta \le \frac{1}{4C'L}$ and assuming $C' \ge 1$. Now, we sum up (114) for all epochs before iteration $t$,

$$f(x_t) \le f(x_0) - \frac{5C'L}{4} \sum_{j=1}^{t} \|x_j - x_{j-1}\|^2 + \frac{t\eta C^2 \sigma^2}{B}.$$

Then, the proof is finished as

$$\|x_t - x_0\| \le \sum_{j=1}^{t} \|x_j - x_{j-1}\| \le \sqrt{t \sum_{j=1}^{t} \|x_j - x_{j-1}\|^2} \le \sqrt{\frac{4t(f(x_0) - f(x_t))}{5C'L} + \frac{4t^2 \eta C^2 \sigma^2}{5C'LB}}.$$

$\square$

**Lemma 6 (Small Stuck Region)** *If the initial point $\widetilde{x}$ satisfies $-\gamma := \lambda_{\min}(\nabla^2 f(\widetilde{x})) \le -\delta$, then let $\{x_t\}$ and $\{x'_t\}$ be two coupled sequences by running SSRGD update steps (Line 8–12 of Algorithm 2) with the same choice of batches and minibatches (i.e., $I_B$'s (see (11) and Line 8) and $I_b$'s (see Line 12)) from $x_0$ and $x'_0$ with $w_0 := x_0 - x'_0 = r_0 e_1$, where $x_0 \in \mathbb{B}_{\widetilde{x}}(r)$, $x'_0 \in \mathbb{B}_{\widetilde{x}}(r)$, $r_0 = \frac{\zeta' r}{\sqrt{d}}$ and $e_1$ denotes the smallest eigenvector direction of Hessian $\nabla^2 f(\widetilde{x})$. Moreover, let the super epoch length $t_{\mathrm{thres}} = \frac{2 \log(\frac{8\delta\sqrt{d}}{C_1 \rho \zeta' r})}{\eta\delta} = \widetilde{O}(\frac{1}{\eta\delta})$, the step size $\eta \le \min\left(\frac{1}{16\log(\frac{8\delta\sqrt{d}}{C_1 \rho \zeta' r})L}, \frac{1}{8C_2 L \log t_{\mathrm{thres}}}\right) = \widetilde{O}(\frac{1}{L})$, minibatch size $b \ge m$, batch size $B = \widetilde{O}(\frac{\sigma^2}{g_{\mathrm{thres}}^2})$ and the perturbation radius $r \le \frac{\delta}{C_1 \rho}$, then with probability $1 - \zeta$, we have*

$$\exists T \le t_{\mathrm{thres}}, \quad \max\{\|x_T - x_0\|, \|x'_T - x'_0\|\} \ge \frac{\delta}{C_1 \rho},$$

*where $C_1 \ge \frac{20C_2}{\eta L}$, $C_2 = O(\log \frac{dt_{\mathrm{thres}}}{\zeta}) = \widetilde{O}(1)$ and $C'_2 = O(\log \frac{dt_{\mathrm{thres}}}{\zeta m}) = \widetilde{O}(1)$.*

**Proof of Lemma 6.** We prove this lemma by contradiction. Assume the contrary,

$$\forall t \le t_{\mathrm{thres}} \; , \|x_t - x_0\| \le \frac{\delta}{C_1 \rho} \text{ and } \|x'_t - x'_0\| \le \frac{\delta}{C_1 \rho} \tag{115}$$

We will show that the distance between these two coupled sequences $w_t := x_t - x'_t$ will grow exponentially since they have a gap in the $e_1$ direction at the beginning, i.e., $w_0 := x_0 - x'_0 = r_0 e_1$, where $r_0 = \frac{\zeta' r}{\sqrt{d}}$ and $e_1$ denotes the smallest eigenvector direction of Hessian $\mathcal{H} := \nabla^2 f(\widetilde{x})$. However, $\|w_t\| = \|x_t - x'_t\| \le \|x_t - x_0\| + \|x_0 - \widetilde{x}\| + \|x'_t - x'_0\| + \|x'_0 - \widetilde{x}\| \le 2r + 2\frac{\delta}{C_1 \rho}$ according to (115) and the perturbation radius $r$. It is not hard to see that the exponential increase will break this upper bound, thus we get a contradiction.

In the following, we prove the exponential increase of $w_t$ by induction. First, we need the expression of $w_t$ (recall that $x_t = x_{t-1} - \eta v_{t-1}$ (see Line 11 of Algorithm 2)):

$$
\begin{aligned}
w_t &= w_{t-1} - \eta(v_{t-1} - v'_{t-1}) \\
&= w_{t-1} - \eta\big(\nabla f(x_{t-1}) - \nabla f(x'_{t-1}) + v_{t-1} - \nabla f(x_{t-1}) - v'_{t-1} + \nabla f(x'_{t-1})\big) \\
&= w_{t-1} - \eta\Big(\int_0^1 \nabla^2 f(x'_{t-1} + \theta(x_{t-1} - x'_{t-1}))d\theta(x_{t-1} - x'_{t-1}) \\
&\qquad\qquad\qquad + v_{t-1} - \nabla f(x_{t-1}) - v'_{t-1} + \nabla f(x'_{t-1})\Big) \\
&= (I - \eta\mathcal{H})w_{t-1} - \eta(\Delta_{t-1}w_{t-1} + y_{t-1}) \\
&= (I - \eta\mathcal{H})^t w_0 - \eta\sum_{\tau=0}^{t-1}(I - \eta\mathcal{H})^{t-1-\tau}(\Delta_\tau w_\tau + y_\tau)
\end{aligned}
\tag{116}
$$

where $\Delta_\tau := \int_0^1(\nabla^2 f(x'_\tau + \theta(x_\tau - x'_\tau)) - \mathcal{H})d\theta$ and $y_\tau := v_\tau - \nabla f(x_\tau) - v'_\tau + \nabla f(x'_\tau)$. Note that the first term of (116) is in the $e_1$ direction and is exponential with respect to $t$, i.e., $(1 + \eta\gamma)^t r_0 e_1$, where $-\gamma := \lambda_{\min}(\mathcal{H}) = \lambda_{\min}(\nabla^2 f(\widetilde{x})) \leq -\delta$. To prove the exponential increase of $w_t$, it is sufficient to show that the first term of (116) will dominate the second term. We inductively prove the following two bounds

1. $\frac{1}{2}(1 + \eta\gamma)^t r_0 \leq \|w_t\| \leq \frac{3}{2}(1 + \eta\gamma)^t r_0$
2. $\|y_t\| \leq 2\eta\gamma L(1 + \eta\gamma)^t r_0$

First, check the base case $t = 0$, $\|w_0\| = \|r_0 e_1\| = r_0$ holds for Bound 1. However, for Bound 2, we use Bernstein inequality (Proposition 2) to show that $\|y_0\| = \|v_0 - \nabla f(x_0) - v'_0 + \nabla f(x'_0)\| \leq \eta\gamma L r_0$. According to (11), we know that $v_0 = \frac{1}{B}\sum_{j\in I_B}\nabla f_j(x_0)$ and $v'_0 = \frac{1}{B}\sum_{j\in I_B}\nabla f_j(x'_0)$ (recall that these two coupled sequence $\{x_t\}$ and $\{x'_t\}$ use the same choice of batches and minibatches (i.e., $I_B$'s and $I_b$'s). Now, we have

$$
\begin{aligned}
y_0 &= v_0 - \nabla f(x_0) - v'_0 + \nabla f(x'_0) \\
&= \frac{1}{B}\sum_{j\in I_B}\nabla f_j(x_0) - \nabla f(x_0) - \frac{1}{B}\sum_{j\in I_B}\nabla f_j(x'_0) + \nabla f(x'_0) \\
&= \frac{1}{B}\sum_{j\in I_B}\Big(\nabla f_j(x_0) - \nabla f_j(x'_0) - (\nabla f(x_0) - \nabla f(x'_0))\Big).
\end{aligned}
\tag{117}
$$

We first bound each individual term of (117):

$$
\|\nabla f_j(x_0) - \nabla f_j(x'_0) - (\nabla f(x_0) - \nabla f(x'_0))\| \leq 2L\|x_0 - x'_0\| = 2L\|w_0\| = 2Lr_0, \tag{118}
$$

where the inequality holds due to the gradient Lipschitz Assumption 1. Then, consider the variance term of (117):

$$
\begin{aligned}
&\sum_{j\in I_B}\mathbb{E}[\|\nabla f_j(x_0) - \nabla f_j(x'_0) - (\nabla f(x_0) - \nabla f(x'_0))\|^2] \\
&\leq \sum_{j\in I_B}\mathbb{E}[\|\nabla f_j(x_0) - \nabla f_j(x'_0)\|^2] \\
&\leq BL^2\|x_0 - x'_0\|^2 \\
&= BL^2\|w_0\|^2 = BL^2 r_0^2,
\end{aligned}
\tag{119}
$$

where the first inequality uses the fact $\mathbb{E}[\|x - \mathbb{E}x\|^2] \leq \mathbb{E}[\|x\|^2]$, and the last inequality uses the gradient Lipschitz Assumption 1. According to (118) and (119), we can bound $y_0$ by Bernstein inequality (Proposition 2) as

$$
\begin{aligned}
\mathbb{P}\Big\{\|y_0\| \geq \frac{\alpha}{B}\Big\} &\leq (d+1)\exp\Big(\frac{-\alpha^2/2}{\sigma^2 + R\alpha/3}\Big) \\
&= (d+1)\exp\Big(\frac{-\alpha^2/2}{BL^2 r_0^2 + 2Lr_0\alpha/3}\Big) \\
&= \zeta,
\end{aligned}
$$

where the last equality holds by letting $\alpha = C_5 L\sqrt{B}r_0$, where $C_5 = O(\log \frac{d}{\zeta})$. Note that we can further relax the parameter $C_5$ to $C_2' = O(\log \frac{dt_{\text{thres}}}{\zeta m}) = \widetilde{O}(1)$ for making sure the above arguments hold with probability $1 - \zeta$ for all epoch starting points $y_{sm}$ with $sm \le t_{\text{thres}}$. Thus, we have with probability $1 - \zeta$,

$$\|y_0\| \le \frac{C_2' L r_0}{\sqrt{B}} \le \eta\gamma L r_0, \tag{120}$$

where the last inequality holds due to $B = \widetilde{O}(\frac{\sigma^2}{g_{\text{thres}}^2})$ (recall that $-\gamma := \lambda_{\min}(\mathcal{H}) = \lambda_{\min}(\nabla^2 f(\widetilde{x})) \le -\delta$ and $g_{\text{thres}} \le \delta^2/\rho$).

Now, we know that Bound 1 and Bound 2 hold for the base case $t = 0$ with high probability. Assume they hold for all $\tau \le t - 1$, we now prove they hold for $t$ one by one. For Bound 1, it is enough to show the second term of (116) is dominated by half of the first term.

$$\|\eta \sum_{\tau=0}^{t-1}(I - \eta\mathcal{H})^{t-1-\tau}(\Delta_\tau w_\tau)\| \le \eta \sum_{\tau=0}^{t-1}(1 + \eta\gamma)^{t-1-\tau}\|\Delta_\tau\|\|w_\tau\|$$

$$\le \frac{3}{2}\eta(1 + \eta\gamma)^{t-1}r_0 \sum_{\tau=0}^{t-1}\|\Delta_\tau\| \tag{121}$$

$$\le \frac{3}{2}\eta(1 + \eta\gamma)^{t-1}r_0 \sum_{\tau=0}^{t-1}\rho D_\tau^x \tag{122}$$

$$\le \frac{3}{2}\eta(1 + \eta\gamma)^{t-1}r_0 t\rho\left(\frac{\delta}{C_1\rho} + r\right) \tag{123}$$

$$\le \frac{3}{C_1}\eta\delta t(1 + \eta\gamma)^{t-1}r_0 \tag{124}$$

$$\le \frac{6\log(\frac{8\delta\sqrt{d}}{C_1\rho\zeta'r})}{C_1}(1 + \eta\gamma)^{t-1}r_0 \tag{125}$$

$$\le \frac{1}{4}(1 + \eta\gamma)^t r_0, \tag{126}$$

where (121) uses the induction for $w_\tau$ with $\tau \le t - 1$, (122) uses the definition $D_\tau^x := \max\{\|x_\tau - \widetilde{x}\|, \|x_\tau' - \widetilde{x}\|\}$, (123) follows from $\|x_t - \widetilde{x}\| \le \|x_t - x_0\| + \|x_0 - \widetilde{x}\| = \frac{\delta}{C_1\rho} + r$ due to (115) and the perturbation radius $r$, (124) holds by letting the perturbation radius $r \le \frac{\delta}{C_1\rho}$, (125) holds since $t \le t_{\text{thres}} = \frac{2\log(\frac{8\delta\sqrt{d}}{C_1\rho\zeta'r})}{\eta\delta}$, and (126) holds by letting $C_1 \ge 24\log(\frac{8\delta\sqrt{d}}{\rho\zeta'r})$.

$$\|\eta \sum_{\tau=0}^{t-1}(I - \eta\mathcal{H})^{t-1-\tau}y_\tau\| \le \eta \sum_{\tau=0}^{t-1}(1 + \eta\gamma)^{t-1-\tau}\|y_\tau\|$$

$$\le \eta \sum_{\tau=0}^{t-1}(1 + \eta\gamma)^{t-1-\tau}2\eta\gamma L(1 + \eta\gamma)^\tau r_0 \tag{127}$$

$$= 2\eta\eta\gamma Lt(1 + \eta\gamma)^{t-1}r_0$$

$$\le 2\eta\eta\gamma L\frac{2\log(\frac{8\delta\sqrt{d}}{C_1\rho\zeta'r})}{\eta\delta}(1 + \eta\gamma)^{t-1}r_0 \tag{128}$$

$$\le 4\eta\log(\frac{8\delta\sqrt{d}}{C_1\rho\zeta'r})L(1 + \eta\gamma)^{t-1}r_0 \tag{129}$$

$$\le \frac{1}{4}(1 + \eta\gamma)^t r_0, \tag{130}$$

where (127) uses the induction for $y_\tau$ with $\tau \le t - 1$, (128) holds since $t \le t_{\text{thres}} = \frac{2\log(\frac{8\delta\sqrt{d}}{C_1\rho\zeta'r})}{\eta\delta}$, (129) holds $\gamma \ge \delta$ (recall $-\gamma := \lambda_{\min}(\mathcal{H}) = \lambda_{\min}(\nabla^2 f(\widetilde{x})) \le -\delta$), and (130) holds by letting $\eta \le \frac{1}{16\log(\frac{8\delta\sqrt{d}}{C_1\rho\zeta'r})L}$.

Combining (126) and (130), we proved the second term of (116) is dominated by half of the first term. Note that the first term of (116) is $\|(I - \eta\mathcal{H})^t w_0\| = (1 + \eta\gamma)^t r_0$. Thus, we have

$$\frac{1}{2}(1 + \eta\gamma)^t r_0 \le \|w_t\| \le \frac{3}{2}(1 + \eta\gamma)^t r_0 \tag{131}$$

Now, the remaining thing is to prove the second bound $\|y_t\| \le \eta\gamma L(1 + \eta\gamma)^t r_0$. First, we write the concrete expression of $y_t$:

$$
\begin{aligned}
y_t &= v_t - \nabla f(x_t) - v'_t + \nabla f(x'_t) \\
&= \frac{1}{b}\sum_{i \in I_b} \left(\nabla f_i(x_t) - \nabla f_i(x_{t-1})\right) + v_{t-1} - \nabla f(x_t) \\
&\quad - \frac{1}{b}\sum_{i \in I_b}\left(\nabla f_i(x'_t) - \nabla f_i(x'_{t-1})\right) - v'_{t-1} + \nabla f(x'_t) \\
&= \frac{1}{b}\sum_{i \in I_b}\left(\nabla f_i(x_t) - \nabla f_i(x_{t-1})\right) + \nabla f(x_{t-1}) - \nabla f(x_t) \\
&\quad - \frac{1}{b}\sum_{i \in I_b}\left(\nabla f_i(x'_t) - \nabla f_i(x'_{t-1})\right) - \nabla f(x'_{t-1}) + \nabla f(x'_t) \\
&\quad + v_{t-1} - \nabla f(x_{t-1}) - v'_{t-1} + \nabla f(x'_{t-1}) \\
&= \frac{1}{b}\sum_{i \in I_b}\left(\nabla f_i(x_t) - \nabla f_i(x'_t) - \nabla f_i(x_{t-1}) + \nabla f_i(x'_{t-1})\right) \\
&\quad - \left(\nabla f(x_t) - \nabla f(x'_t) - \nabla f(x_{t-1}) + \nabla f(x'_{t-1})\right) + y_{t-1},
\end{aligned}
\tag{132}
$$

where (132) due to the definition of the estimator $v_t$ (see Line 12 of Algorithm 2). We further define the difference $z_t := y_t - y_{t-1}$. It is not hard to verify that $\{y_t\}$ is a martingale sequence and $\{z_t\}$ is the associated martingale difference sequence. We will apply the Azuma-Hoeffding inequalities to get an upper bound for $\|y_t\|$ and then we prove $\|y_t\| \le 2\eta\gamma L(1 + \eta\gamma)^t r_0$ based on that upper bound. In order to apply the Azuma-Hoeffding inequalities for martingale sequence $\|y_t\|$, we first need to bound the difference sequence $\{z_t\}$. We use the Bernstein inequality to bound the differences as follows.

$$
\begin{aligned}
z_t &= y_t - y_{t-1} \\
&= \frac{1}{b}\sum_{i \in I_b}\left(\nabla f_i(x_t) - \nabla f_i(x'_t) - \nabla f_i(x_{t-1}) + \nabla f_i(x'_{t-1})\right) \\
&\quad - \left(\nabla f(x_t) - \nabla f(x'_t) - \nabla f(x_{t-1}) + \nabla f(x'_{t-1})\right) \\
&= \frac{1}{b}\sum_{i \in I_b}\Big(\left(\nabla f_i(x_t) - \nabla f_i(x'_t)\right) - \left(\nabla f_i(x_{t-1}) - \nabla f_i(x'_{t-1})\right) \\
&\qquad - \left(\nabla f(x_t) - \nabla f(x'_t)\right) + \left(\nabla f(x_{t-1}) - \nabla f(x'_{t-1})\right)\Big).
\end{aligned}
\tag{133}
$$

We define $u_i := \left(\nabla f_i(x_t) - \nabla f_i(x'_t)\right) - \left(\nabla f_i(x_{t-1}) - \nabla f_i(x'_{t-1})\right) - \left(\nabla f(x_t) - \nabla f(x'_t)\right) + \left(\nabla f(x_{t-1}) - \nabla f(x'_{t-1})\right)$, and then we have

$$
\begin{aligned}
\|u_i\| &= \|\left(\nabla f_i(x_t) - \nabla f_i(x'_t)\right) - \left(\nabla f_i(x_{t-1}) - \nabla f_i(x'_{t-1})\right) \\
&\quad - \left(\nabla f(x_t) - \nabla f(x'_t)\right) + \left(\nabla f(x_{t-1}) - \nabla f(x'_{t-1})\right)\| \\
&\le \Big\| \int_0^1 \nabla^2 f_i(x'_t + \theta(x_t - x'_t))d\theta(x_t - x'_t) - \int_0^1 \nabla^2 f_i(x'_{t-1} + \theta(x_{t-1} - x'_{t-1}))d\theta(x_{t-1} - x'_{t-1}) \\
&\quad - \int_0^1 \nabla^2 f(x'_t + \theta(x_t - x'_t))d\theta(x_t - x'_t) + \int_0^1 \nabla^2 f(x'_{t-1} + \theta(x_{t-1} - x'_{t-1}))d\theta(x_{t-1} - x'_{t-1})\Big\| \\
&= \|\mathcal{H}_i w_t + \Delta_t^i w_t - (\mathcal{H}_i w_{t-1} + \Delta_{t-1}^i w_{t-1}) - (\mathcal{H} w_t + \Delta_t w_t) + (\mathcal{H} w_{t-1} + \Delta_{t-1} w_{t-1})\|
\end{aligned}
\tag{134}
$$

$$
\begin{aligned}
&\le \|(\mathcal{H}_i - \mathcal{H})(w_t - w_{t-1})\| + \|(\Delta_t^i - \Delta_t)w_t - (\Delta_{t-1}^i - \Delta_{t-1})w_{t-1}\| \\
&\le 2L\|w_t - w_{t-1}\| + 2\rho D_t^x\|w_t\| + 2\rho D_{t-1}^x\|w_{t-1}\|,
\end{aligned}
\tag{135}
$$

where (134) holds since we define $\Delta_t := \int_0^1 (\nabla^2 f(x'_t + \theta(x_t - x'_t)) - \mathcal{H}) d\theta$ and $\Delta_t^i := \int_0^1 (\nabla^2 f_i(x'_t + \theta(x_t - x'_t)) - \mathcal{H}_i) d\theta$, and the last inequality holds due to the gradient Lipschitz Assumption 1 and Hessian Lipschitz Assumption 2 (recall $D_t^x := \max\{\|x_t - \widetilde{x}\|, \|x'_t - \widetilde{x}\|\}$). Then, consider the variance term

$$
\begin{aligned}
&\sum_{i \in I_b} \mathbb{E}[\|u_i\|^2] \\
&\leq \sum_{i \in I_b} \mathbb{E}[\|(\nabla f_i(x_t) - \nabla f_i(x'_t)) - (\nabla f_i(x_{t-1}) - \nabla f_i(x'_{t-1}))\|^2] \\
&= \sum_{i \in I_b} \mathbb{E}[\|\mathcal{H}_i w_t + \Delta_t^i w_t - (\mathcal{H}_i w_{t-1} + \Delta_{t-1}^i w_{t-1})\|^2] \\
&\leq b(L\|w_t - w_{t-1}\| + \rho D_t^x \|w_t\| + \rho D_{t-1}^x \|w_{t-1}\|)^2,
\end{aligned}
\tag{136}
$$

where the first inequality uses the fact $\mathbb{E}[\|x - \mathbb{E}x\|^2] \leq \mathbb{E}[\|x\|^2]$, and the last inequality uses the gradient Lipschitz Assumption 1 and Hessian Lipschitz Assumption 2. According to (135) and (136), we can bound the difference $z_k$ by Bernstein inequality (Proposition 2) as (where $R = 2L\|w_t - w_{t-1}\| + 2\rho D_t^x \|w_t\| + 2\rho D_{t-1}^x \|w_{t-1}\|$ and $\sigma^2 = b(L\|w_t - w_{t-1}\| + \rho D_t^x \|w_t\| + \rho D_{t-1}^x \|w_{t-1}\|)^2$)

$$
\mathbb{P}\Big\{\|z_t\| \geq \frac{\alpha}{b}\Big\} \leq (d+1) \exp\Big(\frac{-\alpha^2/2}{\sigma^2 + R\alpha/3}\Big) = \zeta_k,
$$

where the last equality holds by letting $\alpha = C_4 \sqrt{b}(L\|w_t - w_{t-1}\| + \rho D_t^x \|w_t\| + \rho D_{t-1}^x \|w_{t-1}\|)$, where $C_4 = O(\log \frac{d}{\zeta_k}) = \widetilde{O}(1)$.

Now, we have a high probability bound for the difference sequence $\{z_k\}$, i.e.,

$$
\|z_k\| \leq c_k = \frac{C_4(L\|w_t - w_{t-1}\| + \rho D_t^x \|w_t\| + \rho D_{t-1}^x \|w_{t-1}\|)}{\sqrt{b}} \quad \text{with probability } 1 - \zeta_k.
\tag{137}
$$

Now, we are ready to get an upper bound for $y_t$ by using the martingale Azuma-Hoeffding inequality. Note that we only need to focus on the current epoch that contains the iteration $t$ since the martingale sequence $\{y_t\}$ starts with a new point $y_{sm}$ for each epoch $s$ due to the estimator $v_{sm}$. Also note that the starting point $y_{sm}$ can be bounded with the same upper bound (120) for all epoch $s$. Let $s$ denote the current epoch, i.e, iterations from $sm + 1$ to current $t$, where $t$ is no larger than $(s+1)m$. According to Azuma-Hoeffding inequality (Proposition 4) and letting $\zeta_k = \zeta/m$, we have

$$
\mathbb{P}\Big\{\|y_t - y_{sm}\| \geq \beta\Big\} \leq (d+1) \exp\Big(\frac{-\beta^2}{8 \sum_{k=sm+1}^t c_k^2}\Big) + \zeta
$$
$$
= 2\zeta,
$$

where the last equality is due to $\beta = \sqrt{8 \sum_{k=sm+1}^t c_k^2 \log \frac{d}{\zeta}} = \frac{C_3 \sqrt{\sum_{k=sm+1}^t (L\|w_t - w_{t-1}\| + \rho D_t^x \|w_t\| + \rho D_{t-1}^x \|w_{t-1}\|)^2}}{\sqrt{b}}$, where $C_3 = O(C_4 \sqrt{\log \frac{d}{\zeta}}) = \widetilde{O}(1)$. Recall that $y_k := v_k - \nabla f(x_k) - v'_k + \nabla f(x'_k)$ and at the beginning point of this epoch $y_{sm} = \|v_{sm} - \nabla f(x_{sm}) - v'_{sm} + \nabla f(x'_{sm})\| \leq \eta\gamma L r_0$ with probability $1 - \zeta$ (see (120)). Combining with (120) and using a union bound, we have

$$
\|y_t\| \leq \beta + \|y_{sm}\| \leq \frac{C_3 \sqrt{\sum_{k=sm+1}^t (L\|w_t - w_{t-1}\| + \rho D_t^x \|w_t\| + \rho D_{t-1}^x \|w_{t-1}\|)^2}}{\sqrt{b}} + \eta\gamma L r_0
\tag{138}
$$

with probability $1 - 3\zeta$, where $t$ belongs to $[sm+1, (s+1)m]$. Note that we can further relax the parameter $C_3$ in (138) to $C_2 = O(\log \frac{dt_{\text{thres}}}{\zeta})$ (see (139)) for making sure the above arguments hold with probability $1 - \zeta$ for all $t \leq t_{\text{thres}}$ by using a union bound for $\zeta_t$'s:

$$\|y_t\| \le \frac{C_2\sqrt{\sum_{k=sm+1}^{t}(L\|w_t-w_{t-1}\|+\rho D_t^x\|w_t\|+\rho D_{t-1}^x\|w_{t-1}\|)^2}}{\sqrt{b}} + \eta\gamma L r_0, \qquad (139)$$

where $t$ belongs to $[sm+1,(s+1)m]$.

Now, we will show how to bound the right-hand-side of (139) to finish the proof, i.e., prove the remaining second bound $\|y_t\| \le 2\eta\gamma L(1+\eta\gamma)^t r_0$.

First, we show that the last two terms in the first term of right-hand-side of (139) can be bounded as

$$\rho D_t^x\|w_t\| + \rho D_{t-1}^x\|w_{t-1}\| \le \rho\Big(\frac{\delta}{C_1\rho}+r\Big)\frac{3}{2}(1+\eta\gamma)^t r_0 + \rho\Big(\frac{\delta}{C_1\rho}+r\Big)\frac{3}{2}(1+\eta\gamma)^{t-1}r_0$$

$$\le 3\rho\Big(\frac{\delta}{C_1\rho}+r\Big)(1+\eta\gamma)^t r_0$$

$$\le \frac{6\delta}{C_1}(1+\eta\gamma)^t r_0, \qquad (140)$$

where the first inequality follows from the induction of $\|w_{t-1}\| \le \frac{3}{2}(1+\eta\gamma)^{t-1}r_0$ and the already proved $\|w_t\| \le \frac{3}{2}(1+\eta\gamma)^t r_0$ in (131), and the last inequality holds by letting the perturbation radius $r \le \frac{\delta}{C_1\rho}$.

Now, we show that the first term in (139) can be bounded as

$$L\|w_t - w_{t-1}\|$$

$$= L\Big\| - \eta\mathcal{H}(I-\eta\mathcal{H})^{t-1}w_0 - \eta\sum_{\tau=0}^{t-2}\eta\mathcal{H}(I-\eta\mathcal{H})^{t-2-\tau}(\Delta_\tau w_\tau + y_\tau) + \eta(\Delta_{t-1}w_{t-1}+y_{t-1})\Big\|$$

$$\le L\eta\gamma(1+\eta\gamma)^{t-1}r_0 + L\Big\|\eta\sum_{\tau=0}^{t-2}\eta\mathcal{H}(I-\eta\mathcal{H})^{t-2-\tau}(\Delta_\tau w_\tau + y_\tau)\Big\| + L\|\eta(\Delta_{t-1}w_{t-1}+y_{t-1})\|$$

$$\le L\eta\gamma(1+\eta\gamma)^{t-1}r_0 + L\eta\Big\|\sum_{\tau=0}^{t-2}\eta\mathcal{H}(I-\eta\mathcal{H})^{t-2-\tau}\Big\| \max_{0\le k\le t-2}\|\Delta_k w_k + y_k\|$$

$$\quad + L\eta\rho\Big(\frac{\delta}{C_1\rho}+r\Big)\|w_{t-1}\| + L\eta\|y_{t-1}\| \qquad (141)$$

$$\le L\eta\gamma(1+\eta\gamma)^{t-1}r_0 + L\eta\sum_{\tau=0}^{t-2}\frac{1}{t-1-\tau}\max_{0\le k\le t-2}\|\Delta_k w_k + y_k\|$$

$$\quad + L\eta\rho\Big(\frac{\delta}{C_1\rho}+r\Big)\|w_{t-1}\| + L\eta\|y_{t-1}\| \qquad (142)$$

$$\le L\eta\gamma(1+\eta\gamma)^{t-1}r_0 + L\eta\log t \max_{0\le k\le t-2}\|\Delta_k w_k + y_k\|$$

$$\quad + L\eta\rho\Big(\frac{\delta}{C_1\rho}+r\Big)\|w_{t-1}\| + L\eta\|y_{t-1}\|$$

$$\le L\eta\gamma(1+\eta\gamma)^{t-1}r_0 + L\eta\log t \max_{0\le k\le t-2}\|\Delta_k w_k + y_k\|$$

$$\quad + L\eta\rho\Big(\frac{\delta}{C_1\rho}+r\Big)\frac{3}{2}(1+\eta\gamma)^{t-1}r_0 + 2L\eta\eta\gamma L(1+\eta\gamma)^{t-1}r_0 \qquad (143)$$

$$\le L\eta\gamma(1+\eta\gamma)^{t-1}r_0 + L\eta\log t\Big(\rho\Big(\frac{\delta}{C_1\rho}+r\Big)\frac{3}{2}(1+\eta\gamma)^{t-2}r_0 + 2\eta\gamma L(1+\eta\gamma)^{t-2}r_0\Big)$$

$$\quad + L\eta\rho\Big(\frac{\delta}{C_1\rho}+r\Big)\frac{3}{2}(1+\eta\gamma)^{t-1}r_0 + 2L\eta\eta\gamma L(1+\eta\gamma)^{t-1}r_0 \qquad (144)$$

$$\leq L\eta\gamma(1+\eta\gamma)^{t-1}r_0 + L\eta\log t\Big(\frac{3\delta}{C_1}(1+\eta\gamma)^{t-2}r_0 + 2\eta\gamma L(1+\eta\gamma)^{t-2}r_0\Big)$$

$$+ \frac{3L\eta\delta}{C_1}(1+\eta\gamma)^{t-1}r_0 + 2L\eta\eta\gamma L(1+\eta\gamma)^{t-1}r_0 \tag{145}$$

$$\leq \Big(\frac{4}{C_1}\log t + 4L\eta\log t\Big)\eta\gamma L(1+\eta\gamma)^t r_0, \tag{146}$$

where the first equality follows from (116), (141) holds from the following (147),

$$\|\Delta_t\| \leq \rho D_t^x \leq \rho\Big(\frac{\delta}{C_1\rho}+r\Big), \tag{147}$$

where (147) holds due to Hessian Lipschitz Assumption 2, (115) and the perturbation radius $r$ (recall that $\Delta_t := \int_0^1 (\nabla^2 f(x_t' + \theta(x_t - x_t')) - \mathcal{H})d\theta$, $\mathcal{H} := \nabla^2 f(\widetilde{x})$ and $D_t^x := \max\{\|x_t - \widetilde{x}\|, \|x_t' - \widetilde{x}\|\}$), (142) holds due to $\|\eta\mathcal{H}(I-\eta\mathcal{H})^t\| \leq \frac{1}{t+1}$, (143) holds by plugging the induction $\|w_{t-1}\| \leq \frac{3}{2}(1+\eta\gamma)^{t-1}r_0$ and $\|y_{t-1}\| \leq 2\eta\gamma L(1+\eta\gamma)^{t-1}r_0$, (144) follows from (147), the induction $\|w_k\| \leq \frac{3}{2}(1+\eta\gamma)^k r_0$ and $\|y_k\| \leq 2\eta\gamma L(1+\eta\gamma)^k r_0$ (hold for all $k \leq t-1$), (145) holds by letting the perturbation radius $r \leq \frac{\delta}{C_1\rho}$, and the last inequality holds due to $\gamma \geq \delta$ (recall $-\gamma := \lambda_{\min}(\mathcal{H}) = \lambda_{\min}(\nabla^2 f(\widetilde{x})) \leq -\delta$).

By plugging (140) and (146) into (139), we have

$$\|y_t\| \leq C_2\left(\frac{6\delta}{C_1}(1+\eta\gamma)^t r_0 + \Big(\frac{4}{C_1}\log t + 4L\eta\log t\Big)\eta\gamma L(1+\eta\gamma)^t r_0\right) + \eta\gamma L r_0$$

$$\leq C_2\Big(\frac{6}{C_1\eta L} + \frac{4}{C_1}\log t + 4L\eta\log t\Big)\eta\gamma L(1+\eta\gamma)^t r_0 + \eta\gamma L r_0$$

$$\leq 2\eta\gamma L(1+\eta\gamma)^t r_0, \tag{148}$$

where the second inequality holds due to $\gamma \geq \delta$, and the last inequality holds by letting $C_1 \geq \frac{20C_2}{\eta L}$ and $\eta \leq \frac{1}{8C_2 L\log t}$. Recall that $C_2 = O(\log\frac{dt_{\text{thres}}}{\zeta})$ is enough to let the arguments in this proof hold with probability $1-\zeta$ for all $t \leq t_{\text{thres}}$.

From (131) and (148), we know that the two induction bounds hold for $t$. We recall the first induction bound here:

1. $\frac{1}{2}(1+\eta\gamma)^t r_0 \leq \|w_t\| \leq \frac{3}{2}(1+\eta\gamma)^t r_0$

Thus, we know that $\|w_t\| \geq \frac{1}{2}(1+\eta\gamma)^t r_0 = \frac{1}{2}(1+\eta\gamma)^t \frac{\zeta' r}{\sqrt{d}}$. However, $\|w_t\| := \|x_t - x_t'\| \leq \|x_t - x_0\| + \|x_0 - \widetilde{x}\| + \|x_t' - x_0'\| + \|x_0' - \widetilde{x}\| \leq 2r + 2\frac{\delta}{C_1\rho} \leq \frac{4\delta}{C_1\rho}$ according to (115) and the perturbation radius $r$. The last inequality is due to the perturbation radius $r \leq \frac{\delta}{C_1\rho}$ (we already used this condition in the previous arguments). This will give a contradiction for (115) if $\frac{1}{2}(1+\eta\gamma)^t \frac{\zeta' r}{\sqrt{d}} \geq \frac{4\delta}{C_1\rho}$ and it will happen if $t \geq \frac{2\log(\frac{8\delta\sqrt{d}}{C_1\rho\zeta' r})}{\eta\delta}$.

So the proof of this lemma is finished by contradiction if we let $t_{\text{thres}} := \frac{2\log(\frac{8\delta\sqrt{d}}{C_1\rho\zeta' r})}{\eta\delta}$, i.e., we have

$$\exists T \leq t_{\text{thres}}, \quad \max\{\|x_T - x_0\|, \|x_T' - x_0'\|\} \geq \frac{\delta}{C_1\rho}.$$

$\square$