[Reviews · NeurIPS 2019]

Reviewer 1



============================================================== After response----I keep my evaluation on the technical innovation and suboptimality of this paper. 1. The basic Spider and SpiderBoost algorithms are both for first-order stationary point, they are almost the same, and both give n^{1/2} rate. The simple way to modify both algorithms to escape saddle point is to add Negative Curvature Search (NCS) subroutine (which can be done in a very modular way, and is already shown in the Spider paper). I'd say it's almost trivial to also show SpiderBoost+NCS to find second-order stationary point with n^{1/2} rate. Comparing this paper with SpiderBoost+NCS, there's no improvement from n^{2/3} to n^{1/2} (since Spiderboost is already n^{1/2}), no simplification of Spider (as Spiderboost already did so). The only difference is replacing NCS by perturbations, which again requires some work, but most techniques are already there. 2. There are two error quantities in the guarantee---epsilon and delta. When people talk about optimal rate, it should be optimal respect to both epsilon and delta, i.e. for any choice of the pair. It is already suboptimal when one says they can match the best rate only in some restricted range. It is also clear from Table 2 when we choose epsilon = delta^2, which is a nature setting in most previous paper, this paper gives 1/epsilon^{3.5}, while SPIDER-SFO + (+Neon2) gives 1/epsilon^3. It is clearly suboptimal. ============================================================== I think the result is solid, but this paper lacks innovation since most techniques are available in the field. It uses SPIDER [11] to do variance reduction and achieve nearly optimal rate, and uses perturbations to escape saddle points which is also available in [15, 18]. It seems to be the combination of both paper, and the rate in escaping saddle points is actually worse than [11]. I do not view the contribution to be significant enough to be published in NIPS, especially the improvement is rather straightforward and incremental compared to [15].

Reviewer 2



- originality: This paper proposes a novel and simple algorithm called SSRGD for finding local minima in nonconvex optimization. - quality: This work is solid. - clarity: This paper is well-written and easy to follow. - significance: The proposed SSRGD algorithm does not need an extra negative-curvature search subroutine unlike previous algorithms and improves previous results to the almost optimal results.

Reviewer 3



This paper proposes a simple algorithm for escaping saddle points in nonconvex optimization. Unlike previous algorithms, the proposed SSRGD algorithm does not need an extra negative-curvature search subroutine and it only needs to add a uniform perturbation sometimes. Thus it is more attractive and easy to use in practice. Moreover, the authors prove that the convergence results of SSRGD improve previous results and are near-optimal now. The authors also give a clear interpretation for the comparison of the proposed SSRGD and SVRG algorithm, which is very useful for better understanding these two algorithms. The paper is well-written and the proofs are easy to follow.

Reviewer 4



=== after author's rebuttal === Although the rebuttal did not address much about my concern about a high level intuition of the improvement, and R1's opinion that is incremental, I think this is still a good result, so I suggest for a poster admission. === after author's rebuttal === Originality: This paper reviews many related versions and improve the theoretical bound upon them. I believe it is a totally novel analysis. Quality: This paper gives detailed analysis and the flow of the proof is clear and inspiring. Clarity: The main paper gives the high level sketch of their proof and also the intuition why they can improve the existing result. The supplement is also well organized with full details. Significance: Improves the known bound and match the $n^{-1/2}$ lower bound. I think it's of high significance.

[Author Response · NeurIPS 2019]

We thank the reviewers for appreciating our work, and for their insightful and constructive comments.

**Response to Reviewer #1:**

**1. Regarding the innovation of this paper.**

**A1:** For the comparison with [15], Ge et al. [15] uses the SVRG estimator $v_t = \frac{1}{b} \sum_{i \in I_b} \left( \nabla f_i(x_t) - \nabla f_i(\tilde{x}) \right) + \nabla f(\tilde{x})$
which reuses a fixed snapshot full gradient $\nabla f(\tilde{x})$, and achieves a suboptimal convergence $\tilde{O}(n^{2/3}/\epsilon^2)$. Here we use the
stochastic recursive gradient estimator $v_t = \frac{1}{b} \sum_{i \in I_b} \left( \nabla f_i(x_t) - \nabla f_i(x_{t-1}) \right) + v_{t-1}$ to obtain the improved optimal
convergence $\tilde{O}(n^{1/2}/\epsilon^2)$ for escaping saddle points which matches the lower bound $\Omega(n^{1/2}/\epsilon^2)$. Moreover, we also
have discussed with the authors of [15], we cannot directly add a perturbation step to SPIDER [11] for escaping saddle
points. The original SPIDER [11] uses an additional negative-curvature search subroutine (Neon2 [4]) to escape saddle
points. Thus, in this paper, we use a few simple but effective modifications to obtain a simple algorithm for achieving
the new convergence results. There are three differences between our SSRGD algorithm and SPIDER. 1) SPIDER [11]
uses the normalized gradient update $x_{t+1} = x_t - \eta(v_t/\|v_t\|)$, while our SSRGD uses the direct update $x_{t+1} = x_t - \eta v_t$,
where $v_t$ is the gradient direction estimator. 2) SPIDER uses a very small step size $\eta = O(\epsilon/L)$, while our SSRGD
uses the more standard step size $\eta = O(1/L)$. 3) SPIDER uses an additional negative-curvature search subroutine (e.g.,
Neon2 [4]) to escape saddle points, while our SSRGD directly escapes saddle points by adding a uniform perturbation
sometimes. These key differences are crucial for achieving the new convergence result $\tilde{O}(n^{1/2}/\epsilon^2)$ and also show that
our SSRGD is more attractive in practice. Intuitively, in the large gradient situations (e.g., at the beginning phase),
SPIDER goes very slowly due to the normalized gradient update. Also note that SPIDER can still go slowly even
in the small gradient situations due to its tiny step size. As a result, SPIDER may get worse convergence results in
some situations (see the following response A2). Moreover, our SSRGD which uses direct update (non-normalized),
standard step size and simple perturbation step (no negative-curvature search subroutine) is more attractive and easy to
implement in practice. Besides, we believe our simple/natural analysis will be useful for future work.

**2. Regarding the convergence results.**

**A2:** We would like to point out that our convergence rate is not worse than SPIDER [11]. Moreover, our SSRGD is
better than SPIDER if $\delta \leq 1/\sqrt{n}$ (i.e., high second-order accuracy situation) due to the $1/\delta^5$ term in SPIDER (see our
Table 1). Concretely, for achieving an $(\epsilon, \delta)$-second-order stationary point ($\|\nabla f(x)\| \leq \epsilon$ and $\lambda_{\min}(\nabla^2 f(x)) \geq -\delta$),
the last three algorithms in our Table 1 can obtain the best rate $\tilde{O}(n^{1/2}/\epsilon^2)$ in a large range of $\epsilon$ and $\delta$. SNVRG+Neon2
[32] and our SSRGD achieve the rate $\tilde{O}(n^{1/2}/\epsilon^2)$ if $1/\epsilon \geq 1/\delta^2$ and $n \leq 1/\epsilon$. SPIDER+Neon2 [11] achieves the rate
$\tilde{O}(n^{1/2}/\epsilon^2)$ if $1/\epsilon \geq 1/\delta^2$ and $n \geq 1/\epsilon$. Similarly, for the online case (Table 2), SNVRG+Neon2 [32], SPIDER+Neon2
[11] and our SSRGD can also obtain the best rate $\tilde{O}(1/\epsilon^3)$ in a large range of $\epsilon$ and $\delta$ for achieving an $(\epsilon, \delta)$-second-
order stationary point, i.e., SNVRG+Neon2 [32] and our SSRGD achieve the best rate $\tilde{O}(1/\epsilon^3)$ if $1/\delta^3 \leq 1/\epsilon$, and
SPIDER+Neon2 [11] achieves the best rate $\tilde{O}(1/\epsilon^3)$ if $1/\delta^2 \leq 1/\epsilon$.

**Response to Reviewer #2:**

**Future directions:** The achieved convergence rate $O(n^{1/2}/\epsilon^2)$ matches the lower bound $\Omega(n^{1/2}/\epsilon^2)$ in finite-sum
case. However, whether $\Omega(1/\epsilon^3)$ is the lower bound for online case is unknown. This should be an important future
problem. The combination of the stabilized trick proposed in Ge et al. [15] is also indeed an interesting extension.

**Experiments:** We will try to add some experiments although previous related papers also did not do the experiments.

**Response to Reviewer #3:**

**Regarding the difficulties to avoid using the negative-curvature search subroutine:** Previous algorithms use the
negative-curvature search subroutine (e.g. Neon2) to find the approximate smallest eigenvector of the Hessian (i.e.,
decreasing direction) near saddle points. However, if we just use a uniform perturbation step, it is not very easy to
obtain the smallest eigenvector direction since we only roughly have $O(r/\sqrt{d})$ amount in the smallest eigenvector
direction in a uniform perturbation ball with radius $r$ in $\mathbb{R}^d$ space. In our analysis, by simultaneously bounding the
coupled distance and coupled variance, we can show that the amount in the smallest eigenvector direction will increase
exponentially and thus avoid the negative-curvature search subroutine.

**Response to Reviewer #4:**

**Subroutine:** The subroutine Neon [31] extracts the negative curvature based on a perturbation step, power method and
Nesterov's accelerated gradient method. Neon2 [4] is based on a perturbation step, Oja's algorithm and Chebyshev
polynomial. Thus they are at least more complicated in implementation than just a perturbation step as our SSRGD
used. Please see A2 in Response to Reviewer #1 for the convergence comparison among the last three algorithms.

**Optimal among simple algorithms:** Note that Du et al. [10] showed that it is necessary to add the perturbation step
for gradient descent to escape saddle points efficiently. Our SSRGD indeed just adds a perturbation step on the simple
recursive gradient descent to achieve the optimal convergence $\tilde{O}(n^{1/2}/\epsilon^2)$. **Improvement/Comparison:** Intuitively, it
can improve upon SVRG [21] since it uses more aggressive recursive gradient estimator instead of the conservative
SVRG estimator which uses a fixed full gradient snapshot. Please see A1 in Response to Reviewer #1 for more details
and a comparison with SPIDER [11], and see Response to Reviewer #3 for the discussion of saddle point iterations.

[Meta-Review · NeurIPS 2019]

After extensive back and forth discussion by the reviewers, ultimately we felt that despite trivial solutions that are possible by mixing spider with negative curvature search, the approach of the present paper has its usefulness (as noted in the reviews too), and that the paper can be accepted. The authors are, however, strongly encouraged to look into the reviews carefully, and implement the points mentioned in the rebuttal -- because it would be a pity if after all the back-n-forth that this paper's review cycle has witnessed, if the authors did not update the paper to clarify its contribution, its value, and its contrasts with less implementable methods.